# Gradient Flow Through Diagram Expansions: Learning Regimes and Explicit Solutions

**Dmitry Yarotsky** [1 2]   **Eugene Golikov** [1]   **Yaroslav Gusev** [1]

## Abstract

We develop a general mathematical framework to analyze scaling regimes and derive explicit analytic solutions for gradient flow (GF) in large learning problems. Our key innovation is a formal power series expansion of the loss evolution, with coefficients encoded by diagrams akin to Feynman diagrams. We show that this expansion has a well-defined large-size limit that can be used to reveal different learning phases and, in some cases, to obtain explicit solutions of the nonlinear GF. We focus on learning Canonical Polyadic (CP) decompositions of high-order tensors, and show that this model has several distinct extreme lazy and rich GF regimes such as free evolution, NTK and under- and over-parameterized mean-field. We show that these regimes depend on the parameter scaling, tensor order, and symmetry of the model in a specific and subtle way. Moreover, we propose a general approach to summing the formal loss expansion by reducing it to a PDE; in a wide range of scenarios, it turns out to be first-order and solvable by the method of characteristics. We observe a very good agreement of our theoretical predictions with experimental results.

## 1. Introduction

**Motivation.** A major theoretical challenge in modern machine learning is a quantitative description of gradient-descent-based learning of large-scale models. The difficulty arises from the high complexity of the learning dynamics, which leads to qualitatively different limit learning regimes; see A.1, A.2 for a survey. Among the limit learning regimes, explicit solutions are known for only a few. A notable example is the NTK limit (Jacot et al., 2018), for which learning

becomes linear in its parameters, therefore no *feature learning* occurs. A complete classification of large-scale learning regimes together with explicit limit solutions, including nonlinear ones, is still an open problem for neural nets.

**Our contribution.** In this work we propose a new general mathematical framework for the analysis of gradient flow in large-scale problems. The central element of this framework is a power series expansion of the loss evolution with respect to time. The coefficients of this expansion can be described in terms of graphs akin to Feynman diagrams used in physics. We develop a general mathematical theory of such diagrams and show that it can be used to describe different learning regimes, and, in some cases, obtain explicit formulas for the loss evolution. In this paper we mostly focus on a particular class of models associated with the CP-decomposition of tensors, and consider a specific "identity" target (see Section 2), but we expect the proposed methods to have a much wider applicability. Our specific contributions are as follows.

1. In Section 3 we introduce the power series expansion of the loss evolution in time $t$ and show that for a suitable class of targets its coefficients are polynomials that can be described by certain (hyper-)graphs (*diagrams*). We develop a "diagram calculus" showing how higher-order diagrams and expectations can be found by *merging* and *contracting* diagrams.

2. In Section 4 we propose to analyze large-size scaling limits of the model through *Pareto-optimal* terms of the polynomials representing the loss coefficients. We rigorously find complete *"Pareto polygons"* of such terms for two scenarios and a particular target.

3. In Section 5 we associate different *learning regimes* with different faces of the Pareto polygon. We identify several regimes with specific scaling conditions and a natural interpretation (e.g. NTK, mean field).

4. In Section 6 we propose a general method for summation of formal power series appearing in the large-size limit of the loss expansion in different learning regimes. This method is illustrated in four important cases described in the following items.

---

[1]Applied AI Institute, Moscow, Russia [2]Steklov Mathematical Institute of Russian Academy of Sciences, Moscow, Russia. Correspondence to: Eugene Golikov <e.golikov@applied-ai.ru>.

*Proceedings of the 43rd International Conference on Machine Learning*, Seoul, South Korea. PMLR 306, 2026. Copyright 2026 by the author(s).

5. In Section 7, we present explicit solutions for the loss evolution with zero target, which we call *free*, in under- and over-parameterized settings. This regime is also relevant at the beginning of training for nonzero targets that are small relative to the model initialization.

6. In Section 8, we present an explicit solution for the model associated with one of the vertices of the Pareto polygon and demonstrate that in the *asymmetric scenario* it corresponds to a *Neural Tangent Kernel* regime (Jacot et al., 2018). We also explain why there is no NTK limit for *symmetric models*. The existence of this NTK limit does not follow from NTK convergence results known in the literature.

7. In Section 9, we present a *complete* solution in a large-size limit for the symmetric matrix model defined in Section 2. Our solution is valid for any double limit of our two size parameters $p$ and $H$.

8. In Section 10, we consider a more complex scenario of a 4-order identity tensor decomposition. While the analytic solution we found is valid only for gradient *ascent*, it allows us to identify two qualitatively different regimes of *unlearning*, depending on weight initialization magnitude.

We stress that, while some conceptually related expansions have been proposed earlier for the analysis of learning by GF (notably in the context of NTK hierarchy (Huang & Yau, 2020) and correlation functions (Dyer & Gur-Ari, 2020)), we are not aware of any previous works where such expansions have been used to systematically obtain specific results for gradient-based learning of particular models. In contrast, our focus is to demonstrate the utility of the properly organized time series/diagram expansion as a conceptual and computational tool for obtaining concrete information on the structure of learning phases and the loss evolution.

**Conflict of Interest Disclosure.** We are not aware of any conflicts of interest associated with this work.

## 2. The setting: GF-learned CP-decomposition

We assume that we fit a large target $F$ by a large model $f$. The size of the target is characterized by a growing parameter $p$, while the size of the model is characterized by a growing parameter $H$. Throughout the paper, we focus on the particular problem in which the target $F$ is an order-$\nu$ tensor ($\nu = 2, 3, \ldots$), and the model $f$ is a respective rank-$H$ CP (Canonical Polyadic) decomposition of the tensor:

$$f, F \in (\mathbb{R}^p)^{\otimes \nu}; f = (f_{i_1,\ldots,i_\nu}); F = (F_{i_1,\ldots,i_\nu}), \ i_m \in \overline{1, p},$$

$$f_{i_1,\ldots,i_\nu} = \sum_{k=1}^H \prod_{m=1}^\nu u_{k,i_m}^{(m)}. \tag{1}$$

The values $\mathbf{u} = (u_{k,i}^{(m)}) \in \mathbb{R}^{H \times p \times \nu}$ are the learnable parameters. We consider two scenarios: *symmetric* and *asymmetric*, for brevity denoted SYM and ASYM. In SYM the tensor $F$ is symmetric, as well as the CP decomposition:

$$u_{k,i}^{(1)} = \ldots = u_{k,i}^{(\nu)} =: u_{k,i}. \tag{2}$$

In ASYM no symmetry conditions are imposed. We consider the standard quadratic loss:

$$L(\mathbf{u}) = \frac{1}{2} \sum_{i_1,\ldots,i_\nu=1}^p (f_{i_1,\ldots,i_\nu} - F_{i_1,\ldots,i_\nu})^2. \tag{3}$$

The model parameters are learned by standard gradient flow:

$$\frac{d\mathbf{u}}{dt} = -\frac{1}{T} \partial_\mathbf{u} L(\mathbf{u}), \tag{4}$$

where $1/T$ is the learning rate. As $p, H \to \infty$, it is occasionally convenient to suitably rescale $T$ to ensure that $L(t)$ has a finite and nonvanishing limiting time scale.

To obtain detailed information on the loss evolution, our framework requires the target $F$ to scale in a specific way with growing $p$. In this paper we mostly focus on the simple scenario in which $F$ is an identity tensor:

$$F_{i_1,\ldots,i_\nu} = \delta_{i_1=\ldots=i_\nu}. \tag{5}$$

At $\nu = 2$ this corresponds to learning the identity matrix, while at $\nu = 3$ this target is closely related to learning modular addition (see Appendix C). More general naturally scalable targets that can be covered by our framework include linear combinations of tensors defined by general delta functions, as well as various random teacher models. The architectures of the learned models can also be significantly generalized, including, for example, neural networks with polynomial activations. However, these extensions are beyond the scope of the present paper.

We assume the weights $u_{k,i_m}^{(m)}$ to have i.i.d. random normal initializations with variance $\sigma^2$. In the sequel we will be interested in classifying and analytically computing the expected loss trajectory $\mathbb{E}[L(t)]$ of the gradient flow in the large-size limit $p, H \to \infty$. We will see that the results depend in a complex way on the tensor order $\nu$, the presence of symmetry, and the mutual scaling among $p, H$ and $\sigma^2$.

## 3. Loss evolution expansion and diagrams

**The loss expansion.** We start with the power series representing the time evolution of the loss and its expectation:

$$L(t) \sim \sum_{s=0}^\infty \frac{d^s L}{dt^s}(0) \frac{t^s}{s!}, \quad \mathbb{E}[L(t)] \sim \sum_{s=0}^\infty \mathbb{E}\Big[\frac{d^s L}{dt^s}(0)\Big] \frac{t^s}{s!}. \tag{6}$$

Here and in the sequel, we denote by $\sim$ formal power expansions. For any finite $p, H$ and initialization, the loss $L$ is an analytic function of $t$ at least in a sufficiently small neighborhood of $t = 0$, but this will not generally be the case after large-size limits and averaging. Nevertheless, all expansions that we consider will be well-defined as formal power series in $t$, i.e. have well-defined finite coefficients.

Any quantity $G$ depending on the model parameters evolves under GF as

$$\frac{dG}{dt} = -\frac{1}{T} \sum_u \frac{\partial G}{\partial u} \frac{\partial L}{\partial u}, \tag{7}$$

where $\sum_u$ is the sum over all learnable parameters. In particular, any derivative $d^s L/dt^s$ in (6) is a polynomial in partial derivatives $\partial L/\partial u$ and hence in the weights $u_{k,i}^{(m)}$, since our loss (3) is polynomial in the weights. The averaging over the Gaussian initialization can then be performed using the Wick formula (see Section B).

An important observation that we will use in the sequel is that for a broad class of targets the expressions $T^s \mathbb{E}[d^s L/dt^s(0)]$ are polynomials in $H, p$ and $\sigma^2$. We will state a relevant theorem for one particular class including our main "identity" target (5), but it is clear that the theorem can be further generalized.

**Theorem 3.1** (B). *Suppose that the target tensor $F_{i_1,\ldots,i_\nu}$ can be written as a polynomial in $H, p$, indices $i_1, \ldots, i_\nu$ and Kronecker deltas $\delta_{i_a = i_b}$ for $a, b \in \overline{1, \nu}$. Then, for any $s$, $T^s \mathbb{E}[d^s L/dt^s(0)]$ is a polynomial in $H, p, \sigma^2$.*

The theorem covers our target (5) since it can be written as $\delta_{i_1 = i_2} \delta_{i_2 = i_3} \ldots \delta_{i_{\nu-1} = i_\nu}$. We will denote the polynomials expressing $T^s \mathbb{E}[d^s L/dt^s(0)]$ by $Y_s = Y_s(H, p, \sigma^2)$.

**Diagrammatic representation.** To keep track of the polynomials $Y_s$, it is convenient to represent relevant polynomials in the weights $u_{k,i}^{(m)}$ by (hyper-)graphs that we call *diagrams*. Note first that the loss (3) can be written as the sum of three terms representing the pure model, the model-target interaction, and the pure target:

$$L = \frac{1}{2} \sum_{i_1,\ldots,i_\nu=1}^{p} \sum_{k=1}^{H} \sum_{k'=1}^{H} \prod_{m=1}^{\nu} u_{k,i_m}^{(m)} u_{k',i_m}^{(m)} \tag{8}$$
$$- \sum_{i_1,\ldots,i_\nu=1}^{p} \sum_{k=1}^{H} \prod_{m=1}^{\nu} u_{k,i_m}^{(m)} F_{i_1,\ldots,i_\nu} + \frac{1}{2} \sum_{i_1,\ldots,i_\nu=1}^{p} F_{i_1,\ldots,i_\nu}^2.$$

We can formally describe the first term by a graph $D_{2\nu}$ that contains $\nu$ nodes corresponding to summation over $i_m$ and 2 nodes corresponding to summation over $k$ and $k'$ (see Fig. 1 a). We refer to the nodes of the first kind as *p-nodes*, and to the nodes of the second kind as *H-nodes*. The edges of the graph connect the $p$-nodes to the $H$-nodes and correspond to

the weights $u_{k,i}^{(m)}$. We occasionally refer to the edge indices $m \in \overline{1, \nu}$ as *colors* (so in the symmetric scenario all edges have the same color). The full weight polynomial representing the first term then results by forming the products of the weights corresponding to all the edges, and summing these products over all the node index assignments.

The second term representing the model-target interaction can be described in a similar way, but in the case of a general target $F$ it requires considering a hypergraph $R_\nu$ with a hyper-edge $\{i_1, \ldots, i_\nu\}$ representing the factor $F_{i_1,\ldots,i_\nu}$. In this paper we will mostly focus on the "identity" target (5), for which the diagrammatic representation simplifies: the respective loss term becomes $\sum_{i=1}^{p} \sum_{k=1}^{H} \prod_{m=1}^{\nu} u_{k,i}^{(m)}$ and can be described by a usual graph having one $p$-node, one $H$-node, and $\nu$ edges connecting them (Fig. 1 b, c).

The third, pure-target term in (8) (it can be written in terms of the tensor Frobenius norm as $\frac{1}{2} \|F\|_F^2$) does not involve model weights and is only relevant for the initial $s = 0$ term of the loss expansion (6); in the case of "identity" target (5) it equals $\frac{p}{2}$ and corresponds to a trivial single-node graph.

**Diagram merging.** Computing $d^s L/dt^s$ using expansions (7) and (8) can be described in terms of *merging* weight polynomials and associated diagrams. Suppose that $G$ is a weight polynomial associated with some diagram (graph) that we denote by the same letter $G$. As discussed before, we have a similar diagram representation for the loss (8):

$$L = \frac{1}{2} D_{2\nu} - R_\nu + \frac{1}{2} \|F\|_F^2. \tag{9}$$

We represent now the evolution law (7) as

$$\frac{dG}{dt} = -\frac{1}{T} G \star \left( \frac{1}{2} D_{2\nu} - R_\nu \right), \tag{10}$$

where $\star$ is the *binary merging operation* defined for pairs of diagrams and extended bilinearly to their linear combinations. Specifically, given diagrams $G$ and $G'$, their merger $G \star G'$ is the sum of diagrams (i.e., respective polynomials), obtained as follows. We choose an edge $g = (k, i)$ in $G$ and an edge $g' = (k', i')$ in $G'$ of the same color, remove $g$ from $G$ and $g'$ from $G'$, and join the resulting diagrams by identifying $k$ with $k'$ and $i$ with $i'$ (see Fig. 1 d-f). Clearly, this construction corresponds exactly to the computation $\sum_u \frac{\partial G}{\partial u} \frac{\partial G'}{\partial u}$. In particular, the full expansion of $L(t)$ can be written in terms of repeated mergers:

$$L(t) \sim \frac{1}{2} \|F\|_F^2 + \sum_{s=0}^{\infty} \left( \frac{1}{2} D_{2\nu} - R_\nu \right)^{\star(s+1)} (0) \frac{(-t)^s}{T^s s!}. \tag{11}$$

**Averaging and diagram contractions.** We sketch now how the polynomials $Y_s$ are obtained (see B for details).

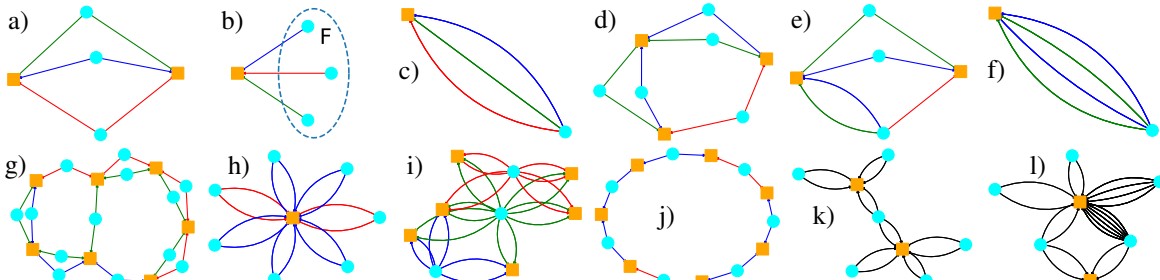

*Figure 1.* **Top row:** Diagrams in ASYM with $\nu = 3$ (three colors). Yellow squares: $H$-nodes; cyan circles: $p$-nodes. a) $D_6$; b) $R_3$ with a general target hyper-edge $F$; c) $R_3$ for the identity target (5); diagrams appearing d) in $D_6 \star D_6$ (up to recoloring); e) in $D_6 \star R_3$; f) in $R_3 \star R_3$. **Bottom row:** generic diagrams in different regimes. g) a diagram from $D_6^{\star s}$ (free evolution); h) a "flower" with one $H$-node (contracted, underparameterized); i) an optimally contracted free overparameterized; j) circular ($\nu = 2$); k) circular optimally contracted to a tree (SYM, $\nu = 2$); l) generic optimally contracted in SYM, $\nu = 4$.

Since the model is initialized as $u_{k,i}^{(m)} \sim \mathcal{N}(0, \sigma^2)$, for any diagram/polynomial $G$ its mean $\mathbb{E}[G]$ at initial time $t = 0$ can be computed by the Wick theorem. This means that $\mathbb{E}[G]$ is the sum of terms indexed by all partitions of the edges/weights $u_{k,i}^{(m)}$ into pairs $(u_{k,i}^{(m)}, u_{k',i'}^{(m')})$. For each such pair, $\mathbb{E}[u_{k,i}^{(m)} u_{k',i'}^{(m')}] = \sigma^2 \delta_{i=i'} \delta_{k=k'} \delta_{m=m'}$. The condition $m = m'$ means that we pair edges of the same color, while the conditions $i = i', k = k'$ mean that we *contract* (identify) the respective nodes. As a result, the contribution of a partition to $\mathbb{E}[G]$ equals a deterministic combinatorial expression associated with the contracted diagram. In the case of the identity target this expression is simply the monomial $p^q H^n \sigma^{2l}$, where $2l$ is the number of edges, $q$ is the number of $p$-nodes and $n$ is the number of $H$-nodes after the contraction. The full polynomial $Y_s$ is obtained by adding all monomials arising from various diagrams in $(\frac{1}{2} D_{2\nu} - R_\nu)^{\star(s+1)}$ and from various pairings of their edges.

## 4. The large-size scaling limit

**Power-law scaling.** We are interested in the properties of the loss evolution at large $p$ and $H$. These properties will depend on the scaling among $p$, $H$, and initialization noise $\sigma^2$. One natural assumption is that these variables are connected by a *power-law scaling*

$$p \asymp a^{\alpha_p}, H \asymp a^{\alpha_H}, \sigma \asymp a^{\alpha_\sigma}, \quad a \to +\infty, \qquad (12)$$

with specific exponents $\boldsymbol{\alpha} = (\alpha_p, \alpha_H, \alpha_\sigma)$. One can also consider cruder or, conversely, more refined scaling conditions that include the coefficients in addition to the powers.

**Leading and Pareto-optimal terms.** Consider the polynomials $Y_s = \sum_{q,n,l} c_{q,n,l;s} p^q H^n \sigma^{2l}$ representing the loss expansion coefficients by Theorem 3.1. Given a power-law scaling (12) with some exponents $\boldsymbol{\alpha}$, the generic[1] respective

---

[1] Excluding degenerate cases with several leading monomials cancelling each other

scaling of $Y_s$ is

$$Y_s \asymp a^{\alpha_{Y_s}}, \quad \alpha_{Y_s} = \max_{(q,n,l)} (\alpha_p q + \alpha_H n + 2\alpha_\sigma l), \quad (13)$$

where $\max$ is taken over all monomials present in $Y_s$. We call the respective monomials *leading*. In general, different triplets $\boldsymbol{\alpha}$ correspond to different leading terms.

To analyze this correspondence, it is convenient to consider what we call *Pareto-optimal* terms. We define them as the monomials in $Y_s$ having the largest powers $(q, n)$ in the Pareto sense among the monomials with the same $l$. For example, if $Y_s = pH\sigma^2 + pH^2\sigma^4 + pH^3\sigma^4 + p^2H\sigma^4$, then all terms are Pareto-optimal except $pH^2\sigma^4$, as it is dominated by $pH^3\sigma^4$.

In our context $\alpha_p$ and $\alpha_H$ (but not necessarily $\alpha_\sigma$) are positive, since $H$ and $p$ grow. Hence, any term leading for some triple $\boldsymbol{\alpha}$ will be among the Pareto-optimal terms. On the other hand, Pareto-optimal terms form a relatively small subset of all terms and can often be explicitly described thanks to originating from *minimally-contracted* diagrams. Namely, in the diagram picture $2l$ is the number of edges, while $q$ and $n$ are the numbers of $p$-nodes and $H$-nodes after the node contraction induced by the edge pairing. Pareto-optimal terms correspond to the diagrams and pairings that minimally constrain summations over the node indices.

The power $2l$ corresponds to the number of edges in a diagram and can be computed by binomially expanding the expression $(\frac{1}{2} D_{2\nu} - R_\nu)^{\star(s+1)}$. Suppose that a diagram $G$ results from merging $s_D$ diagrams $D_{2\nu}$ and $s_R$ diagrams $R_\nu$ in some order. Then $G$ has $2l = 2\nu s_D + \nu s_R - 2s$ edges, since merging any diagram with $D_{2\nu}$ adds $2(\nu - 1)$ edges, while merging with $R_\nu$ adds $\nu - 2$ edges. Since $s_D + s_R = s + 1$, we can equivalently write $2l = \nu(s_D + 1) + (\nu - 2)s$.

Finding the full set of Pareto-optimal triples $(q, n, l)$ is more subtle. It requires a careful analysis of different diagrams and generally depends on the model scenario and the type of target. We give a complete parametric description of Pareto-

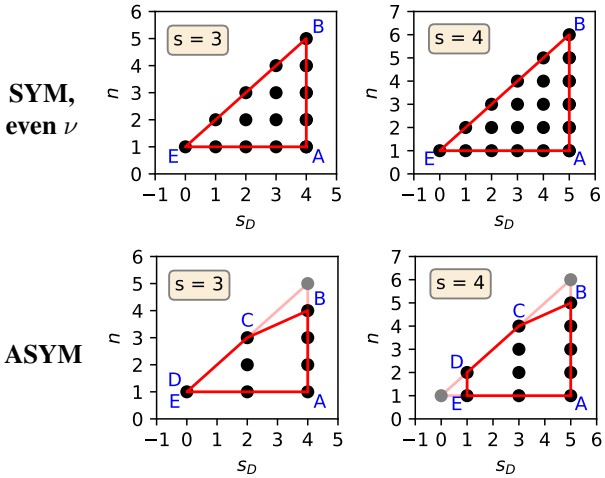

*Figure 2.* Pareto optimal terms (black, see Theorem 4.1) and the corresponding Pareto polygons (red) for the target (5). The extremal points present in SYM but missing from ASYM are colored gray.

optimal terms for the identity target in all scenarios except SYM with odd $\nu$ (in which case the description is more complicated and will be analyzed elsewhere), see Fig. 2.

**Theorem 4.1** (D). *Consider GF with the identity target* (5). *Up to nonzero numerical coefficients, the Pareto-optimal terms in $Y_s$ are*

$$p^{Q(n,s_D)} H^n \sigma^{\nu(s_D+1)+(\nu-2)s}, \qquad (14)$$

*where $0 \leq s_D \leq s+1, 1 \leq n \leq s_D+1$ and*

$$Q(n,s_D) = \begin{cases} 1+(\nu-1)s_D - \frac{\nu}{2}(n-1), & \text{SYM, even } \nu, \\ 1+(\nu-1)(s_D+1-n), & \text{ASYM.} \end{cases} \qquad (15)$$

*In SYM with even $\nu$ all of these terms occur. ASYM has exceptional terms that do not occur: a) terms with odd $s_R = s+1-s_D$; b) the term $(n, s_D) = (s+2, s+1)$.*

## 5. Pareto polygons and learning regimes

Theorem 4.1 implies the following remarkable fact: for each $s$ the Pareto power triplets $(q, n, 2l)$ form a planar polygon in $\mathbb{R}^3$, with the same normal vector

$$(\alpha_p, \alpha_H, \alpha_\sigma) = \begin{cases} \left(1, \frac{\nu}{2}, \frac{1-\nu}{\nu}\right), & \text{SYM, even } \nu, \\ \left(1, \nu-1, \frac{1-\nu}{\nu}\right), & \text{ASYM.} \end{cases}$$

This shows that there is a special joint scaling $p^{\nu-1} \asymp H^{2(\nu-1)/\nu} \asymp \sigma^{-\nu}$ (SYM, even $\nu$) or $p^{\nu-1} \asymp H \asymp \sigma^{-\nu}$ (ASYM) at which all the Pareto monomials in $Y_s$ will be leading. Moreover, we can similarly identify scaling conditions associated with various faces (i.e., sides or vertices) of the Pareto polygons. In general, a $d$-dimensional face corresponds to a $(3-d)$-dimensional set of scaling vectors $\boldsymbol{\alpha}$, or a $(2-d)$-dimensional set of normalized scaling vectors.

We identify the different faces and respective scaling conditions with *extreme learning regimes* of the model. Most of them have a natural interpretation summarized in Table 1 for the ASYM scenario, and in Table 2 for both scenarios. For example, some of the sides correspond to limit regimes that previously appeared in the literature: namely, the constant NTK and the mean-field regimes; see Appendix E for a detailed description and derivations of the scaling conditions for each extremal simplex, and Appendix A.1 for a detailed literature overview on limit regimes.

For each of the extreme regimes, there is a natural scaling of the inverse learning rate $T$ that leads to a nondegenerate evolution of the average loss as a function of $t$. It is obtained by balancing the base of power $s$ in the leading Pareto-term provided by Theorem 4.1 with $T$ (see Appendix E).

Here we describe high-level properties of the Pareto polygon. As Theorem 4.1 shows, it is a triangle in the symmetric scenario, while in the asymmetric one, one or two summits have to be removed, which turns the polygon either into a quadrangle or into a pentagon, see Figure 2.

As one moves from left to right, $s_D$ increases, hence $s_R = s+1-s_D$ decreases. The rightmost edge, A-B, corresponds to $s_R = 0$, which means that the target is never used for computing loss derivatives, hence the target is never learned. This edge becomes dominant when initialization is large: in this case, the identity target is essentially zero compared to the initial model, and the model attempts to learn zero throughout most of its training process. We call this regime *free evolution* and consider it in detail in Section 7.

Moving left gives $s_R > 0$, and we expect learning to occur. When only points with $s_R$ bounded from above by a constant independent of $s$ become dominant, we say that *weak learning* occurs: loss derivatives do depend on the target but only weakly (polynomially with bounded degree). A notable qualitative difference between the symmetric and the asymmetric scenarios is that weak learning exists in the latter, but does not exist in the former. Indeed, Point C and Edge B-C are extremal simplices with $s_R = 2$ and $s_R \in \{0, 2\}$, respectively. On the other hand, the only extremal simplices that contain $s_R > 0$ in the symmetric scenario are Edges B-E and E-A, Point E, and the whole triangle A-B-E. All three contain Point E with $s_R = s+1$. In Sec. 8, we explain why linearized training does not exist in the symmetric scenario but exists in the asymmetric one.

Following previous works (e.g. Chizat et al. (2019)), we call the training process *rich* if it is not lazy: i.e. when higher-order target correlations occur when computing loss derivatives. Important examples of rich regimes are Edges B-E (symmetric) and C-D (asymmetric), which require $\sigma^2 \asymp 1/H$ and $\sigma^2 \asymp 1/H^{2/\nu}$, respectively. For $\nu = 2$, these scalings coincide and yield a mean-field limit discovered

*Table 1.* Extremal simplices along with their interpretations; asymmetric scenario. See Table 2 for a similar table covering both scenarios.

| Simplex | Scaling condition | Natural $T$ | Parameterization | Learning | Interpretation |
|---------|-------------------|-------------|------------------|----------|----------------|
| A – B | $H \asymp p^{\nu-1}, p^{\nu-1}\sigma^\nu \to \infty$ | $H\sigma^{2\nu-2}$ | Balanced | No | Free evolution |
| B – C | $p^{\nu-1}H\sigma^{2\nu} \asymp 1, H\sigma^\nu \to \infty$ | $H\sigma^{2\nu-2}$ | Over- | Lazy | NTK |
| C | $p^{\nu-1}H\sigma^{2\nu} \to 0, H\sigma^\nu \to \infty$ | $H\sigma^{2\nu-2}$ | Over- | Lazy | NTK, $f(0) \equiv 0$ |
| C – D | $H\sigma^\nu \asymp 1, H/p^{\nu-1} \to \infty$ | $\sigma^{\nu-2}$ | Over- | Rich | Mean-field |
| D – E | $H \asymp p^{\nu-1}, H\sigma^\nu \to 0$ | $\sigma^{\nu-2}$ | Balanced | Rich | — |
| E – A | $p^{\nu-1}\sigma^\nu \asymp 1, H/p^{\nu-1} \to 0$ | $\sigma^{\nu-2}$ | Under- | Rich | — |

independently in a number of works: see Appendix A.1. Moving bottom-up in the Pareto polygon corresponds to increasing $n$. By Theorem 4.1, at larger $n$ increasing $H$ gets more important than increasing $p$. For this reason, top edges, B-E (resp. B-C and C-D), correspond to overparameterized regimes, while the bottom edge, E-A, is underparameterized. In turn, side edges, A-B and E (resp. D-E), give balanced regimes. We note that the balancedness condition depends on $\nu$: $H^2 \asymp p^\nu$ (resp. $H \asymp p^{\nu-1}$), while $H = p$ is always necessary and sufficient to fit an identity tensor for any $\nu$.

## 6. Explicit solutions of formal loss expansions

In previous sections we showed that the polynomials $Y_s$ have specific leading terms as $p, H \to \infty$, depending on the mutual scaling among $p$, $H$, and $\sigma$. By suitably rescaling $T$, we can ensure that each coefficient $\mathbb{E}[d^s L/dt^s(0)] = T^s Y_s$ of the formal loss expansion (6) has a finite limit. We can then consider the *formal coefficient-wise limiting expansion* of $\mathbb{E}[L(t)]$, and try summing this series.

This procedure is not easily mathematically justified and may not be valid in general. Even if valid, it may require non-standard summation methods (e.g., Borel summation). Nevertheless, we show below that it does work in a wide range of cases, and theoretical predictions agree very well with experimental results. Our general methodology is as follows.

1. We write a recurrence relation for the formal coefficients of the loss expansion. In these coefficients, we generally keep only the Pareto-optimal terms, and further drop terms that are subleading in the particular regime under consideration.

   Typically, such a recurrence relation cannot be written without considering additional auxiliary variables. Accordingly, this requires us to consider the diagram expansion of the loss as a special case of a more general generating function (g.f.) $h(\mathbf{x})$.

2. Using Theorem 6.1 below, we convert the recurrence relation into a (formal) partial differential equation (PDE) on $h$.

3. In many cases, the resulting PDE is a first-order PDE of

the form $\Phi(\mathbf{x})^T \nabla h(\mathbf{x}) = \phi(\mathbf{x})$ with some particular vector field $\Phi(\mathbf{x})$ and function $\phi(\mathbf{x})$. Such an equation can be solved by the method of characteristics:

$$h(\mathbf{x}(\tau_0)) = h(\mathbf{x}(\tau_1))e^{-\int_{\tau_0}^{\tau_1} \phi(\mathbf{x}(\tau))d\tau}, \quad (16)$$

where $\mathbf{x}(\tau)$ is any integral curve of the field $\Phi$, i.e. a solution of the ordinary differential equation (ODE) $\frac{d}{d\tau}\mathbf{x}(\tau) = \Phi(\mathbf{x}(\tau))$. We use Eq. (16) to transfer the values of $h$ from the points $\mathbf{x}(\tau_1)$ where we know $h$ to points $\mathbf{x}(\tau_0)$ of our interest.

The PDE for the generating function $h$ is derived by the following general theorem (see F for proof and comments).

**Theorem 6.1.** *Given a formal multivariate power series $h(\mathbf{x}) \sim \sum_{\mathbf{n} \in \mathbb{N}_0^d} C_\mathbf{n}\mathbf{x}^\mathbf{n}$, suppose that its coefficients satisfy the conditions*

$$\sum_{\mathbf{k} \in K} C_{\mathbf{n}+\mathbf{k}}P_\mathbf{k}(\mathbf{n} + \mathbf{k}) = 0, \quad \forall \mathbf{n} \in \mathbb{Z}^d, \quad (17)$$

*where $K$ is a finite subset of $\mathbb{Z}^d$, $P_\mathbf{k}$ are some $d$-variate polynomials, and $C_\mathbf{n} = 0$ if $\mathbf{n} \in \mathbb{Z}^d \setminus \mathbb{N}_0^d$. Then $h$ formally satisfies the differential equation*

$$\sum_{\mathbf{k} \in K} \mathbf{x}^{-\mathbf{k}}P_\mathbf{k}(\mathbf{x}\frac{\partial}{\partial\mathbf{x}})h = 0. \quad (18)$$

Sections 7, 9 and 10 below show successful applications of this procedure.

## 7. Free evolutions [G]

We refer to GF with zero target as *free evolution*. Free evolution is also relevant for nonzero targets when they are sufficiently small compared to the randomly initialized model: in this case free evolution naturally approximates the "deflation" of the model during the initial learning stage.

In our setting free evolutions form a one-parameter family characterized by the model size parameter $H$ (segment A-B in Fig. 2). This family includes two extreme regimes: *underparameterized* (A) and *overparameterized* (B), admitting explicit solutions and suggestive interpretations. The analytic results agree well with experiment (see Fig. 3).

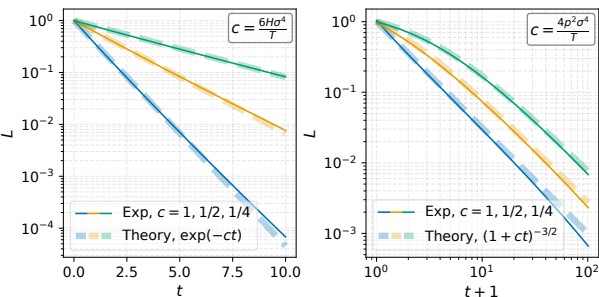

*Figure 3.* Free ASYM $\nu = 3$ (Sec. 7). **Left**: theory for the overparameterized case verified experimentally with $p = 32$, $H = p^3 = 32768$. **Right:** theory for the underparameterized case verified experimentally with $p = H = 128$.

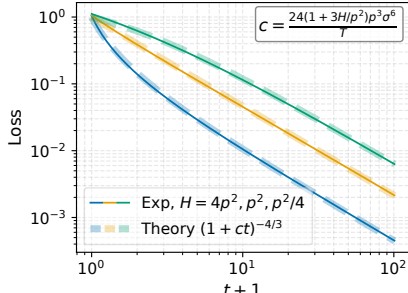

*Figure 4.* Free SYM $\nu = 4$ (Sec. 7). The experiments were conducted with $p = 64$ and with different values of $H$ proportional to $p^2$, while we fixed $\mathbb{E}[L(0)] = 1, T = 96p^3\sigma^6$.

The extreme underparameterized evolution (point A) corresponds to "flower" contracted diagrams $D_{2\nu}^{\star s}$ with a single contracted $H$-node (Fig. 1h). The expected loss is an asymptotic power law in both scenarios:

$$\frac{\mathbb{E}[L(t)]}{\mathbb{E}[L(0)]} \sim \begin{cases} \left(1 + \frac{2(\nu-1)p^{\nu-1}\sigma^{2(\nu-1)}t}{T}\right)^{-\frac{\nu}{\nu-1}}, & \text{ASYM}, \\ \left(1 + \frac{2\nu(\nu-1)p^{\nu-1}\sigma^{2(\nu-1)}t}{T}\right)^{-\frac{\nu}{\nu-1}}, & \substack{\text{SYM} \\ \text{even } \nu} \end{cases}.$$

The extreme overparameterized evolution (point B) is more complex. While for the symmetric even-$\nu$ model the loss is still an asymptotic power law:

$$\frac{\mathbb{E}[L(t)]}{\mathbb{E}[L(0)]} \sim \left(1 + \frac{2\nu(\nu-1)!!(\nu-1)p^{\nu/2-1}H\sigma^{2(\nu-1)}t}{T}\right)^{-\frac{\nu}{\nu-1}},$$

in ASYM it falls off exponentially (see Fig. 3):

$$\frac{\mathbb{E}[L(t)]}{\mathbb{E}[L(0)]} \sim e^{-2\nu\sigma^{2(\nu-1)}Ht/T},$$

showing a significant difference between the two scenarios.

Additionally, the method of Section 6 allows us to find explicit solutions for general free evolutions (i.e., the full segment AB) in even-$\nu$ SYM. The cases $\nu = 2$ and $\nu \geq 4$ are qualitatively different. For $\nu = 2$, the free evolution

can be derived in terms of the Narayana generating function as an extreme case of the general solution (19)-(20) of the triangle A-B-E given in Section 9. The relevant diagrams are single loops optimally contracted to trees. On the other hand, for even $\nu \geq 4$ the diagrams are multi-looped, but (in contrast to the case $\nu = 2$) all optimal contractions of merged diagrams are mergers of optimally contracted diagrams, which allows us to derive a simple recurrence yielding

$$\frac{\mathbb{E}[L(t)]}{\mathbb{E}[L(0)]} \sim \left(1 + \frac{2\nu(\nu-1)\left[1 + \frac{(\nu-1)!!H}{p^{\nu/2}}\right]p^{\nu-1}\sigma^{2(\nu-1)}t}{T}\right)^{-\frac{\nu}{\nu-1}}$$

(see Fig. 4).

## 8. The NTK regime [H]

With any differentiable parametric model $f_{\mathbf{x}}(\mathbf{u})$ one associates the *Neural Tangent Kernel* (NTK) $\Theta_{\mathbf{x},\mathbf{x}'}(\mathbf{u}) = \nabla^\top f_{\mathbf{x}}(\mathbf{u})\nabla f_{\mathbf{x}'}(\mathbf{u})$, where $\mathbf{x}$ and $\mathbf{x}'$ are model inputs, while $\mathbf{u}$ are its learnable parameters.

As shown in Jacot et al. (2018), for neural nets under a specific parameterization, as the network width approaches infinity, their NTK converges to a deterministic non-evolving limit, while the training process becomes equivalent to a kernel method.

The NTK limit exists for all neural networks whose training process can be described as a *Tensor Program* (TP) (Yang, 2020; Yang & Littwin, 2021). We emphasize that the model we consider *cannot* be described as a TP even at initialization for all $\nu \geq 3$. That is why the results we present below are novel and are not covered by any of the existing works.

As we consider a scenario where not only the hidden dimension $H$, but also the input dimension $p$ grows to infinity, it is not immediately obvious how to define the model (or kernel) convergence. Fortunately, due to symmetries, we are able to describe element-wise convergence of the model and the initial NTK. We prove the following in Appendix H:

**Proposition 8.1.** *Consider $f$ defined in Equation (1) in the ASYM case. Then for $T \sim \eta^{-1}H\sigma^{2\nu-2}$,*

1. *The initial NTK of $f$ converges a.s. to the identity tensor as $H \to \infty$: $T^{-1}\Theta_{i_1,\ldots,i_\nu;i_1',\ldots,i_\nu'}(0) \to \eta\nu\prod_{m=1}^\nu \delta_{i_m = i_m'}$.*

2. *Under the scalings corresponding to Point C and Point B of Figure 2, the $t$-expansion terms of $f_{i_1,\ldots,i_\nu}$ converge to those of $(1 - e^{-\eta t})\delta_{i_1 = \ldots = i_\nu}$ in probability.*

In Figure 2 for the SYM, even $\nu$ case, there is no extreme point that would correspond to Point C of the ASYM case. This suggests that no scaling leads to the NTK limit in this case. We prove the following in Appendix I:

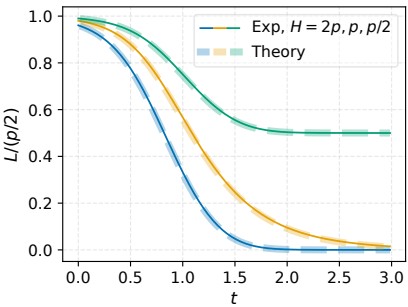

*Figure 5.* Experimental confirmation of theoretical predictions (eq. (19)) for the loss evolution in SYM $\nu = 2$. Experiments were performed for a fixed $p = 512$ and varying $H = 1024, 512, 256$.

**Proposition 8.2.** *For the SYM model with $\nu = 2$, $\Theta_{i,j;i',j'}(0) = \lim_{t\to\infty} \Theta_{i,j;i',j'}(t)$ under gradient flow $\forall i, j, i', j' \in [p]$ is equivalent to $f_{i,j}(0) = \lim_{t\to\infty} f_{i,j}(t)$ under gradient flow $\forall i, j \in [p]$.*

This means that if the NTK at the end of training is the same as at the beginning, the model does not learn anything.

## 9. SYM $\nu = 2$: general solution [J]

As a showcase of complex and non-lazy dynamics solvable by the general method of Section 6 we give the solution of the SYM $\nu = 2$ model with the identity target (5):

$$\mathbb{E}[L(t)] \sim \frac{p}{2} + p^2\sigma^2\Psi(-t/T, H/p, p\sigma^2), \qquad (19)$$

$$\Psi(x, y, z) = \frac{ze^{-8x}\partial_1 h(z(1 - e^{-4x}), y)}{2} \qquad (20)$$
$$- e^{-4x}h(z(1 - e^{-4x}), y),$$

$$h(z, y) = \frac{1 - z(y + 1) - \sqrt{1 - 2z(y + 1) + z^2(y - 1)^2}}{2z^2}.$$

The solution agrees very well with experiment: see Fig. 5. Despite the small-$t$ nature of the loss expansion, the solution correctly predicts the limiting loss $\lim_{t\to+\infty} \mathbb{E}[L(t)] = \max(\frac{p-H}{2}, 0)$ corresponding to the optimal approximation of the identity tensor by rank-$H$ decompositions (namely, using one of the $H$ terms to fill one diagonal component).

## 10. SYM $\nu = 4$: gradient ascent [K]

**Solution.** For SYM $\nu = 4$ with target (5), the method of Section 6 equipped with Borel summation yields

$$\mathbb{E}[L(t)] \sim \frac{p}{2} + p^2 g(-t/T, H/p^2), \qquad (21)$$

$$g(-\tau, y) = \frac{y\theta}{2}\left(p\sigma^2\phi(\tau)F(\sigma^2\psi(\tau))\right)^4$$
$$- \frac{y}{8}p\sigma^4\phi(\tau)^2 F'(\sigma^2\psi(\tau)),$$

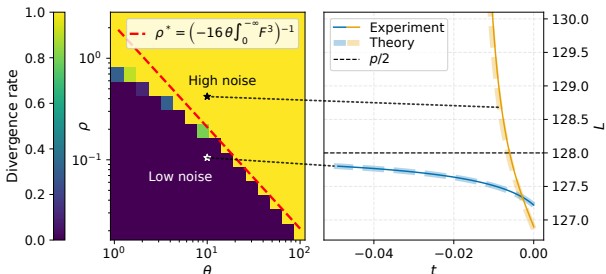

*Figure 6.* SYM $\nu = 4$. **Left:** experimental confirmation of the boundary (23). Gradient ascent divergence rates were obtained in 10 independent experiments for different values of $\rho = p^3\sigma^4$, $\theta = 1 + 3H/p^2$ with a fixed $p = 64$. **Right:** typical examples of experimental curves ($p = 256$) together with theoretical predictions (eq. (21)) for both *high-noise* and *low-noise* scenarios.

where $\theta = 1 + 3y$, $F(a) = \int_0^\infty e^{4au^2 - u}u\,du$, and $\phi, \psi$ are determined by the first integral

$$\phi(\tau)^{-2} = 1 + 16\theta p^3\sigma^4\int_0^{\sigma^2\psi(\tau)} F(u)^3 du, \qquad (22)$$

$$\psi'(\tau) = \phi(\tau), \ \psi(0) = 0, \ \phi(0) = 1.$$

However, note that this solution is only applicable when $t < 0$, i.e. for *gradient ascent*. Indeed, $F$ diverges unless $\sigma^2\psi(\tau) \le 0$, and $\psi(\tau) = \int_0^\tau \phi$ has the same sign as $\tau$. We leave discussion of gradient descent ($t > 0$) to another publication. For gradient ascent, our solution yields a nontrivial quantitative prediction of two regimes – with or without divergence – confirmed by experiments (see Figure 6).

**Gradient ascent: two regimes.** Since $F(a) > 0$ for $\sigma^2\psi \le 0$, we have $\int_0^{\sigma^2\psi} F(u)^3 du \le 0$ and the condition for the first integral (22) may fail to remain non-negative at sufficiently negative $\psi$. Therefore, if we require the solution to exist for all $\tau \le 0$, we get a condition on weight scaling $\sigma$:

$$p^3\sigma^4 < \rho^* = \left(-16\theta\int_0^{-\infty} F(u)^3 du\right)^{-1}. \qquad (23)$$

Thus, we identify two regimes:

1. *Low-noise*: if (23) holds, then $g(-\tau, y) \asymp -\frac{1}{\tau^2}$, $\tau \to -\infty$, and the loss (21) converges to $p/2$ as $t \to -\infty$.

2. *High-noise*: if (23) is violated, then $\phi = \psi'$ blows up at a finite $\tau_{\text{crit}} < 0$ and

$$g(-\tau, y) \asymp |\tau - \tau_{\text{crit}}|^{-4/3}, \ \tau \downarrow \tau_{\text{crit}}, \qquad (24)$$

which is in agreement with the divergence for negative times in the free evolution and matching the divergence exponent $-\nu/(\nu - 1) = -4/3$ (see Sec. 7).

## 11. Discussion

**Theoretical predictions.** While our paper primarily aims to develop a new general mathematical framework, our approach led to several specific and non-obvious new theoretical predictions confirmed by experiments, for example:

1. The NTK regime is present in the CP decomposition problem in the asymmetric setting for any order $\nu$, but missing from the symmetric setting with even orders.

2. In the free evolution of the overparameterized model, the loss falls off as a power law in the symmetric, even-$\nu$ setting, but exponentially in the asymmetric setting.

3. Gradient ascent (gradient flow for $t < 0$, "unlearning") of the loss may diverge or converge to a finite value; for target (5) we have found an explicit analytic condition.

**Generalizations.** Although our paper focuses on a specific model and target, the methodology we present here is by no means restricted to this setting. As explained in Section 3, we can consider any targets representable with (hyper)graphs: the modular addition target $F_{i_1,\ldots,i_\nu} = \delta_{i_\nu = \sum_{m=1}^{\nu-1} i_m \mod p}$ is an important example. We can as well consider any model representable as a graph; this includes multi-layer nets with polynomial activations.

Although our model's inputs were one-hot vectors, our method naturally covers finite Gaussian datasets. This allows us to study *generalization* by comparing performance on a given dataset with that on the distribution. Applying hypergraph targets, we can also study generalization on datasets of one-hot vectors. This paves the way for studying complex phenomena such as grokking in modular addition.

**Rigorous justification.** Our framework in its present form is not mathematically complete: there are important questions that we leave open for the moment. In particular, while we have shown and experimentally confirmed in several scenarios that the formal power series in the large-size limit can be successfully used to deduce the phase structure and derive explicit solutions of GF by a suitable summation method, these procedures may not be applicable in general. Rigorous conditions of applicability and related convergence questions are certainly an important topic of future research.

These questions closely parallel similar questions for Feynman diagram expansions in quantum field theory and statistical physics (Feynman, 1948): while conceptually and computationally very convenient and present in most textbooks, these expansions are notoriously hard to rigorously justify. Typically they are only considered term-wise or summed by heuristically selecting most relevant components (Mattuck, 1992). Rigorous constructions of interacting theories are normally performed by other methods such as cluster expansions; only after the theory is rigorously constructed it can sometimes be connected to a Feynman diagram expansion (Rivasseau, 2009). Some theories have never been rigorously constructed: for example, this question for the Yang–Mills theory is one of the six unsolved Millennium problems (Jaffe & Witten, 2000).

## Acknowledgments

The research was supported by the Russian Science Foundation grant No. 25-11-00355,
https://rscf.ru/project/25-11-00355/.

## Impact Statement

We believe that our paper introduces a novel and important theoretical perspective on learning dynamics in large machine learning problems.

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

# A. Literature survey

Our work is not based directly on any of the existing works we are aware of. Nevertheless, it draws connections with numerous topics, namely: (1) modular arithmetic as a supervised learning problem, (2) infinitely wide networks, (3) gradient descent dynamics on linear networks, (4) matrix factorization, (5) diagrammatic methods in Deep Learning, and (6) parameter expansions. We overview the first topic in Appendix C and the remaining topics below.

## A.1. Infinitely wide networks

**Neural Tangent Kernel.** A pioneering work of (Jacot et al., 2018) demonstrated that under a specific parameterization and in the limit of infinite width, the process of training a fully-connected neural network is equivalent to kernel gradient descent for a specific kernel coined *Neural Tangent Kernel*, or NTK. The main advantage of this equivalence is that kernel methods are much better understood compared to neural nets in general. The main disadvantage is that the required parameterization is non-standard; because of this, the equivalence does not hold for standardly parameterized neural nets. Since a kernel method could be seen as a linear model in a large (but fixed) feature space, the absence of a kernel method equivalence means that neural networks, in fact, do learn features during training. This fact is believed to explain why neural nets perform better in some tasks compared to conventional kernel methods. Two of the limit regimes we identify in our setting, namely, that corresponding to Point C and Edge B-C in the asymmetric scenario (see Figure 2, right), correspond to the constant NTK limit of (Jacot et al., 2018). Indeed, Point C corresponds only to diagrams in the loss expansion resulting from merging exactly two sub-diagrams $R_\nu$ together with any number of sub-diagrams $D_{2\nu}$. This implies that the loss at any time step $t$ depends only quadratically on the target. This is possible only when the learned weights depend linearly on the target. See Appendix H for further details.

**Mean-field limit.** A number of works demonstrate that the gradient flow dynamics on a two-layer MLP is equivalent to an evolution of a measure: (Chizat & Bach, 2018; Rotskoff & Vanden-Eijnden, 2018; Sirignano & Spiliopoulos, 2020). Under a specific parameter scaling, the moments of this measure stay finite as the number of neurons in the hidden layer grows. Thanks to this fact, a limit of infinite width comes naturally. Note that the required parameterization is different from that required for the NTK equivalence of (Jacot et al., 2018). While the scaling leading to the NTK regime corresponds to Edge B-C on the hyperparameter polygon, that leading to the mean-field regime corresponds to Edge C-D in the asymmetric case, and Edge B-E in the symmetric one. See Appendix E for further details.

The name "mean-field" comes from the fact that the measure interacts with itself not directly, but through interacting with a scalar variable (namely, the loss) which it forms itself. This is in line with a mean-field approximation in particle physics, where particles are assumed not to interact with each other directly, but to interact with a field formed by these particles collectively (e.g. charged particles collectively form a magnetic field and interact with it).

The main advantage of the mean-field limit is that it allows for feature learning (that is, the associated NTK changes throughout training).

The main disadvantage of the mean-field limit is that a measure evolution is much trickier to deal with compared to a kernel machine.

Another big disadvantage of the mean-field limit is that the constructions of the above-mentioned works do not allow for a straightforward generalization to MLPs beyond two layers. Nevertheless, several works propose various formulations of the gradient flow in terms of a measure evolution for deep nets: see (Araújo et al., 2019; Nguyen, 2019; Sirignano & Spiliopoulos, 2022; Nguyen & Pham, 2023). We note also a $\mu P$ limit (Yang & Hu, 2021) which is well-defined for deep nets and coincides with the mean-field one for two-layered nets, but does not allow for a physical interpretation in terms of measures as above.

**Classification of infinite-width regimes.** Note that the network parameterizations required for the NTK limit and the $\mu P$ limit are different and lead to qualitatively different behavior: in the former case, the model becomes equivalent to a kernel method, hence does not learn features, while in the latter case it provably does (Yang & Hu, 2021). Are there any other meaningful infinite-width regimes? One of the contributions of the present work is a classification of infinite $p$ and $H$ regimes for identity tensor decomposition: see Theorem 4.1 visualized with a hyperparameter polygon of Figure 2. A few earlier works (Golikov, 2020a;b) classified all possible infinite-width limits (in our terms, only $H$ is infinite) for a two-layer net. It follows from their classification that all other limits are in some sense degenerate. Later, (Yang & Hu, 2021) provided an alternative classification for multi-layer nets.

### A.2. Gradient descent dynamics on linear networks

In the matrix case, one could think of the problem we study as training a linear network $f_{U,V}(x) = V^\top U x$ to fit an identity map. Linear networks, i.e. MLPs with identity activation functions, attracted a lot of attention during recent years. The reason is that even though they cannot express nonlinear maps, their training dynamics is nonlinear and might exhibit features which are typical for nonlinear nets, but could not be observed for a one-layer linear model.

In contrast to generic non-linear nets, the training dynamics of linear ones could be solved in several important cases which we review below. One of the main contributions of the present work is explicit solutions for our tensor decomposition model in various limit regimes, see Section 6 together with subsequent sections in the main. We underline that our model is a linear net with one hidden layer for $\nu = 2$, but it is not a linear net for $\nu \geq 3$.

The study of gradient descent dynamics for multi-layer linear nets dates back to a seminal work of (Fukumizu, 1998) who presented an exact analytic solution for gradient flow under a crucial assumption of *balanced initialization*. This assumption is quite specific and not satisfied by a random Gaussian initialization. Their solution was later revised by (Braun et al., 2022) and generalized to $\lambda$-balanced initializations by (Domine et al., 2025).

Whereas the balancedness assumption of (Fukumizu, 1998) does not depend on data, an alternative solution proposed by (Saxe et al., 2013) assumed the initial linear map to be aligned with the eigenvectors of the feature correlation matrix of the training dataset. If this is the case, the above alignment property holds throughout the whole training process, and the whole parameter evolution separates into independent scalar evolutions.

While the above assumption is neither automatically satisfied for standard initialization strategies, it is trivially satisfied for zero initialization. Though zero is a saddle point from which the training process cannot start, one can hope that initializing near the origin also approximately keeps the alignment property. (Saxe et al., 2013) empirically observed this claim, while (Braun et al., 2022) proved it for balanced initializations.

We underline that the explicit solutions we present in Section 6 hold for a generic Gaussian initialization without any additional constraints.

**Saddle-to-saddle regime.** An interesting feature of aligned near-zero initialization is that in this case, the gradient flow empirically exhibits a peculiar dynamics. It starts near a saddle point (the origin), stays there for a while, then escapes it until it gets stuck near another saddle point, where the process repeats. The training process ends when the gradient flow arrives at a minimum instead of a saddle. Each saddle corresponds to learning a finite number of strongest principal components of the data. Hence, the data are learned sequentially. This regime is coined *saddle-to-saddle* and is studied in a number of works, including (Li et al., 2021; Jacot et al., 2021).

Since we consider only identity targets, all our principal components have equal strength, and are learned at the same time. Therefore the only saddle we encounter during the optimization process is the one in the origin. However, we claim that our analysis generalizes easily to a number of other targets, including $F^0_{i_1,\ldots,i_\nu} = \delta_{i_1=\ldots=i_\nu=1}$, which corresponds to a tensor with a single "one" on a diagonal with all the remaining entries being zero, and a mix of the latter and the identity target: $F = F^I + \alpha F^0$ for some $\alpha > 0$ possibly depending on $p$ and $H$. In the latter case, the target has two distinct eigenvalues: 1 and $1 + \alpha$. Here, we expect to observe a non-trivial saddle-to-saddle dynamics, for which the strong component is learned first (it corresponds to the first non-trivial saddle in the weight evolution), while the rest are learned afterwards.

### A.3. Applications of diagrams to Deep Learning theory

Diagrams turn out to be a handy tool we use in our analysis to get leading terms of loss time derivatives for large $H$ and $p$; see examples in Figure 1 and the discussion of our approach in Section 4. Diagrammatic approaches akin to Feynman diagrams in particle physics turned out to be a natural tool for studying small deviation corrections for limits of certain quantities. In Deep Learning, the most well-studied limit is the infinite width limit in the NTK regime (see the discussion above).

Before proceeding with a discussion of works that compute finite-width corrections for limit NTKs using diagrammatic approaches, we underline that in contrast to expanding quantities around infinite width, we expand them around zero training time. Surprisingly, our expansions match very well with numerical experiments even for very large training time: see Figures 3, 5 and 12.

**Finite-width corrections to NTK.** The work of (Dyer & Gur-Ari, 2020) considers so-called *correlation functions*, which are expectations of products of derivative tensors of a given scalar function $f$. NNGP, NTK, as well as higher-order kernels and kernel time-derivatives could be expressed as correlation functions. The main result is a simple upper-bound for an exponent of a correlation function as a function of width $n$.

This bound is computed by counting the numbers of even-sized and odd-sized components in the corresponding *cluster graph*. Vertices of this graph correspond to derivative tensors, while edges mark that the two tensors are tied with derivatives. While the definition of correlation functions is agnostic to the structure of $f$, the exponent upper-bound crucially relies on the structure of $f$ being an NTK-parameterized fully-connected network.

The bound is proven in some scenarios (e.g. linear nets), but empirically holds for all scenarios considered (including Tanh and ReLU nets). Feynman diagrams *are not* cluster graphs; they are used merely as a tool for proving the bound for linear nets. That is, for correlation functions not involving any derivatives (that is, $\mathbb{E}[f(x_1), \ldots, f(x_m)]$), the corresponding Feynman diagrams are graphs with vertices corresponding to inputs $x_1, \ldots, x_m$, and edges corresponding to paired weights of the same layer. The pairing is given by Wick's formula. That is, Feynman diagrams encode pairings involved in Wick's formula. Each vertex has $L$ incident edges, where $L$ is the number of layers. When the correlation function involves derivatives, the two $f$'s contracted by the derivative act as a forced edge in the same type of diagrams.

In (Dyer & Gur-Ari, 2020), Feynman diagrams encode pairings in Wick's formula, while vertices encode different model inputs. In our work, these pairings are encoded by diagram contractions, while vertices encode common summation indices for adjacent weight matrices.

Higher-order derivatives we use in the loss expansion could be directly mapped to cluster graphs. However,

1. Initial weight variance $\sigma^2$ is a free parameter, dependence on which we study.

2. There are two non-equivalent notions of "width", which are $p$ and $H$.

3. We are interested not only in the leading term exponents, but also in exact constants in the leading terms, in their dependence on $s$ specifically. This is because we are looking for exact loss behavior for large $t$.

This is why the results of (Dyer & Gur-Ari, 2020) do not suffice in our case.

A follow-up work of (Aitken & Gur-Ari, 2020) aims to prove the main exponent bound of (Dyer & Gur-Ari, 2020) for nets with polynomial activation functions. As before, the main theorem is formulated in terms of cluster graphs. However, Feynman diagrams are no longer used as a proof technique. Instead, they use "tree-like structures" (actually, forests), where vertices correspond to indices of weight matrices (that is, input and output weight matrices are mapped to a single vertex, while intermediary weight matrices are mapped to a pair of vertices), while edges mark the fact that adjacent weight matrices are tied with the corresponding summation index. Pairings defined by Wick's formula are represented also as edges (as a pairing essentially ties the indices). These tree-like structures are more similar to diagrams in our work, but are still not the same thing.

(Andreassen & Dyer, 2020) is another follow-up work. It aims to generalize the results of (Dyer & Gur-Ari, 2020) to networks with convolutions, global average poolings, and skip connections. Their main conjecture is very similar to that of (Dyer & Gur-Ari, 2020) but slightly more general: they consider *mixed correlation functions* for which derivative tensors are taken for different functions sharing weights (this is needed to model convolutions). The proof technique also relies on Feynman diagrams as in (Dyer & Gur-Ari, 2020).

### A.4. Parameter expansions in Deep Learning

Recall that we expand the loss function as a function of time $t$: $L(t) = \sum_{k=0}^{\infty} \frac{d^k L(0)}{dt^k} \frac{t^k}{k!}$, see Section 4. A similar expansion appeared earlier in (Dyer & Gur-Ari, 2020) for the NTK: $\Theta(t) = \sum_{k=0}^{\infty} \frac{d^k \Theta(0)}{dt^k} \frac{t^k}{k!}$. The subsequent derivatives in that work were computed with a recursive formula. A parallel work of (Huang & Yau, 2020) expresses this recurrence as an infinite system of ODEs, where the time derivative of $\Theta$ is expressed using a higher-order kernel, whose derivative in turn is expressed with an even higher-order one.

## B. Loss expansion and diagram calculus

**Polynomials and diagrams.**   We give now more details on the polynomials and associated diagrams introduced in Section 3. In the context of our CP decomposition problem (1), it is natural to generally define a diagram as a hypergraph $G = (V, E)$. Here $V$ is the set of nodes, and it consists of two subsets, $V = V_p \cup V_H$, representing $p$-nodes and $H$-nodes. The $p$-nodes describe the "external" degrees of freedom, indexing the target, while the $H$-nodes describe the internal degrees of freedom in the model.

The set $E = E_u \cup E_F$ consists of the set $E_u$ of *model weight edges* and the set $E_F$ of *target hyper-edges*. A weight edge $u_{k,i}^{(m)} \in E_u$ connects an $H$-node $k \in V_H$ to a $p$-node $i \in V_p$. The index $m$ (*color*) is a property of the edge and can take values $1, \ldots, \nu$. In general, there may be many edges $u_{k,i}^{(m)}$, even with the same color $m$, between two given nodes $k$ and $i$ (i.e., $G$ is a *multi*-hyper-graph).

An element of the set $E_F$ is a hyper-edge $F_{i_1,\ldots,i_\nu}$ connecting $\nu$ $p$-nodes $i_1, \ldots, i_\nu \in V_p$. Again, there may be more than one hyper-edge connecting the same $p$-nodes.

To each diagram $G$ we associate a polynomial in the model weights which we call *diagram polynomial* and, by slightly abusing notation, denote by the same letter $G$. For clarity, let us enumerate all vertices and (hyper)-edges in $G$:

$$V_p = \{i_1, \ldots, i_Q\}, \tag{25}$$

$$V_H = \{k_1, \ldots, k_N\}, \tag{26}$$

$$E_u = \{u_{k^{(r)},i^{(r)}}^{(m^{(r)})}, r = 1, \ldots, |E_u|\}, \quad k^{(r)} \in V_H, i^{(r)} \in V_p, \tag{27}$$

$$E_F = \{F_{i^{(r,1)},\ldots,i^{(r,\nu)}}, r = 1, \ldots, |E_F|\}, \quad i^{(r,s)} \in V_p. \tag{28}$$

Then the polynomial $G$ is defined by

$$G = \sum_{i_1=1}^{p} \cdots \sum_{i_Q=1}^{p} \sum_{k_1=1}^{H} \cdots \sum_{k_N=1}^{H} \left( \prod_{r=1}^{|E_u|} u_{k^{(r)},i^{(r)}}^{(m^{(r)})} \right) \left( \prod_{r=1}^{|E_F|} F_{i^{(r,1)},\ldots,i^{(r,\nu)}} \right) \tag{29}$$

In other words, each node implies summation over the respective index, and for each assignment of node indices the respective monomial is formed by multiplying over all (hyper-)edges the model weights and target values associated with respective node indices. (Here we also slightly abuse notation and use $i_r, k_r$ to denote both a node in the hyper-graph and a respective summation index.)

There are two key reasons for these definitions of diagrams and associated polynomials:

1. All three terms of the loss expansion (8) are expressible in the above form:

$$L = \frac{1}{2} \underbrace{\sum_{i_1,\ldots,i_\nu=1}^{p} \sum_{k=1}^{H} \sum_{k'=1}^{H} \prod_{m=1}^{\nu} u_{k,i_m}^{(m)} u_{k',i_m}^{(m)}}_{\text{model self-interaction } D_{2\nu}} - \underbrace{\sum_{i_1,\ldots,i_\nu=1}^{p} \sum_{k=1}^{H} \prod_{m=1}^{\nu} u_{k,i_m}^{(m)} F_{i_1,\ldots,i_\nu}}_{\text{model-target interaction } R_\nu} + \frac{1}{2} \underbrace{\sum_{i_1,\ldots,i_\nu=1}^{p} F_{i_1,\ldots,i_\nu}^2}_{\text{target self-interaction}}$$

   (the last term does not involve the model weights and corresponds to an empty diagram).

2. Given two diagram polynomials $G, G'$, the expression $\sum_u \frac{\partial G}{\partial u} \frac{\partial G'}{\partial u}$, where summation is over all model weights, is also a diagram polynomial (see discussion of *merging* below).

These properties ensure, by Eqs. (6), (7), that all coefficients $d^s L/dt^s(0)$ in the loss expansion are expressible as linear combinations of diagram polynomials.

**The identity target.** The above description may be simplified for some specialized targets. In particular, in the case of the "identity" target (5), i.e., $F_{i_1,\ldots,i_\nu} = \delta_{i_1} = \ldots = i_\nu$, we can consider only usual graphs rather than hyper-graphs, since the hyper-edges are associated with the target tensor $F$. In this case the model-target interaction term

$$R_\nu = \sum_{i_1,\ldots,i_\nu=1}^{p} \sum_{k=1}^{H} \prod_{m=1}^{\nu} u_{k,i_m}^{(m)} F_{i_1,\ldots,i_\nu}$$

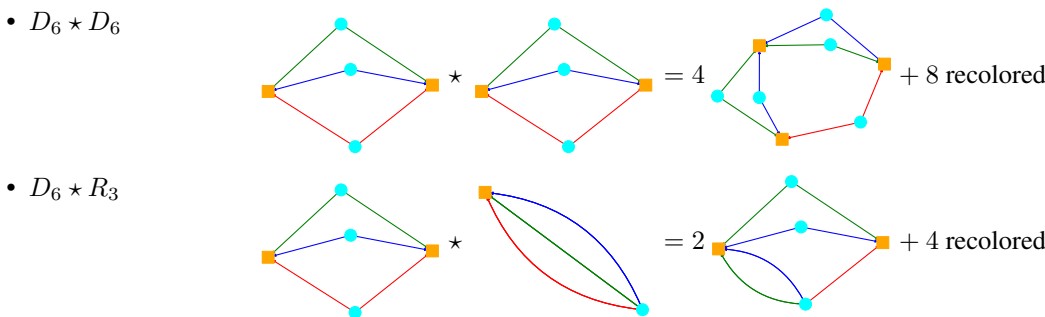

$$\tag{30}$$

simplifies to

$$R_\nu = \sum_{i=1}^{p} \sum_{k=1}^{H} \prod_{m=1}^{\nu} u_{k,i}^{(m)},$$

$$\tag{31}$$

not involving hyper-edges.

**Diagram merging.** To see why property 2 above holds, note that the model weights $u$ in question are the variables $u_{k,i}^{(m)}$. Computing a derivative $\partial G/\partial u_{k,i}^{(m)}$ for a polynomial requires locating each weight edge of the diagram with matching color $m^{(r)} = m$, setting $k^{(r)} = k, i^{(r)} = i$ (i.e. removing summation at the two nodes of the edge), and removing the edge variable $\partial u_{k,i}^{(m)}$ from the product. When we perform summation over $k, i$ in $\sum_u \frac{\partial G}{\partial u} \frac{\partial G'}{\partial u}$, we restore summation over the node indices, but now they are shared between the two diagrams. This is exactly the *merging operation* $\star$ introduced in Section 3. Let us summarize its action:

1. Given diagram polynomials $G_1$ and $G_2$, the merger $G_1 \star G_2$ is a sum of new diagram polynomials $G$.

2. Each of these $G$ is obtained as follows. We choose a color $m$ and locate an edge $g_1 = (k_1, i_1)$ of this color in $G_1$ and an edge $g_2 = (k_2, i_2)$ of this color in $G_2$. We remove these two edges and identify the node $k_1$ with $k_2$ and the node $i_1$ with $i_2$.

3. Merging is extended bilinearly to linear combinations of diagrams.

Examples:

- $D_6 \star D_6$

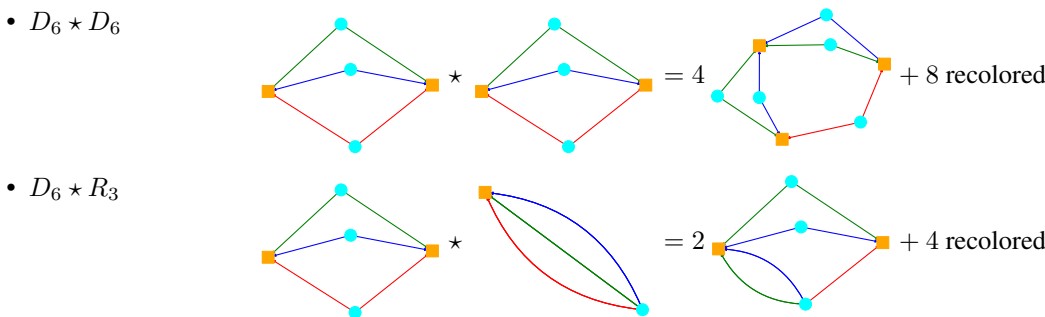

- $D_6 \star R_3$

**Wick's theorem and diagram contractions.** Recall that Wick's theorem states that the expectation $\mathbb{E}[X_1 \cdots X_{2l}]$ of a product of jointly Gaussian variables equals the sum of products $\prod_{r=1}^{l} \mathbb{E}[X_r X_{\phi(r)}]$ over all partitions $\{1, \ldots, 2l\} = \sqcup_{r=1}^{l} \{r, \phi(r)\}$ of the variables into disjoint pairs. In our context, the variables $X$ are the initial model weights $u_{k,i}^{(m)}$. By linearity, expectations of diagram polynomials (29) reduce to expectations of monomials

$$\mathbb{E}\left[ \prod_{r=1}^{|E_u|} u_{k^{(r)},i^{(r)}}^{(m^{(r)})} \right].$$

$$\tag{32}$$

Since the weights are represented by edges, this means that we need to consider all *edge pairings* in the diagram $G$ (these only involve the weight edges $E_u$, not the target hyper-edges $E_F$). Since the initial weights are independent, a pairing has a non-vanishing contribution only if the paired edges have a matching color $(m)$. Moreover, the expectation $\mathbb{E}[u_{k,i}^{(m)} u_{k',i'}^{(m')}] = \sigma^2 \delta_{m=m',i=i',k=k'}$ vanishes unless $k = k'$ and $i = i'$, meaning that we need to impose these additional constraints when summing over the indices.

This shows that $\mathbb{E}[G]$ can be computed as follows.

1. Consider all possible pairings of edges with matching colors.

2. For each such pairing, identify ("*contract*") the diagram nodes corresponding to paired edges. (Note that an identification can only involve same-type nodes, i.e. $p$-nodes are never identified with $H$-nodes).

3. If the resulting diagram is left with $n$ contracted $H$-nodes $\{k_1, \ldots, k_n\}$ and $q$ contracted $p$-nodes, and the number of edges is $2l$, then this pairing contributes to $\mathbb{E}[G]$ the term

$$A = \left[ \sum_{i_1=1}^{p} \cdots \sum_{i_q=1}^{p} \sum_{k_1=1}^{H} \cdots \sum_{k_n=1}^{H} \prod_{r=1}^{|E_F|} F_{i^{(r,1)},\ldots,i^{(r,\nu)}} \right] \sigma^{2l}. \tag{33}$$

We henceforth refer to the identification of some of the same-type nodes as *diagram contraction*.

In the special case of the identity target (5), assuming that the target hyper-edge has been incorporated into contracted nodes per Eq. (31) so that the diagram only has weight edges, expression (33) simplifies to $p^q H^n \sigma^{2l}$:

$$A = \left[ \sum_{i_1=1}^{p} \cdots \sum_{i_q=1}^{p} \sum_{k_1=1}^{H} \cdots \sum_{k_n=1}^{H} 1 \right] \sigma^{2l} = p^q H^n \sigma^{2l}. \tag{34}$$

**Example:** In the ASYM scenario, $\mathbb{E}[D_{2\nu}] = p^\nu H \sigma^{2\nu}$, since there is only one valid edge pairing corresponding to the $\nu$ pairs of same-color edges, and this pairing contracts the two $H$-nodes:

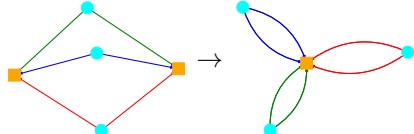

In SYM, all the edges pairings contribute non-vanishing terms, e.g., at $\nu = 2$ and $\nu = 3$

$$\mathbb{E}[D_4] = (p^2 H + pH^2 + pH)\sigma^4, \tag{35}$$

$$\mathbb{E}[D_6] = (p^3 + 5p^2 + 9p)H\sigma^6. \tag{36}$$

**Proof of Theorem 3.1.** Based on representation (33), we will prove a more general statement that for any diagram $G$ the expectation $\mathbb{E}[G]$ is a polynomial in $p, H, \sigma^2$ if the target tensor $F_{i_1,\ldots,i_\nu}$ can be written as a polynomial in $H, p$, indices $i_1, \ldots, i_\nu$ and Kronecker deltas $\delta_{i_a=i_b}$ for $a, b \in \overline{1, \nu}$. This conclusion will then also hold for $T^s \mathbb{E}[d^s L/dt^s(0)] = \mathbb{E}[(\frac{1}{2} D_{2\nu} - R_\nu)^{\star(s+1)}]$, which is the expectation of a linear combination of diagram polynomials.

By linearity, it is sufficient to consider the case when $F_{i_1,\ldots,i_\nu}$ is a monomial rather than polynomial, in $H, p$, indices $i_1, \ldots, i_\nu$ and Kronecker deltas $\delta_{i_a=i_b}$ for $a, b \in \overline{1, \nu}$. Clearly, the powers of $H$ and $p$ simply add the respective powers to the result, and the delta-factors $\delta_{i_a=i_b}$ constrain summation by effectively contracting the nodes. Then, the proof is completed by noting that expressions of the form

$$\sum_{i_1=1}^{p} \cdots \sum_{i_{q'}=1}^{p} \sum_{k_1=1}^{H} \cdots \sum_{k_{n'}=1}^{H} i_1^{s_1} \cdots i_{q'}^{s'_q} k_1^{r_1} \cdots k_{n'}^{r'_n} \tag{37}$$

with nonnegative integers $s_1, \ldots, s_{q'}, r_1, \ldots, r_{n'}$ are polynomials in $p$ and $H$.

## C. Connection between modular addition and $\nu = 3$ problem

In this section we discuss a direct connection between our general framework and the modular addition task. This connection can be established in the case $\nu = 3$, where the tensor factorization problem can be interpreted as learning modular addition with a simple neural architecture.

### C.1. Modular addition setup

Consider one-hot inputs $\mathbf{x}, \mathbf{y} \in \mathbb{R}^p$, and define a one-hidden-layer bilinear network, where the activation is given by multiplication:

$$f_l(\mathbf{x}, \mathbf{y}) = \sum_{k=1}^{H} w_{kl} \left( \sum_{i=1}^{p} u_{ki} x_i \right) \left( \sum_{j=1}^{p} v_{kj} y_j \right), \qquad l = 1, \ldots, p, \tag{38}$$

where $\{u_{ki}\}, \{v_{kj}\}$ are the input weights and $\{w_{kl}\}$ are the output weights. The target function is the modular addition map

$$f_{*,l}(\mathbf{x}, \mathbf{y}) = (\mathbf{x} * \mathbf{y})_l = \sum_{s=1}^{p} x_s y_{l-s}, \qquad l \in \mathbb{Z}_p, \tag{39}$$

and we train with the quadratic loss

$$L = \frac{1}{2} \mathbb{E}_{\mathbf{x}, \mathbf{y}} \sum_{l=1}^{p} \left( f_l(\mathbf{x}, \mathbf{y}) - f_{*,l}(\mathbf{x}, \mathbf{y}) \right)^2, \tag{40}$$

where the expectation is taken over the training dataset.

### C.2. Fourier analysis

The map (39) is a cyclic convolution in $l$ and therefore diagonalizes under the unitary discrete Fourier transform (DFT) over $\mathbb{Z}_p$. We use the unitary DFT matrix $F \in \mathbb{C}^{p \times p}$ with entries

$$F_{rs} = p^{-1/2} e^{-2\pi i rs/p}, \qquad r, s \in \{1, \ldots, p\}, \tag{41}$$

so that $F^* F = FF^* = I$ (with $F^*$ the Hermitian conjugate). For any vector $\mathbf{a} = (a_1, \ldots, a_p)^\top \in \mathbb{C}^p$ we write

$$\hat{\mathbf{a}} = F\mathbf{a}, \qquad \check{\mathbf{a}} = F^* \mathbf{a}, \tag{42}$$

i.e., componentwise,

$$\hat{a}_r = p^{-1/2} \sum_{s=1}^{p} e^{-2\pi i rs/p} a_s, \qquad \check{a}_s = p^{-1/2} \sum_{r=1}^{p} e^{2\pi i rs/p} a_r. \tag{43}$$

We use $\hat{\mathbf{x}} = F\mathbf{x}, \hat{\mathbf{y}} = F\mathbf{y}, \hat{\mathbf{f}} = F\mathbf{f}$ for the Fourier transforms of inputs/outputs.

When taking the DFT along the *output* index $l$ (with hidden index $k$ fixed), we apply (43) to the vector $(w_{k1}, \ldots, w_{kp})^\top$:

$$\hat{w}_{kl} = p^{-1/2} \sum_{t=1}^{p} e^{-2\pi i lt/p} w_{kt}, \qquad l = 1, \ldots, p, \tag{44}$$

with inverse

$$w_{kt} = p^{-1/2} \sum_{l=1}^{p} e^{2\pi i lt/p} \hat{w}_{kl}, \qquad t = 1, \ldots, p. \tag{45}$$

Similarly, to rewrite the linear forms $\sum_{i=1}^{p} u_{ki} x_i$ and $\sum_{j=1}^{p} v_{kj} y_j$ in the Fourier basis, we apply the *inverse* DFT $F^*$ along the corresponding input indices: for each $k$, define

$$\check{u}_{kr} := p^{-1/2} \sum_{i=1}^{p} e^{2\pi i ri/p} u_{ki}, \qquad \check{v}_{kq} := p^{-1/2} \sum_{j=1}^{p} e^{2\pi i qj/p} v_{kj}, \tag{46}$$

so that, using $F^* F = I$,

$$\sum_{i=1}^{p} u_{ki} x_i = \sum_{r=1}^{p} \check{u}_{kr} \hat{x}_r, \qquad \sum_{j=1}^{p} v_{kj} y_j = \sum_{q=1}^{p} \check{v}_{kq} \hat{y}_q. \tag{47}$$

**Diagonalization of the target.** Applying the forward DFT (in $l$) to (39) and using changes of variables and sum reordering, we get

$$\hat{f}_{*,l}(\hat{\mathbf{x}}, \hat{\mathbf{y}}) = p^{-1/2} \sum_{t=1}^{p} e^{-2\pi i l t/p} f_{*,t}(\mathbf{x}, \mathbf{y}) = p^{-1/2} \sum_{t=1}^{p} e^{-2\pi i l t/p} \sum_{s=1}^{p} x_s y_{t-s} \tag{48}$$

$$= p^{-1/2} \sum_{s=1}^{p} x_s \sum_{t=1}^{p} e^{-2\pi i l t/p} y_{t-s} \qquad \text{(swap the order of summation)} \tag{49}$$

$$= p^{-1/2} \sum_{s=1}^{p} x_s \sum_{u=1}^{p} e^{-2\pi i l(u+s)/p} y_u \qquad \text{(let } u = t - s; \text{ indices are modulo } p) \tag{50}$$

$$= p^{-1/2} \left( \sum_{s=1}^{p} x_s \, e^{-2\pi i l s/p} \right) \left( \sum_{u=1}^{p} e^{-2\pi i l u/p} y_u \right) \qquad \text{(factor } e^{-2\pi i l s/p}) \tag{51}$$

$$= p^{-1/2} \left( \sqrt{p} \, \hat{x}_l \right) \left( \sqrt{p} \, \hat{y}_l \right) = p^{1/2} \, \hat{x}_l \, \hat{y}_l. \tag{52}$$

Thus the modular addition target is diagonal in the Fourier basis:

$$\hat{f}_{*,l}(\hat{\mathbf{x}}, \hat{\mathbf{y}}) = \sqrt{p} \, \hat{x}_l \, \hat{y}_l.$$

**Network in the Fourier basis.** By definition of the DFT of the output,

$$\hat{f}_l(\hat{\mathbf{x}}, \hat{\mathbf{y}}) = p^{-1/2} \sum_{t=1}^{p} e^{-2\pi i l t/p} f_t(\mathbf{x}, \mathbf{y}).$$

Substituting (38) and interchanging sums yields

$$\hat{f}_l(\hat{\mathbf{x}}, \hat{\mathbf{y}}) = \sum_{k=1}^{H} \left( p^{-1/2} \sum_{t=1}^{p} e^{-2\pi i l t/p} w_{kt} \right) \left( \sum_{i=1}^{p} u_{ki} x_i \right) \left( \sum_{j=1}^{p} v_{kj} y_j \right)$$

$$= \sum_{k=1}^{H} \hat{w}_{kl} \underbrace{\left( \sum_{i=1}^{p} u_{ki} x_i \right)}_{= \sum_{r=1}^{p} \check{u}_{kr} \hat{x}_r} \underbrace{\left( \sum_{j=1}^{p} v_{kj} y_j \right)}_{= \sum_{q=1}^{p} \check{v}_{kq} \hat{y}_q} \tag{53}$$

$$= \sum_{k=1}^{H} \hat{w}_{kl} \left( \sum_{r=1}^{p} \check{u}_{kr} \, \hat{x}_r \right) \left( \sum_{q=1}^{p} \check{v}_{kq} \, \hat{y}_q \right), \tag{54}$$

where $\sum_i u_{ki} x_i = \sum_r \check{u}_{kr} \hat{x}_r$ follows from $u_k^\top x = (F^* u_k)^\top (Fx)$ (and similarly for $v_k$).

**Averaging over the full dataset.** Because $F$ is unitary, Parseval's identity lets us evaluate the loss in the Fourier domain:

$$L = \frac{1}{2} \mathbb{E}_{\mathbf{x}, \mathbf{y}} \sum_{l=1}^{p} \left| \hat{f}_l(\hat{\mathbf{x}}, \hat{\mathbf{y}}) - \hat{f}_{*,l}(\hat{\mathbf{x}}, \hat{\mathbf{y}}) \right|^2 \tag{55}$$

$$= \frac{1}{2} \mathbb{E}_{\mathbf{x}, \mathbf{y}} \sum_{l=1}^{p} \left| \sum_{k=1}^{H} \sum_{r=1}^{p} \sum_{q=1}^{p} \hat{w}_{kl} \check{u}_{kr} \check{v}_{kq} \hat{x}_r \hat{y}_q - \sqrt{p} \hat{x}_l \hat{y}_l \right|^2, \tag{56}$$

where we used (54) and (52).

**Computing the expectations inside the sum.** Fix $l \in \{1, \ldots, p\}$ and set

$$A_{rq}^{(l)} := \sum_{k=1}^{H} \hat{w}_{kl} \check{u}_{kr} \check{v}_{kq} \qquad (r, q = 1, \ldots, p).$$

Then (56) reads

$$\hat{f}_l - \hat{f}_{*,l} \;=\; \sum_{r,q} A^{(l)}_{rq}\,\hat{x}_r\,\hat{y}_q \;-\; \sqrt{p}\,\hat{x}_l\,\hat{y}_l.$$

Expand the squared modulus and take the dataset expectation:

$$\mathbb{E}_{\mathbf{x},\mathbf{y}}\Big|\hat{f}_l - \hat{f}_{*,l}\Big|^2 = \mathbb{E}_{\mathbf{x},\mathbf{y}}\left[\Big(\sum_{r,q} A^{(l)}_{rq}\hat{x}_r\hat{y}_q - \sqrt{p}\,\hat{x}_l\hat{y}_l\Big)\Big(\sum_{r',q'} \overline{A^{(l)}_{r'q'}}\,\check{x}_{r'}\check{y}_{q'} - \sqrt{p}\,\check{x}_l\check{y}_l\Big)\right]$$

$$= \underbrace{\sum_{r,q}\sum_{r',q'} A^{(l)}_{rq}\overline{A^{(l)}_{r'q'}}\,\mathbb{E}[\hat{x}_r\check{x}_{r'}]\,\mathbb{E}[\hat{y}_q\check{y}_{q'}]}_{(\mathrm{I})}$$

$$- \underbrace{\sqrt{p}\sum_{r,q} A^{(l)}_{rq}\,\mathbb{E}[\hat{x}_r\check{x}_l]\,\mathbb{E}[\hat{y}_q\check{y}_l]}_{(\mathrm{II})}$$

$$- \underbrace{\sqrt{p}\sum_{r',q'} \overline{A^{(l)}_{r'q'}}\,\mathbb{E}[\hat{x}_l\check{x}_{r'}]\,\mathbb{E}[\hat{y}_l\check{y}_{q'}]}_{(\mathrm{III})}$$

$$+ \underbrace{p\,\mathbb{E}[\hat{x}_l\check{x}_l]\,\mathbb{E}[\hat{y}_l\check{y}_l]}_{(\mathrm{IV})}. \tag{57}$$

Here we used independence of $\mathbf{x}$ and $\mathbf{y}$ to factor the expectations. Since the dataset is the full Cartesian product with a uniform measure and the Fourier modes are orthogonal:

$$\mathbb{E}_{\mathbf{x}}[\hat{x}_r\check{x}_{r'}] = \delta_{rr'}, \qquad \mathbb{E}_{\mathbf{y}}[\hat{y}_q\check{y}_{q'}] = \delta_{qq'}.$$

Therefore each line in (57) simplifies as follows:

$$(\mathrm{I}) \;=\; \sum_{r,q}\sum_{r',q'} A^{(l)}_{rq}\overline{A^{(l)}_{r'q'}}\,\delta_{rr'}\delta_{qq'} \;=\; \sum_{r,q}\big|A^{(l)}_{rq}\big|^2,$$

$$(\mathrm{II}) \;=\; \sqrt{p}\sum_{r,q} A^{(l)}_{rq}\,\delta_{rl}\,\delta_{ql} \;=\; \sqrt{p}\,A^{(l)}_{ll},$$

$$(\mathrm{III}) \;=\; \sqrt{p}\sum_{r',q'} \overline{A^{(l)}_{r'q'}}\,\delta_{lr'}\,\delta_{lq'} \;=\; \sqrt{p}\,\overline{A^{(l)}_{ll}},$$

$$(\mathrm{IV}) \;=\; p\cdot 1\cdot 1 \;=\; p.$$

Putting the pieces together gives, for each fixed $l$,

$$\mathbb{E}_{\mathbf{x},\mathbf{y}}\Big|\hat{f}_l - \hat{f}_{*,l}\Big|^2 \;=\; \sum_{r,q}\big|A^{(l)}_{rq}\big|^2 \;-\; 2\sqrt{p}\,\Re\big(A^{(l)}_{ll}\big) \;+\; p.$$

Finally, use the elementary identity (valid for any $A_{rq}$)

$$\sum_{r,q}\Big|A_{rq} - \sqrt{p}\,\delta_{r=q=l}\Big|^2 = \sum_{r,q}|A_{rq}|^2 - 2\sqrt{p}\,\Re(A_{ll}) + p,$$

to rewrite the previous line as

$$\mathbb{E}_{\mathbf{x},\mathbf{y}}\Big|\hat{f}_l - \hat{f}_{*,l}\Big|^2 \;=\; \sum_{r,q}\Big|A^{(l)}_{rq} - \sqrt{p}\,\delta_{r=q=l}\Big|^2.$$

Summing over $l$ and restoring $A^{(l)}_{rq} = \sum_k \hat{w}_{kl}\check{u}_{kr}\check{v}_{kq}$ yields

$$L = \frac{1}{2}\sum_{l=1}^{p}\sum_{r=1}^{p}\sum_{q=1}^{p}\Big|\sum_{k=1}^{H}\hat{w}_{kl}\check{u}_{kr}\check{v}_{kq} - \sqrt{p}\,\delta_{r=q=l}\Big|^2 = \frac{1}{2}\sum_{i,j,l=1}^{p}\Big|\sum_{k=1}^{H}\hat{w}_{kl}\check{u}_{ki}\check{v}_{kj} - \sqrt{p}\,\delta_{i=j=l}\Big|^2. \tag{58}$$

**Problem simplification.** The following two simplifications let us connect the modular-addition objective to our general CP setup and keep the notation lightweight.

1. **Absorbing the $\sqrt{p}$ factor (w.l.o.g.).** In (58) the target carries a factor $\sqrt{p}$. We remove this by the readout reparameterization

$$\widetilde{w}_{kl} \; := \; p^{-1/2}\, \hat{w}_{kl},$$

which turns $\sqrt{p}\,\delta_{i=j=l}$ into $\delta_{i=j=l}$. This is a one-to-one change of variables on the readout that leaves the optimization landscape unchanged. For brevity we drop the tilde and continue to write $w$.

2. **Passing to real-valued weights (surrogate).** The diagonal expression in (58) is written in complex Fourier coordinates (with conjugate symmetry when spatial-domain weights are real). One could keep the complex formulation or split into real and imaginary parts, but this introduces additional couplings and heavier notation. To streamline the analysis and align with our general framework, we adopt the following *real*, diagonal surrogate. This surrogate is not mathematically identical to the complex formulation; we proceed under the working assumption that it preserves the key features of the corresponding gradient-flow dynamics.

With these conventions, we get the real-valued tensor factorization objective

$$L_{\text{real}} = \frac{1}{2} \sum_{i,j,l=1}^{p} \left| \sum_{k=1}^{H} u_{ki} v_{kj} w_{kl} - \delta_{i=j=l} \right|^2. \tag{59}$$

This is exactly the instance of our general setup (3) with $\nu = 3$ (ASYM); imposing the constraint (2) gives the SYM version.

### C.3. Related works

The primary motivation for the problem of identity tensor decomposition we study in the present paper is the problem of training a two-layer fully-connected network with a quadratic activation function on modular addition. We review various instances of modular arithmetic problems in Deep Learning.

**Modular arithmetic as a case study for grokking.** To the best of our knowledge, the problem of modular arithmetic first appeared in (Power et al., 2022). In this work, the authors noticed that a shallow Transformer (Vaswani et al., 2017) trained to solve various modular arithmetic problems exhibits a remarkable phenomenon dubbed *grokking*: the model achieves perfect generalization much after reaching the perfect train accuracy. By the time the model achieves a perfect train accuracy, the test one usually stays at the random guess level.

Attempts to provide a theoretical explanation for this phenomenon in the original setting (i.e., Transformers trained on modular arithmetic) include *slingshot mechanism* (Thilak et al., 2022) and *circuit formation* (Nanda et al., 2023). This phenomenon has later been observed also for MLPs (Morwani et al., 2024; Mohamadi et al., 2024) and for non-neural models trained not with gradient methods (Mallinar et al., 2025). Grokking has also been observed for various problems that differ from modular arithmetic, including (1) group operations (Chughtai et al., 2023), (2) sparse parity (Barak et al., 2022; Bhattamishra et al., 2023), (3) greatest common divisor (Charton, 2024), (4) image classification (Liu et al., 2022; Radhakrishnan et al., 2022), as well as generic supervised learning problems under specific conditions (Lyu et al., 2024).

**Mechanistic interpretation of learned algorithms for models trained on modular arithmetic.** A problem of modular addition could be naturally solved by first, Fourier-transforming one-hot representations of the summands, convolving them next, and finally, transforming them back to the original space. Applying methods of mechanistic interpretability, (Nanda et al., 2023) demonstrated that a trained Transformer indeed converges to this kind of algorithm. Their conclusion was later refined by (Zhong et al., 2023) who demonstrated that the claimed algorithm (dubbed *Clock*) is not unique a trained Transformer could converge to, but there is another typical one, dubbed *Pizza*, as well as a class of algorithms that mixes between these two. As a sequel, (Furuta et al., 2024) extended their analysis to other modular arithmetic problems.

In the meantime, (Gromov, 2023) demonstrated that a two-layer MLP with a quadratic activation function could implement the above Fourier-feature algorithm in the limit of infinite width by explicitly providing the corresponding weights. Importantly, this work does *not* analyze training dynamics (gradient descent or gradient flow); convergence to the constructed

minimizers is supported empirically rather than proved. However, our goal in the present work is dynamical: we aim to derive explicit, time-dependent formulas for the loss expectation $\mathbb{E}\left[L(t)\right]$ under the gradient flow.

A subsequent work of (Doshi et al., 2024) extends their explicit construction to modular multiplication and modular addition with multiple terms. Finally, a recent work of (McCracken et al., 2026) building on (Nanda et al., 2023) and (Zhong et al., 2023) claims that both MLPs and Transformers trained on modular addition implement an abstract algorithm they dub *approximate Chinese Remainder Problem* implicitly providing the corresponding weights.

# D. Proof of Theorem 4.1

In a nutshell, the proof consists in finding out which diagrams appearing in the merger $(\frac{1}{2}D_{2\nu} - R_\nu)^{\star(s+1)}$ can support an edge pairing that leaves the given numbers of contracted $p$- and $H$-nodes. The easier part of the proof is to show that for particular $(q, n)$ there are suitable diagrams: this can be done by explicit construction. The trickier, complementary part is to show that for particular $(q, n)$ there are no diagrams. This is done by induction on the number $s_D$ of free diagrams appearing in the merger: we show that if there is a diagram with $s_D$ factors $D_{2\nu}$ and some numbers of contracted nodes, then there must be a diagram with $s_D - 1$ factors $D_{2\nu}$ and suitably decremented numbers of nodes. Also, parity considerations allow to rule out exceptional ASYM cases: diagrams merged using an odd number of interaction diagrams $R_\nu$, and free diagrams with fully uncontracted $H$-nodes.

## D.1. Symmetric models, even $\nu$

### D.1.1. REALIZABILITY

We need to prove that for each term

$$p^{1+(\nu-1)s_D-\frac{\nu}{2}(n-1)}H^n\sigma^{\nu(s_D+1)+(\nu-2)s} \tag{60}$$

with some $0 \leq s_D \leq s+1$ and $1 \leq n \leq s_D + 1$, there exists a diagram $G$ obtained by merging, in some order, $s_D$ diagrams $D_{2\nu}$ and $s_R = s + 1 - s_D$ diagrams $R_\nu$, such that some contraction $\widehat{G}$ of $G$ has $n$ $H$-nodes, $1 + (\nu - 1)s_D - \frac{\nu}{2}(n - 1)$ $p$-nodes, and admits an edge pairing. If true, such a diagram contributes the respective term (60) to $\mathbb{E}[(\frac{1}{2}D_{2\nu} - R_\nu)^{\star(s+1)}]$ with the coefficient $\frac{(-1)^{s_R}}{2^{s_D}}$ multiplied by the number of admitted pairings.

Note that there can be no cancellations of contributions of different diagrams: the triplet of powers

$$\begin{pmatrix} q \\ n \\ 2l \end{pmatrix} = \begin{pmatrix} 1 + (\nu - 1)s_D - \frac{\nu}{2}(n - 1) \\ n \\ \nu(s_D + 1) + (\nu - 2)s \end{pmatrix} \tag{61}$$

in (60) is uniquely determined by $s, s_D, n$, therefore all the diagrams with the same triplet of powers have the same coefficient $\frac{(-1)^{s_R}}{2^{s_D}}$. This shows that we only need to find one suitable diagram $G$ to guarantee the presence of the respective term $p^q H^n \sigma^{2l}$ in $\mathbb{E}[(\frac{1}{2}D_{2\nu} - R_\nu)^{\star(s+1)}]$.

We construct desired examples by induction on $s_D$. The base of induction is $s_D = 0$. The respective diagrams $G$ have the form $R_\nu^{\star(s+1)}$ and consist of one $p$-node and one $H$-node connected by $\nu + s(\nu - 2)$ edges. Since $\nu$ is even, these diagrams admit an edge-pairing and contribute a term with $(n, q) = (1, 1)$, consistent with Eq. (60).

Now we make an induction step. As the induction hypothesis, suppose that we have a diagram $G$ satisfying our conditions. We will show that merging $G \star D_{2\nu}$ produces another diagram $G'$ satisfying our conditions, with new numbers $n', q'$ of contracted $p$-nodes and $H$-nodes that also satisfy our conditions. We consider two options.

1. **[Retaining $n$]** We construct the new diagram $G'$ such that the new numbers $n', q'$ are related to the numbers $n, q$ for $G$ by

   $$n' = n, \quad q' = q + \nu - 1. \tag{62}$$

   To this end, we simply take any edge $u_{ki}$ in $G$ and merge $G$ with $D_{2\nu}$ over this edge. This adds a new $H$-node $k'$ and $\nu - 1$ new $p$-nodes $i_1, \ldots, i_{\nu-1}$ to $G$, and replaces the edge $u_{ki}$:

   $$u_{ki} \rightsquigarrow u_{k'i} \prod_{m=1}^{\nu-1} u_{ki_m} u_{k'i_m}. \tag{63}$$

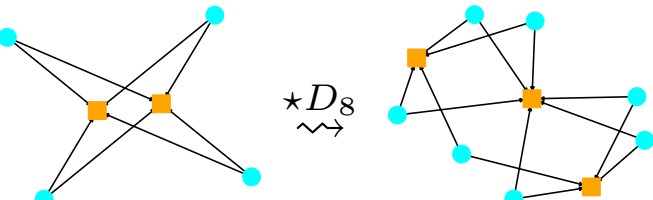

Now, we contract the resulting diagram $G'$ by retaining the contractions in $G$ and additionally contracting the node $k'$ to $k$.

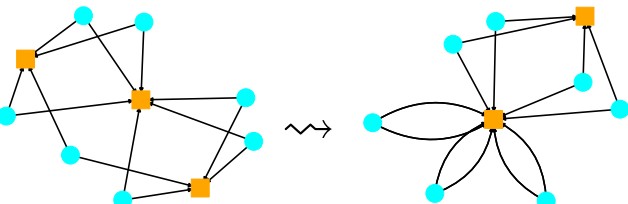

This makes the edges $u_{ki_m}, u_{k'i_m}$ naturally paired, and also allows to identify the new edge $u_{k'i}$ with the old edge $u_{ki}$. We then have an edge pairing in $\widehat{G}'$ obtained by retaining the pairing in $\widehat{G}$ and additionally supplementing it with the pairing of the new edges $u_{ki_m}, u_{k'i_m}$. This creates the desired diagram fulfilling condition (62).

2. **[Increasing $n$]** Alternatively, we can construct a new diagram such that

$$n' = n + 1, \quad q' = q + \frac{\nu}{2} - 1. \tag{64}$$

To this end, we define again $G'$ by merging $G$ with $D_{2\nu}$ over any edge. But now we don't contract the nodes $k$ and $k'$. Instead, we divide the nodes $i_1, \ldots, i_{\nu-1}$ and $i$ into pairs (which is possible since $\nu$ is even) and contract the nodes in each pair. This gives the relation (64).

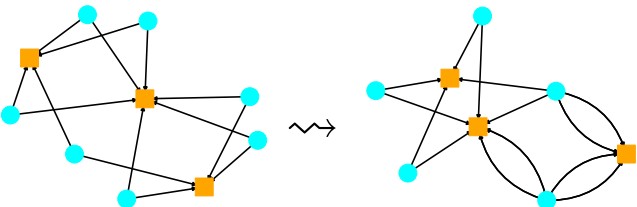

To construct an admissible edge pairing, we form pairs among the edges $(u_{k'i_m})_{m=1,\ldots,\nu-1}$ and $u_{k'i}$ using the introduced contraction. Similarly, we form pairs among the edges $(u_{ki_m})_{m=1,\ldots,\nu-1}$ except for $m_*$ such that $i_{m_*}$ is contracted with $i$ (this $u_{ki_{m_*}}$ is left unpaired because of the missing edge $u_{ki}$ in $G'$). We also retain the edge pairing from $G$, except for the edge $u_{k''u''}$ paired with $u_{ki}$ (again, because $u_{ki}$ is no longer present in $G'$). To complete the pairing in $G'$, we can pair $u_{ki_{m_*}}$ with $u_{k''i''}$, because the node $k''$ is contracted to $k$ while the node $i_{m_*}$ is contracted to $i$ and hence to $i''$.

Clearly, the above two transformations allow to generate all terms (60) (for given $s, s_D, n$ just start from a diagram $R_{2\nu}^{s+1-s_D}$, perform $n$-retaining transformation $s_D + 1 - n$ times and $n$-increasing transformation $n - 1$ times).

### D.1.2. OPTIMALITY

We need to show that each term listed in Eq. (14) is Pareto-optimal with respect to the factor $p^q H^n$, i.e. for given $s, s_D$ there are no diagrams in the binomial expansion $(\frac{1}{2} D_{2\nu} - R_\nu)^{\star(s+1)}$ with exactly $s_D$ factors $D_{2\nu}$ that would contribute a strictly Pareto-dominating factor $p^{q'} H^{n'}$.

Since merging with $D_{2\nu}$ creates one additional $H$-node while merging with $R_\nu$ creates no $H$-nodes, possible values of $n$ are necessarily restricted to $1, \ldots, s_D + 1$. Thus, the optimality condition that we need to prove can be stated as follows: for any contracted diagram with given $s, s_D$ and admitting a pairing, the number of contracted $H$-nodes $n$ and the number of contracted $p$-nodes $q$ satisfy the inequality

$$q \leq 1 + (\nu - 1)s_D - \frac{\nu}{2}(n - 1). \tag{65}$$

We prove this by induction on $s_D$ mimicking the induction used in the proof of realizability. The base of induction is $s_D = 0$, i.e. diagrams $R_\nu^{\star(s+1)}$. Such diagrams consist of one $p$-node and one $H$-node connected by $\nu + s(\nu - 2)$ edges. Since $\nu$ is even, such diagrams admit pairings and thus contribute a term with $(n, q) = (1, 1)$, satisfying condition (65).

Now we make the induction step. We will use the following general argument, Let $G$ be some diagram obtained by merging $s_D$ diagrams $D_{2\nu}$ and some number of diagrams $R_\nu$, in some order. Suppose that $G$ has a contraction – call it $\widehat{G}$ – that has $n$ $H$-nodes and $q$ $p$-nodes and admits a pairing. We will then construct a diagram $G'$ obtained by merging $s_D - 1$ diagrams $D_{2\nu}$ and some number of diagrams $R_\nu$, in some order, such that $G'$ has some contraction $\widehat{G}'$ admitting a pairing and having $n'$ $H$-nodes and $q'$ $p$-nodes, where

$$q' + \frac{\nu}{2}n' \geq q + \frac{\nu}{2}n - \nu + 1. \tag{66}$$

Condition (65) then follows for $G$ by the induction hypothesis:

$$q \leq q' + \frac{\nu}{2}(n' - n) + \nu - 1 \tag{67}$$

$$\leq 1 + (\nu - 1)(s_D - 1) - \frac{\nu}{2}(n' - 1) + \frac{\nu}{2}(n' - n) + \nu - 1 \tag{68}$$

$$= 1 + (\nu - 1)s_D - \frac{\nu}{2}(n - 1). \tag{69}$$

Now we detail the construction of $G'$. The given diagram $G$ can be viewed as obtained by first merging $s_D - 1$ diagrams $D_{2\nu}$ and some diagrams $R_\nu$ into a diagram $G_1$, then merging $G_1$ with $D_{2\nu}$, and then merging with some number of $R_\nu$'s[2]:

$$G \in (\ldots ((G_1 \star D_{2\nu}) \star R_\nu) \ldots) \star R_\nu. \tag{70}$$

Observe that in the present case of symmetric models of even order $\nu$, merging with $R_\nu$ simply adds an even number of edges between some $H$-node and some $p$-node. If any diagram $F$ merged with $R_\nu$ admits a pairing, then $F$ also admits a pairing, with the same contraction (if necessary, we can rewire a pairing in $F \star R_\nu$ so that the edges of $F$ are paired within themselves, and the newly added edges are also paired within themselves).

Thus, without loss of generality we can ignore the trailing mergers with $R_\nu$ in Eq. (70) and simply assume that

$$G \in G_1 \star D_{2\nu}. \tag{71}$$

Merging $G_1$ with $D_{2\nu}$ over an edge $u_{ki}$ in $G_1$ consists in creating $\nu - 1$ new $p$-nodes $i_1, \ldots, i_{\nu-1}$ and one new $H$-node $k'$, and replacing the edge $u_{ki}$ with new edges:

$$u_{ki} \rightsquigarrow u_{k'i} \prod_{m=1}^{\nu-1} u_{ki_m} u_{k'i_m}. \tag{72}$$

Now consider several possibilities regarding the structure of the contracted diagram $\widehat{G}$.

1. **The new $H$-node $k'$ is not contracted to any other node in $\widehat{G}$.** In this case we will show that a desired diagram $G'$ exists with

$$n' = n - 1, \quad q' \geq q - \frac{\nu}{2} + 1, \tag{73}$$

   which satisfies condition (66).

---

[2]Here, the symbol $\in$ means that $G$ is one of the several diagrams produced by merging.

Specifically, since the new $H$-node $k'$ is not contracted to any other node, the incident edges $(u_{k'i_m})_{m=1,\ldots,\nu-1}$ and $u_{k'i}$ must be paired within each other. This requires at least $\frac{\nu}{2}$ contractions among the $\nu$ $p$-nodes $i_1,\ldots,i_{\nu-1}$ and $i$. Consider now the pairings in $\widehat{G}$ of the new edges $(u_{ki_m})_{m=1,\ldots,\nu-1}$ involving the old node $k$. Without loss of generality (by rewiring the pairings if necessary), we can assume these pairings to mimic the respective pairings of the edges $(u_{k'i_m})_{m=1,\ldots,\nu-1}$ and $u_{k'i}$, since they can be served by the same contractions of the nodes $i_1,\ldots,i_{\nu-1}$ and $i$. The only exception involves the edge paired with $u_{k'i}$ – call it $u_{k'j}$ (where $j \in \{i_1,\ldots,i_{\nu-1}\}$). The edge $u_{kj}$ is then paired in $\widehat{G}$ with some other edge $u_{kh}$, where $h$ is some node in $G$ contracted with $i$.

Now define $G'$ simply as $G_1$, and define the contraction $\widehat{G}'$ to be inherited from the contraction $\widehat{G}$. Since $k'$ was not contracted to any node, we have $n' = n - 1$ for the number of $H$-nodes. Also, as remarked, there were at least $\frac{\nu}{2}$ contractions among the nodes $i_1,\ldots,i_{\nu-1}$ and $i$, so removing these $\nu$ $p$-nodes means removing at most $\frac{\nu}{2} - 1$ contracted $p$-nodes. This ensures condition (73). The pairing admitted by $G'$ is the pairing inherited from $G$, with the exception that, on the one hand, we now need to pair the edge $u_{ki}$ not present in $G$, and we need to pair the edge $u_{kh}$ that in $\widehat{G}$ was paired with the edge $u_{kj}$ not present in $G'$. However, as pointed out, the nodes $j$ and $h$ were contracted in $G$, so we can just pair the edges $u_{kh}$ and $u_{kj}$.

2. **The new $H$-node $k'$ is contracted to the node $k$.** In this case we will show that a desired diagram $G'$ exists with

$$n' = n, \quad q' \geq q - \nu + 1. \tag{74}$$

Indeed, in this case each edge $u_{ki_m}$ is naturally paired with $u_{k'i_m}$ (for $m = 1,\ldots,\nu-1$), and also $u_{k'i}$ can be identified with $u_{ki}$. By rewiring if necessary, we can assume this pairing structure without loss of generality. We can then again choose $G'$ simply as $G_1$ with inherited contraction – this gives above conditions for $n', q'$. The pairing in $G$ is naturally restricted to a pairing in $G'$.

3. **The new $H$-node $k'$ is not contracted to the node $k$, but is contracted to some other nodes in $G$.** In this case we will again show that a desired diagram $G'$ exists with

$$n' = n, \quad q' \geq q - \nu + 1. \tag{75}$$

Let us collectively denote all the nodes contracted to $k$ in $G$ by $K$, and all the nodes contracted to $k'$ by $K'$. Also collectively denote by $I_1,\ldots,I_r$ all the contracted groups of $p$-nodes to which the nodes $i_1,\ldots,i_{\nu-1}$ and $i$ belong. The number $r$ of such groups is, clearly, not larger than $\nu$.

Since $k'$ is not contracted to $k$, $k$ does not belong to $K'$, and $k'$ to $K$. Accordingly, the new edges $(u_{ki_m})_{m=1,\ldots,\nu-1}$ cannot be paired with edges $(u_{k'i_m})_{m=1,\ldots,\nu-1}$. If $i_m$ belongs to a contracted group $I_t$, the edge $u_{ki_m}$ is paired with some edge $u_{\widetilde{k}\widetilde{j}}$ with $\widetilde{k} \in K, \widetilde{j} \in I_t$, and the edge $u_{k'i_m}$ is paired with some edge $u_{\widetilde{k}'\widetilde{j}'}$ with $\widetilde{k}' \in K, \widetilde{j}' \in I_t$. The latter also applies to the edge $u$.

Now define $G'$ as $G_1$. Define the contraction $\widehat{G}'$ as inherited from $\widehat{G}$, but with the additional contraction of all the groups $I_1,\ldots,I_r$ into a single group $I$. Since $k'$ was contracted and $r \leq \nu$, we have conditions (75).

Now we check that there is a pairing admitted by this contraction. Consider the set of edges $u_{\widetilde{k}'\widetilde{j}'}$ introduced above. They were paired in $G$ with the edges $u_{k'i_m}$ and $u_{k'i}$ that are no longer present in $G'$. Since $\nu$ is even, the number of such edges $u_{\widetilde{k}'\widetilde{j}'}$ is even. Since we have contracted all the sets $I_1,\ldots,I_r$ into $I$, all these edges have the same contracted nodes ($p$-node $I$ and $H$-node $K'$) and can be paired.

A similar argument applies to the edges $u_{\widetilde{k}\widetilde{j}}$, but the number of these edges is odd. Accordingly, we can pair all of them with each other except for one. However, we recall that $G_1$ has the edge $u_{ki}$ not present in $G$. The unpaired edge $u_{\widetilde{k}\widetilde{j}}$ can then be paired with this $u_{ki}$.

All the other edge pairs in $\widehat{G}'$ are inherited from $\widehat{G}$.

This completes the induction step and thus the proof for symmetric models with even $\nu$.

## D.2. Asymmetric models

### D.2.1. REALIZABILITY

We need to prove that for each $s, n, s_D$ such that

$$0 \leq s_D \leq s + 1, \quad 1 \leq n \leq s_D + 1, \quad s_R = s + 1 - s_D \text{ is even and } (s_D, n) \neq (s+1, s+2), \tag{76}$$

there exists a diagram $G$ constructed by merging $s_D$ diagrams $D_{2\nu}$ and $s_R$ diagrams $R_\nu$, and its contraction $\widehat{G}$ admitting a pairing and having $q$ $p$-nodes and $n$ $H$-nodes, where

$$q = 1 + (\nu - 1)(s_D + 1 - n). \tag{77}$$

As in the symmetric, even-$\nu$ case discussed in Section D.1.1, all the diagrams with the same triplet $(q, n, 2l)$ (where $2l$ is the number of edges) contribute a coefficient of the same sign, so we just need to produce one such diagram.

The key difference between the present asymmetric case and the symmetric case discussed in Section D.1.1 is that the edges are now colored, so edge pairing requires not only that the edges have the same node (after contraction), but also the same color.

An important observation is that merging any diagram $G$ with $R_\nu$ changes the parity of the total number of edges of each particular color, while merging with $D_{2\nu}$ preserves this parity. In particular, this explains the constraint in (76) that the number $s_R$ of mergers with $R_\nu$ must be even.

The second special constraint in (76), that $(s_D, n) \neq (s + 1, s + 2)$, means that diagrams $D_{2\nu}^{\star(s+1)}$, involving only $D_{2\nu}$ but not $R_\nu$, cannot admit an edge pairing without contracting at least two $H$-nodes. This follows by observing that the new $H$-node created by merging any diagram with $D_{2\nu}$ has $\nu$ edges of different colors (one per color), so must be contracted to some other nodes for a valid edge pairing.

The *n-retaining* transformation considered in Section D.1.1 (see Eq. (62)) remains valid in the present colored context since it involves pairing same-color edges of $D_{2\nu}$. In contrast, the *n-increasing* transformation (see Eq. (64)) is no longer valid since it involves pairing edges that are now differently colored.

It is possible to consider a suitable version of the $n$-increasing transformation in the present colored context, but it requires extra assumptions on the structure of the diagram $G$ appearing in the induction step. Therefore, we find it more convenient to directly construct a suitable diagram for given $(s, s_D, n)$.

First, consider the case $s_D = 0$. Note that this case is realized only if $s$ is odd (otherwise $s_R = s + 1$ is odd). The relevant diagrams are those appearing in $R_\nu^{\star(s+1)}$. These diagram have one $p$-node and one $H$-node connected by multiple edges. By above remark, since $s + 1$ is even, the parity of the number of edges of each color is even, so there is a valid pairing, as desired.

Note also that, similarly and more generally, given any diagram $G$ admitting an edge pairing, we can always merge it with an even number of diagrams $R_\nu$ without changing the number of nodes and so as to extend the pairing.

Now consider a general case with $s_D \geq 1$. It is useful to construct an auxiliary "ring-like" diagram $D_{2\nu}^{\star s_D}$ as follows. Pick some color, say $m = \nu$. Merge $s_D$ diagrams $D_{2\nu}$, always over an edge of this color. As a result we get a ring-like diagram with $s_D + 1$ $H$-nodes and $\nu + s_D(\nu - 1)$ $p$-nodes:

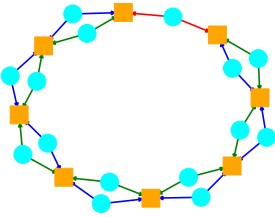

(In this figure $\nu = 3, s_D = 7$.) There are two edges of color $\nu$ (red in the above figure) connecting one $p$-node to two $H$-nodes. All the other edges form $s_D$ groups of $2(\nu - 1)$ edges of colors $1, \ldots, \nu - 1$ (blue and green in the above figure), each connecting $\nu - 1$ $p$-nodes to two $H$-nodes.

Let us now consider separately the cases $n \leq s_D$ and $n = s_D + 1$.

1. $n \leq s_D$. In this case we can construct a desired diagram with $n$ $H$-nodes and $q = 1 + (\nu - 1)(s_D - 1 - n)$ $p$-nodes by suitably contracting the above ring-like diagram. Specifically, first contract the two $H$-nodes having the $\nu$-colored edge:

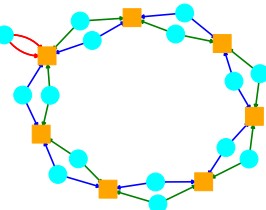

Then additionally contract a contiguous sequence of $s_D - n + 1$ $H$-nodes so as to bring the total number of $H$-nodes to $n$:

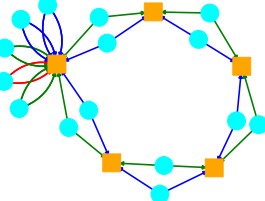

Finally, for each color, contract all the $(\nu - 1)(s_D - 1 - n)$ $p$-nodes remaining in the ring and incident to edges of this color:

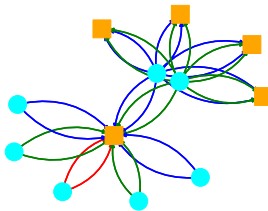

The result is a diagram with exactly $n$ $H$-nodes and $q = 1 + (\nu - 1)(s_D - n + 1)$ $p$-nodes, as desired. Thanks to the implemented contractions of $H$-nodes and $p$-nodes, the diagram admits edge pairing.

The diagram was constructed without merging with diagrams $R_\nu$, but, as remarked, we can merge it with $s_R = s + 1 - s_D$ such diagrams (assuming this number is even) while maintaining the numbers of $H$- and $p$-nodes and extending the edge pairing.

2. $n = s_D + 1$. This case requires all $H$-nodes to stay uncontracted. However, the constructed ring-like diagram $D_{2\nu}^{\star s_D}$ contains two $\nu$-colored edges connected to different $H$-nodes, and so it cannot admit an edge pairing without contracting these nodes. But we can resolve this issue by merging the ring diagram twice with $R_\nu$ over these two special edges:

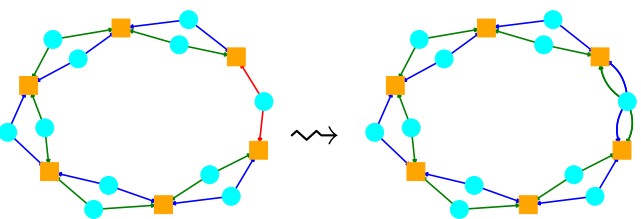

The resulting diagram then has only edges of colors $1, \ldots, \nu - 1$. Edges of each color form a circle going through each $H$-node; all the circles also connect at one $p$-node. We can contract all the $p$-nodes into one node, getting a flower-like diagram with $s_D + 1$ petals, each consisting of $\nu - 1$ pairs of edges of colors $1, \ldots, \nu - 1$:

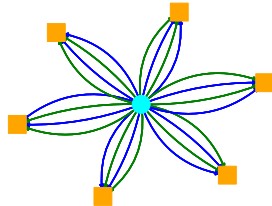

Such a diagram admits an edge pairing and has $(n, q) = (s_D + 1, 1)$, as desired. We can then further merge it with an even number of diagrams $R_\nu$ while keeping these properties.

The described construction requires merging with two diagrams $R_\nu$, so, as expected, it works for all $s$ except $s+1 = s_D$, when $s_R = 0$ and $n = s + 2$. This is exactly the exceptional, unrealized case.

### D.2.2. OPTIMALITY

We have already argued in Section D.2.1 that the terms with odd $s_R$ and $(s_D, n) = (s + 1, s + 2)$ are missing from the expansion. It remains to establish a bound on feasible $q$ analogous to the bound (65) from the symmetric, even $\nu$ scenario:

$$q \leq 1 + (\nu - 1)(s_D + 1 - n). \tag{78}$$

We generally follows the logic of the proof given in Section D.1.2 for that scenario, though the present asymmetric scenario is slightly more complex due to a more subtle influence of diagrams $R_\nu$.

As before, we perform induction on the number $s_D$ of factors $D_{2\nu}$ appearing in a sequence of merging operations (along with some factors $R_\nu$). The base of induction is $s_D = 0$, i.e. diagrams $R_\nu^{s+1}$. They have a vanishing contribution for $s$ even, and a nonzero contribution for $s$ odd with $n = q = 1$, by parity considerations.

For the induction step, suppose again that a diagram $G$ is obtained from a diagram $G_1$ by first merging it with $D_{2\nu}$ and then additionally merging with some number of $R_\nu$'s:

$$G \in (\ldots ((G_1 \star D_{2\nu}) \star R_\nu) \ldots) \star R_\nu. \tag{79}$$

Consider an $m$-colored edge $u_{ki}^{(m)}$ in $G_1$ with a $H$-node $k$ and $p$-node $i$. Without loss of generality, assume $m = \nu$. The merger $G_1 \star D_{2\nu}$ then replaces this edge by

$$u_{ki}^{(\nu)} \rightsquigarrow u_{k'i}^{(\nu)} \prod_{m=1}^{\nu-1} u_{ki_m}^{(m)} u_{k'i_m}^{(m)}. \tag{80}$$

In contrast to the symmetric, even-$\nu$ scenario, we can no longer discard now all the trailing diagrams $R_\nu$, because now they can significantly affect the pairing possibilities.

Observe, however, that merging has *partial commutativity*, in the following sense. Given any three diagrams $G_1, G_2, G_3$, consider the difference between the merging sequences $(G_1 \star G_2) \star G_3$ and $(G_1 \star G_3) \star G_2$. Note that if in $(G_1 \star G_2) \star G_3$ the merger with $G_3$ is performed over an edge in $G_1 \star G_2$ already present in $G_1$, then the same resulting diagram will appear in $(G_1 \star G_3) \star G_2$. In other words, the difference between $(G_1 \star G_2) \star G_3$ and $(G_1 \star G_3) \star G_2$ is exclusively due to the second mergers being over new edges created in the first mergers.

In particular, in sequence (79) we can commute all mergers with $R_\nu$ not performed over the new edges $u_{k'i}^{(\nu)}, (u_{ki_m}^{(m)}, u_{k'i_m}^{(m)})_{m=1}^{\nu-1}$ to be performed before the merger with $D_{2\nu}$. Accordingly, we can assume without loss of generality that the mergers with $R_\nu$ in (79) are performed over these new edges or the edges resulting from subsequent mergers with $R_\nu$.

Moreover, if we merge a diagram $G_1$ with $R_\nu$ over some edge with nodes $k$ and $i$, this simply adds new edges between these two nodes. If we continue the process by merging again with $R_\nu$ over some edges between $k$ and $i$, each merger changes the parity of the number of edges of each color. Accordingly, if the initial diagram admitted an edge pairing, then the diagram resulting after an even number of such mergers will admit an edge pairing. This shows, in particlar, that in (79) we don't need to consider more than one merger with $R_\nu$ over each specific edge among $u_{k'i}^{(\nu)}, (u_{ki_m}^{(m)}, u_{k'i_m}^{(m)})_{m=1}^{\nu-1}$.

We consider now the particular options of transition (analogous to those in Section D.1.2) from $G$ given by (79) to a smaller diagram $G'$ also admitting an edge pairing. To establish desired condition (78), we ensure that at each step we have

$$q' + (\nu - 1)n' \geq q + (\nu - 1)(n - 1). \tag{81}$$

1. **The new $H$-node $k'$ is not contracted to any other node in $\widehat{G}$.** In this case we show that a desired diagram $G'$ exists with

$$n' = n - 1, \quad q' \geq q, \tag{82}$$

which satisfies condition (81).

First we claim that, by color parity considerations, existence of an edge pairing in $G$ in this case requires the sequence (79) to include some trailing mergers with $R_\nu$ over the edges $(u_{k'i_m}^{(m)})_{m=1,\ldots,\nu-1}$ and $u_{k'i}^{(\nu)}$; moreover all new $p$-nodes $i_1, \ldots, i_\nu$ must be contracted together and to $i$.

Indeed, take some subset $A$ of the nodes $i_1, \ldots, i_{\nu-1}, i$ and consider the vector of total color parities of the edges between $k'$ and $A$ (one component of the vector corresponds to one color). Before merging with $R_\nu$, this vector has odd components for nodes in $A$ and even components for nodes in $\{i_1, \ldots, i_{\nu-1}, i\} \setminus A$. Merging with $R_\nu$ over any edge between $k'$ and $A$ changes the parity of each of the $\nu$ colors. If $A$ represents a set of contracted nodes, then for an edge pairing we need to make each component in the respective color parity vector even, This is only possible if $A$ is the whole set $\{i_1, \ldots, i_{\nu-1}, i\}$ and mergers with $R_\nu$ are performed an odd number of times over some edges between $k'$ and $A$.

As a result, if we consider the contraction of the diagram $G_1$ inherited from the contraction of the diagram $G$ admitting a pairing, then it has the same number of $p$-nodes and one fewer $H$-node, in agreement with (82). However, we still need to endow $G_1$ with an edge pairing.

As usual, we form the edge pairing by inheriting it with suitable modifications from the one in $G$. As discussed, the new edges in $G$ connecting $k'$ with $i_1, \ldots, i_{\nu-1}, i$ were paired within each other, so we only need to consider pairings involving the edges connecting $k$ with $(i_m)_{m=1,\ldots,\nu-1}$ and $i$. In $G$, all these $p$-nodes are contracted and the color parity vector for the edges $u_{ki_1;1}, \ldots, u_{ki_{\nu-1}}^{(\nu-1)}$ in $G$ is $(1, \ldots, 1, 0)$. Edge pairing is ensured by complementary edges either present in $G_1$ between $p$-node $i$ and some $H$-nodes contracted with $k$, or created by merging with $R_\nu$. The respective complementary parity vector is also $\mathbf{v} = (1, \ldots, 1, 0)$; if we exclude merging with $R_\nu$, then it may be $\mathbf{v}' = (0, \ldots, 0, 1)$. Now in $G_1$ we have the edge $u_{ki}^{(\nu)}$ instead of the edges $u_{ki_1;1}, \ldots, u_{ki_{\nu-1}}^{(\nu-1)}$ in $G$. This edge has exactly the parity vector $\mathbf{v}'$. By merging with $R_\nu$ over this edge, we can also create a set of edges with the parity vector $\mathbf{v}$. Thus, if $G$ admits a pairing, then also either $G_1$ or $G_1 \star G$ admits a pairing.

2. **The new $H$-node $k'$ is contracted to the node $k$.** In this case a desired diagram $G'$ exists with

$$n' = n, \quad q' \geq q - \nu + 1. \tag{83}$$

Indeed, in this case each edge $u_{ki_m}^{(m)}$ is naturally paired with $u_{k'i_m}^{(m)}$ (for $m = 1, \ldots, \nu - 1$), and also $u_{k'i}^{(\nu)}$ can be identified with $u_{ki}^{(\nu)}$. If there were no subsequent mergers with $R_\nu$ over these edges (after the merger with $D_{2\nu}$), then the desired $G'$ is simply $G_1$ with the pairing inherited from $G$ by removing the pairs $(u_{k'i_m}^{(m)}, u_{ki_m}^{(m)})$.

Suppose now that there have been mergers with $R_\nu$ over the edges in question, say with $u_{ki_m}^{(m)}$ with some $m \in \{1, \ldots, \nu - 1\}$. Such a merger changes the color parity vector of the node $i_m$ from $(0, \ldots, 0)$ to $(1, \ldots, 1)$. Paring the resulting edges requires the node $i_m$ to be contracted to some other node $i'$ (or several nodes) that provide counterpart edges. These counterpart edges must collectively also have the color parity vector $(1, \ldots, 1)$. If we remove the edges at the node $i_m$, since their color parity vector is $(1, \ldots, 1)$, they can be compensated in the pairs with edges at the node $i'$ by new edges at $i'$ created through merging with $R_\nu$ over one of the existing edges.

This argument shows that in any case we can take $G'$ as $G_1$, possibly merged with some $R_\nu$. The bound (83) holds in all cases (if there are extra mergers with $R_\nu$, the bound can even be strengthened since in these cases some of the vertices $i_m$ must be contracted).

3. **The new $H$-node $k'$ is not contracted to the node $k$, but is contracted to some other nodes in $G$.** In this case, again, a desired diagram $G'$ exists with

$$n' = n, \quad q' \geq q - \nu + 1. \tag{84}$$

*Table 2.* Some extremal faces along with their interpretations.

| Face | Scaling condition | Natural $T$ | Parameterization | Learning | Interpretation |
|------|-------------------|-------------|------------------|----------|----------------|
| SYM, even $\nu$ | | | | | |
| A – B | $H^2 \asymp p^\nu,\, p^{\nu-1}\sigma^\nu \to \infty$ | $p^{\nu-1}\sigma^{2\nu-2}$ | Balanced | No | Free evolution |
| B – E | $p^{\nu-2}H^2\sigma^{2\nu} \asymp 1,\, H^2/p^\nu \to \infty$ | $\sigma^{\nu-2}$ | Over- | Rich | Mean-field |
| E | $p^{\nu-2}H^2\sigma^{2\nu} \to 0,\, p^{\nu-1}\sigma^\nu \to 0$ | $\sigma^{\nu-2}$ | Agnostic | Rich | — |
| E – A | $p^{\nu-1}\sigma^\nu \asymp 1,\, H^2/p^\nu \to 0$ | $\sigma^{\nu-2}$ | Under- | Rich | — |
| ASYM | | | | | |
| A – B | $H \asymp p^{\nu-1},\, p^{\nu-1}\sigma^\nu \to \infty$ | $H\sigma^{2\nu-2}$ | Balanced | No | Free evolution |
| B – C | $p^{\nu-1}H\sigma^{2\nu} \asymp 1,\, H\sigma^\nu \to \infty$ | $H\sigma^{2\nu-2}$ | Over- | Lazy | NTK |
| C | $p^{\nu-1}H\sigma^{2\nu} \to 0,\, H\sigma^\nu \to \infty$ | $H\sigma^{2\nu-2}$ | Over- | Lazy | NTK, $f(0) \equiv 0$ |
| C – D | $H\sigma^\nu \asymp 1,\, H/p^{\nu-1} \to \infty$ | $\sigma^{\nu-2}$ | Over- | Rich | Mean-field |
| D – E | $H \asymp p^{\nu-1},\, H\sigma^\nu \to 0$ | $\sigma^{\nu-2}$ | Balanced | Rich | — |
| E – A | $p^{\nu-1}\sigma^\nu \asymp 1,\, H/p^{\nu-1} \to 0$ | $\sigma^{\nu-2}$ | Under- | Rich | — |

The argument is similar to the respective proof for symmetric models in Section D.1.2 and also to arguments above (remove the nodes $i_1, \dots, i_{\nu-1}, i$ and $k'$ and contract together all the $p$-nodes contracted with some of $i_1, \dots, i_{\nu-1}, i$; possibly merge $G_1$ with $R_\nu$ to compensate for the removed edges).

This completes the proof of the theorem in the asymmetric scenario.

## E. Extreme regimes

We interpret some of the extremal simplices of the hyperparameter polygon, Figure 2, below. Our conclusions are summarized in Table 2. We compute the natural scaling for the inverse learning rate $T$ by balancing the base of power $s$ in the leading Pareto-term provided by Theorem 4.1.

### E.1. Symmetric scenario

In this scenario, as Theorem 4.1 states, the hyperparameter polygon is a triangle: see Figure 2, left. Let us denote its summits as A, B, and E, as depicted on the figure.

**Edge A-B.** Here we have $s_D = s + 1$, hence $s_R = 0$. This means that the target is never used for computing the loss derivatives. We call this regime *free evolution*. It becomes dominant when $p^\nu \asymp H^2$, while $p^{\nu-1}\sigma^\nu \to \infty$. That is, when the initialization is huge, the identity target is essentially zero compared to the initial model, and the model attempts to learn zero throughout most of its training time. The natural scaling for $T$ is $T \asymp p^{\nu-1}\sigma^{2\nu-2}$.

The whole edge stays dominant when the model is balanced: $H \asymp p^{\nu/2}$. When this condition gets violated, either Point A (underparameterization), or Point B (overparameterization) becomes solely dominant.

**Edge B-E.** Here we have $n = s_D + 1$, hence $Q(n, s_D) = 1 + \left(\frac{\nu}{2} - 1\right) s_D$. It becomes dominant when $p^{\nu-2}H^2\sigma^{2\nu} \asymp 1$, while $H^2/p^\nu \to \infty$. That is, $H$ has to be large compared to some power of $p$, hence this regime is overparameterized. The natural scaling for $T$ is $T \asymp \sigma^{\nu-2}$.

Recall that $\nu = 2$ corresponds to matrix factorization, and the model is a conventional two-layer fully-connected network. In this case, the edge B-E becomes dominant when $\sigma^2 \asymp 1/H$, while $H/p \to \infty$. This is nothing but a hyperparameter scaling that yields a mean-field limit discovered independently in a number of works (see Appendix A.1). For two-layer nets, the training dynamics could be expressed as an evolution of a measure. Under the above scaling, the moments of this measure stay finite in the limit.

**Edge E-A.** Here we have $n = 1$, hence $Q(n, s_D) = 1 + (\nu - 1)s_D$. It becomes dominant when $p^{\nu-1}\sigma^\nu \asymp 1$, while $H^2/p^\nu \to 0$. The natural scaling for $T$ is $T \asymp \sigma^{\nu-2}$.

For $\nu = 2$, the edge E-A becomes dominant when $\sigma^2 \asymp 1/p$, while $p/H \to \infty$. That is, the required scaling coincides with

the mean-field one with $p$ and $H$ swapped.

**Point E.** This summit deserves a special consideration. Recall that for Edge A-B the initialization variance $\sigma^2$ has to be large, while for the other two edges, it has to balance the growth of $p$ and $H$ in a specific way. For Point E to become dominant, this variance has to be small: $p^{\nu-2}H^2\sigma^{2\nu} \to 0$ along with $p^{\nu-1}\sigma^\nu \to 0$. For $\nu = 2$, this yields a saddle-to-saddle regime extensively studied for linear networks: see Appendix A.2. However, in our case, when the target is identity, the only saddle the training process encounters is the one at the origin. This makes the saddle-to-saddle regime trivial. Same as for Edges B-E and E-A, the natural scaling for $T$ is $T \asymp \sigma^{\nu-2}$.

### E.2. Asymmetric scenario

As Theorem 4.1 states, one or two extremal points are missing compared to the triangle of the symmetric scenario. This gives a quadrangle for odd $s$, or a pentagon for even $s$, hence more edge-cases: see Figure 2, right. In particular, the point C which appears as an extremal point due to exclusion of Point B from the triangle, yields a linearized training regime which was absent in the symmetric scenario; see below.

**Edge A-B.** This is the same free evolution as before, but the dominance condition changes: $p^{\nu-1} \asymp H$, while $p^{\nu-1}\sigma^\nu \to \infty$ and the natural scaling for $T$ is $T \asymp p^{\nu-1}\sigma^{2\nu-2}$ as before, or, equivalently, $T \asymp H\sigma^{2\nu-2}$. Note that this condition coincides with that of the symmetric case when $\nu = 2$ (for matrix factorization).

**Point C.** This point corresponds to $(n, s_D) = (s, s-1)$. In this case, $s_R = 2$, hence in contrast to free evolution, the target does appear in loss derivatives but only weakly.

This corresponds to a linearized learning regime akin to NTK, see Appendix A.1, but starting from a zero model.

The natural scaling for $T$ is $T \asymp H\sigma^{2\nu-2}$. Point C becomes dominant when $p^{\nu-1}H\sigma^{2\nu} \to 0$, while $H\sigma^\nu \to \infty$. In particular, this implies that $H/p^{\nu-1} \to \infty$, hence this regime is overparameterized.

**Edge B-C.** This edge always contains only two Pareto-optimal points: B and C. Point B adds terms with $s_R = 0$; because of this, the corresponding regime is still linearized but the initial model is no longer zero.

Edge B-C becomes dominant when $p^{\nu-1}H\sigma^{2\nu} \asymp 1$, while $H\sigma^\nu \to \infty$. We again need $H/p^{\nu-1} \to \infty$, hence this regime is still overparameterized. The natural scaling for $T$ is again $T \asymp H\sigma^{2\nu-2}$. When $\nu = 2$, for this edge to become dominant one needs $\sigma^2 \asymp 1/\sqrt{pH}$. This is nothing but a conventional NTK initialization scaling: see Appendix A.1.

We note that the original NTK paper of (Jacot et al., 2018) applies a constant learning rate, while we claim $T \asymp H\sigma^2$ to be natural for $\nu = 2$. There is no contradiction. The reason of this discrepancy lies in the fact that (Jacot et al., 2018) initialize neural network's weights from $\mathcal{N}(0, 1)$, while putting $\sigma$ as a pre-factor in the form $\phi(\sigma \times W)$ for $\phi$ being an activation function and $W$ being a weight matrix. In contrast, we initialize our weights from $\mathcal{N}(0, \sigma^2)$ with no pre-factors. The GF dynamics in these two cases become equivalent if we rescale the learning rate appropriately; this gives us the claimed scaling for $T$.

**Edge C-D.** Here we have $n = s_D + 1$, hence $Q(n, s_D) = 1$. It becomes dominant when $H\sigma^\nu \asymp 1$, while $p^{\nu-1}\sigma^\nu \to 0$. In particular, this implies that $H/p^{\nu-1} \to \infty$, hence this regime is overparameterized. Akin to the symmetric scenario, we recover a conventional mean-field scaling when $\nu = 2$.

The natural scaling for $T$ is $T \asymp \sigma^{\nu-2}$.

**Edge D-E.** Here we have $s_D = 1$ for even $s$ and $s_D = 0$ for odd $s$. Therefore $s_R$ is maximal, and only the highest-order target correlations survive in loss derivatives.

The natural scaling for $T$ is again $T \asymp \sigma^{\nu-2}$. For this edge to become dominant, one needs $H\sigma^\nu \to 0$, while $H \asymp p^{\nu-1}$. In words, it is a critically overparameterized regime with small initialization. Same as for Point E under the symmetric scenario, this edge yields a trivial (single saddle) case of a well-studied saddle-to-saddle regime for linear nets: see Appendix A.2.

**Edge E-A.** Similar to the symmetric scenario, we have $n = 1$, hence $Q(n, s_D) = 1 + (\nu - 1)s_D$. This edge becomes dominant when $p^{\nu-1}\sigma^\nu \asymp 1$, while $H/p^{\nu-1} \to 0$. That is, here we have a critically initialized insufficiently overparameterized

regime. Akin to the symmetric scenario, the dominance condition gives that of the mean-field regime (Edge C-D) after swapping $H$ and $p$ when $\nu = 2$.

The natural scaling for $T$ is again $T \asymp \sigma^{\nu-2}$.

## F. Recurrence relations and partial differential equations

We first clarify the context of Theorem 6.1 and then provide its proof. The coefficients $C_{\mathbf{n}}$ appearing in the definition of the generating function are indexed by $\mathbf{n} \in \mathbb{N}_0$, where $\mathbb{N}_0 = \{0, 1, 2, \ldots\}$. However, padding the coefficients by 0 allows us to formally extend summation to all $\mathbf{n} \in \mathbb{Z}$:

$$h(\mathbf{x}) \sim \sum_{\mathbf{n} \in \mathbb{Z}^d} C_{\mathbf{n}} \mathbf{x}^{\mathbf{n}}. \tag{85}$$

It is important for the recurrence to hold for all $\mathbf{n} \in \mathbb{Z}^d$ (otherwise a proper handling of the boundary conditions would be needed).

The differential operator $\sum_{\mathbf{k} \in K} \mathbf{x}^{-\mathbf{k}} P_{\mathbf{k}}(\mathbf{x}\partial_{\mathbf{x}})$ appearing in the statement is understood in the sense

$$\sum_{\mathbf{k} \in K} x_1^{-k_1} \cdots x_d^{-k_d} P_{\mathbf{k}}(x_1 \partial_{x_1}, \ldots, x_d \partial_{x_d}), \tag{86}$$

where $\mathbf{x} = (x_1, \ldots, x_d)$ and $\mathbf{k} = (k_1, \ldots, k_d)$. When applied to a formal power series (85), such a differential operator naturally produces another formal power series with well-defined coefficients. The statement of the theorem is that this series vanishes, i.e. all the coefficients are 0.

*Proof of Theorem 6.1.* For any polynomial $P$

$$P(\mathbf{x}\tfrac{\partial}{\partial \mathbf{x}})h \sim \sum_{\mathbf{n}} C_{\mathbf{n}} P(\mathbf{n}) \mathbf{x}^{\mathbf{n}} = \sum_{\mathbf{n}} C_{\mathbf{n}+\mathbf{k}} P(\mathbf{n}+\mathbf{k}) \mathbf{x}^{\mathbf{n}+\mathbf{k}}. \tag{87}$$

It follows that

$$\sum_{\mathbf{k} \in K} \mathbf{x}^{-\mathbf{k}} P_{\mathbf{k}}(\mathbf{x}\tfrac{\partial}{\partial \mathbf{x}})h \sim \sum_{\mathbf{k} \in K} \sum_{\mathbf{n}} C_{\mathbf{n}+\mathbf{k}} P_{\mathbf{k}}(\mathbf{n}+\mathbf{k}) \mathbf{x}^{\mathbf{n}} \sim 0. \tag{88}$$

$\square$

## G. Free evolutions

**General diagram expansion.** Recall that free evolution is a special regime where the target tensor consist of only zeros, or equivalently when the target is just removed from the loss (3). Consequently, the expected loss evolution in our diagram expansion includes only the diagrams $D_{2\nu}$ and no diagrams $R_\nu$:

$$\mathbb{E}[L(t)] \sim \sum_{s=0}^{\infty} \mathbb{E}[(\tfrac{1}{2}D_{2\nu})^{\star(s+1)}]\frac{(-t)^s}{T^s s!} \sim \sum_{s=0}^{\infty} \mathbb{E}[D_{2\nu}^{\star(s+1)}]\frac{(-t)^s}{2^{s+1}T^s s!}. \tag{89}$$

See Figure 7 (a) for examples of diagrams in $D_{2\nu}^{\star s}$.

### G.1. Underparameterized free evolutions

The large-$p, H$ behavior of the loss function is determined by Pareto-optimal terms described in Theorem 4.1 and corresponding to minimal contractions of diagrams in $D_{2\nu}^{\star s}$. The combinatorics of such minimal contractions is complicated for general free regimes. However, it is substantially simplified in extreme cases, in particular in the underparameterized regime ($H \ll p^{\nu-1}$ in the asymmetric or $H \ll p^{\nu/2}$ in the symmetric even-$\nu$ scenario).

In this regime, by Theorem 4.1, the leading terms in $\mathbb{E}[(\tfrac{1}{2}D_{2\nu})^{\star(s+1)}]$ are given by contracted diagrams that have $n = 1$ $H$-node and $q = \nu + (\nu - 1)s$ $p$-nodes. Such diagrams have the form of "flowers" with $\nu + (\nu - 1)s$ "petals", each consisting of a $p$-node and two edges of matching colors (see Fig. 7 (b)). This contraction has a unique edge pairing.

This picture applies to both symmetric and asymmetric scenario, the difference is only that in the symmetric scenario there is only one color. The arguments in both scenarios will be similar, up to different numerical coefficients due to different color distributions.

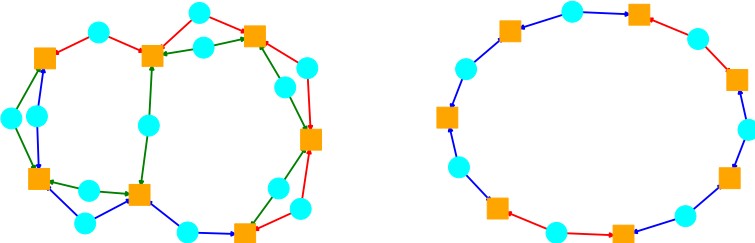

*(a)* Uncontracted diagrams from $D_6^{\star s}$ (**left**) and $D_4^{\star s}$ (**right**).

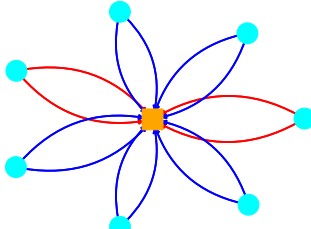

*(b)* An optimally contracted diagram for the free underparameterized regime: a "flower" with one $H$-node and $\nu + (\nu - 1)s$ "petals". Each petal has two edges of a matching color.

*Figure 7.* Diagrams from the free evolutions $D_{2\nu}^{\star s}$.

**Asymmetric scenario.** For convenience, define the generating function $f(x)$ by

$$f(x) = \sum_{s=0}^{\infty} C_s x^s, \tag{90}$$

where the coefficient $C_s$ multiplied by $s!$ is the total number of minimally contracted diagrams (flowers) produced in $\mathbb{E}[D_{2\nu}^{\star(s+1)}]$. By Eq. (89), we can relate the loss to $f$ by

$$\mathbb{E}[L(t)] \sim \sum_{s=0}^{\infty} C_s s! p^{\nu+(\nu-1)s} H \sigma^{2(\nu+(\nu-1)s)} \frac{(-t)^s}{2^{s+1} T^s s!} \tag{91}$$

$$= \frac{p^\nu H \sigma^{2\nu}}{2} f(-p^{\nu-1}\sigma^{2(\nu-1)} t/(2T)). \tag{92}$$

Observe that a contracted diagram in $D_{2\nu}^{\star(s+1)}$ is obtained by merging a contracted diagram in $D_{2\nu}^{\star s}$ with $D_{2\nu}$, and there are $2 \cdot 2(1 + (\nu - 1)s)$ possibilities for that, since $D_{2\nu}^{\star s}$ has $2(1 + (\nu - 1)s)$ edges, and there are 2 edges of matching color in $D_{2\nu}$. It follows that

$$sC_s = 4(1 + (\nu - 1)s)C_{s-1}. \tag{93}$$

This condition also holds for $s = 0$, since both the left- and right-hand sides vanish. Using Theorem 6.1, we then obtain the ODE

$$xf' = 4x(\nu f + x(\nu - 1)f'), \tag{94}$$

or, simplifying,

$$f' = 4(\nu f + x(\nu - 1)f'). \tag{95}$$

Taking into account the initial condition $f(0) = C_0 = 1$, its solution is

$$f(x) = (1 - 4(\nu - 1)x)^{-\frac{\nu}{\nu-1}}. \tag{96}$$

We thus obtain

$$\mathbb{E}[L(t)] \sim \frac{p^\nu H \sigma^{2\nu}}{2}(1 + 2(\nu - 1)p^{\nu-1}\sigma^{2(\nu-1)} t/T)^{-\frac{\nu}{\nu-1}}, \tag{97}$$

as claimed.

**Symmetric even-$\nu$ scenario.** The arguments in this scenario are completely similar, up to replacing the coefficient 4 in Eq. (93) by $4\nu$, since in this case any edge in $D_{2\nu}$ can be used for merging.

## G.2. Overparameterized free evolutions

**Asymmetric scenario.** Consider a free diagram $G$ resulting by merging $D_{2\nu}^{*(s+1)}$. From the description of Pareto-optimal terms, after minimal contraction free overparameterized diagrams have $n = s + 1$ $H$-nodes and $\nu$ $p$-nodes. In particular, only two $H$-nodes are contracted.

By examining the merging process, observe that each $H$-node in $G$ is either *even* or *odd*, in the sense that the associated $\nu$-dimensional edge color parity vector is either $(0, \ldots, 0)$ or $(1, \ldots, 1)$. The parity vector changes every time this node is involved in a merger. Initially, $D_{2\nu}$ has two $H$-nodes. Whenever we merge with $D_{2\nu}$, we create one new odd $H$-node. It follows that the number of odd $H$-nodes always either increases or remains the same. The latter happens iff we merge over an edge of an odd $H$-node. For a pairing, we need all contracted $H$-nodes to be even.

It follows that a diagram $G$ makes a nonvanishing contribution only if it was created by mergers involving only odd $H$-nodes – in this case $G$ has exactly two odd $H$-nodes that can be contracted to ensure all contracted $H$-nodes are even.

It can be easily seen by induction that in such diagrams the edges of a particular color form a non-self-intersecting, non-branching path connecting one odd $H$-node to another. Each $p$-node belongs to one of such paths, so in particular can be assigned a color.

In the optimal contraction the $p$-nodes are contracted to $\nu$ nodes corresponding to different colors. In this case the path of each color forms a flower with center corresponding to the contracted $p$-node and petal ends given by different $H$ nodes. There is a unique valid edge pairing in such a contracted diagram. This gives

$$\mathbb{E}[D_{2\nu}^{\star(s+1)}] \sim (4\nu)^s p^\nu H^{s+1} \sigma^{2(\nu s + \nu - s)} \tag{98}$$

so that

$$\mathbb{E}[L(t)] \sim \frac{p^\nu H \sigma^{2\nu}}{2} e^{-2\nu\sigma^{2(\nu-1)}Ht/T}. \tag{99}$$

**Symmetric even-$\nu$ scenario.** In this scenario all the $H$-vertices are uncontracted ($n = s + 2 = s_D + 1$). The number of contracted $p$-nodes is $q = 1 + (\frac{\nu}{2} - 1)(s + 1)$.

We can enumerate all optimal nonvanishing contracted diagrams and the relevant edge pairings as follows. Consider a diagram $G$ created from a diagram in $D_{2\nu}^{*s}$ by merging with $D_{2\nu}$. The merger creates a new $H$-node, say $k'$. This $H$-node is uncontracted, so all its $\nu$ $p$-neighbors must be contracted to enable an edge pairing. The minimal contraction divides them in pairs; in this case the merger adds exactly $\frac{\nu}{2} - 1$ contracted $p$-nodes. Once the pairing of the new $p$-nodes is chosen, the pairing of new edges is uniquely defined.

Taking into account various possibilities for choosing the merging edges and pairing the new $p$-nodes,

$$\mathbb{E}[D_{2\nu}^{\star(s+1)}] \sim 4\nu(\nu-1)!!(\nu s - s + 1)p^{\nu/2-1}H\sigma^{2(\nu-1)}\mathbb{E}[D_{2\nu}^{\star s}]. \tag{100}$$

Recalling that

$$\mathbb{E}[L(t)] \sim \tfrac{1}{2}\sum_{s=0}^{\infty}\mathbb{E}[D_{2\nu}^{\star(s+1)}]\frac{(-t)^s}{(2T)^s s!}, \tag{101}$$

we get the equation

$$2T\frac{d}{dt}\mathbb{E}[L(t)] = -4\nu(\nu-1)!!p^{\nu/2-1}H\sigma^{2(\nu-1)}\Big(t(\nu-1)\frac{d}{dt}\mathbb{E}[L(t)] + \nu\mathbb{E}[L(t)]\Big). \tag{102}$$

This implies

$$\mathbb{E}[L(t)] \sim \mathbb{E}[L(0)](1 + 2\nu(\nu-1)!!(\nu-1)p^{\nu/2-1}H\sigma^{2(\nu-1)}t/T)^{-\frac{\nu}{\nu-1}} \tag{103}$$

$$= \frac{(\nu-1)!!p^{\nu/2}H^2\sigma^{2\nu}}{2}(1 + 2\nu(\nu-1)!!(\nu-1)p^{\nu/2-1}H\sigma^{2(\nu-1)}t/T)^{-\frac{\nu}{\nu-1}}. \tag{104}$$

### G.3. General free evolutions for SYM with even $\nu \geq 4$

**The recurrence.** For a symmetric, even-$\nu$ scenario, let $C_{ns}$ denote the total number of Pareto-optimal edge pairings in $D_{2\nu}^{\star(s+1)}$ with $n$ contracted $H$-nodes. Recall from Theorem 4.1 that the respective optimal number of $p$-nodes is $q = \nu + (\nu - 1)s - \frac{\nu}{2}(n - 1)$.

**Proposition G.1.** *For even $\nu \geq 4$, the coefficients $C_{ns}$ satisfy the recurrence*

$$C_{ns} = 2\nu(2(\nu - 1)s + 2)[(\nu - 1)!!C_{n-1,s-1} + C_{n,s-1}]. \tag{105}$$

**Remark:** this recurrence **does not hold** at $\nu = 2$! In this case the l.h.s. will generally be larger than the r.h.s. because of additional terms.

*Proof.* We establish a correspondence between Pareto-optimal pairings at $s$ and $s - 1$. Given a diagram $G$, consider a diagram $G_1$ in $G \star D_{2\nu}$ obtained by replacing an edge $u_{ki}$ by the product $u_{k'i} \prod_{r=1}^{\nu-1} u_{ki_r} u_{k'i_r}$. The factor $2\nu(2(\nu - 1)s + 1)$ in Eq. (105) corresponds to the number of diagrams $G_1$ created in this way from $G$ – i.e. the number of ways to choose the edges in $D_{2\nu}$ and $G$ for merging – in the case when the number of edges in $G$ is as in $D_{2\nu}^{\star s}$, i.e. $2(\nu - 1)s + 2$.

Given a pairing in $G$, recall two optimal constructions of pairings for $G_1$:

1. **[increasing $n$]** Divide the edges $(u_{k'i_r})_{r=1}^{\nu-1}$ and $u_{k'i}$ into pairs and contract respective pairs of $p$-nodes $i_1, \ldots, i_{\nu-1}, i$. Assume w.l.o.g. that $i$ is contracted to $i_{\nu-1}$. Form also respective pairs of edges $(u_{ki_r})_{r=1}^{\nu-2}$. Among the newly created edges, we are left with only one unpaired edge $u_{ki_{\nu-1}}$ which is then paired with the edge in $G$ that was previously paired with $u_{ki}$.

   Compared to the contracted $G$, the contracted $G_1$ has 1 additional $H$-node and $\frac{\nu}{2} - 1$ additional $p$-nodes. The number of pairings of $G_1$ generated from one pairing of $G$ is $(\nu - 1)!!$. This gives the first term in Eq. (105).

2. **[retaining $n$]** Contract the $H$-nodes $k$ and $k'$ and pair each $u_{ki_r}$ with $u_{k'i_r}$. The edge $u_{k'i}$ is paired with the edge that was paired with $u_{ki}$ in $G$.

   Compared to the contracted $G$, the contracted $G_1$ has no additional $H$-nodes and $\nu - 1$ additional $p$-nodes. There is only one pairing of $G_1$ generated in this way from one pairing of $G$. This gives the second term in Eq. (105).

We prove now that *all* optimal pairings in $G_1$ are obtained by these two methods if $\nu \geq 4$. Recall from the proof of optimality in Section D.2.2 that any optimal pairing of $G_1$ can be mapped to an optimal pairing of $G$, with three cases:

1. **The new $H$-node $k'$ is not contracted to any other other node in $G_1$.** This case corresponds to the case of increasing $n$ above. The pairing in $G_1$ must necessarily be obtained from a pairing in $G$ as indicated there.

2. **The new $H$-node $k'$ is contracted to $k$.** This case corresponds to the case of retained $n$ above. The pairing in $G_1$ must necessarily be obtained from a pairing in $G$ as indicated there.

3. **The new $H$-node $k'$ is not contracted to $k$, but is contracted to some other node in $G$.** This case does not correspond to either of the two above methods. This option is indeed realized for $\nu = 2$, but we will show that it is not compatible with Pareto-optimality for $\nu \geq 4$.

   Recall that given an edge pairing in $G_1$, we can obtain an edge pairing in $G$ by contracting all the $p$-nodes $i_1, \ldots, i_{\nu-1}, i$ and then restructuring the pairing so as to get the new edges to be paired among themselves, and also the old edges to be paired among themselves. Let the nodes $i_1, \ldots, i_{\nu-1}, i$ form $m$ contracted groups in $G_1$. Denoting the number of contracted $p$-nodes in $G_1$ and $G$ by $q_1$ and $q$, respectively, we have

$$q_1 = q + m - 1. \tag{106}$$

   Since the number of contracted $H$-nodes in $G_1$ and $G$ is the same, for Pareto-optimality of the pairing in $G_1$ we need $q_1 \geq q + \nu - 1$, i.e.

$$m = \nu. \tag{107}$$

   In other words, the Pareto-optimality requires all the $p$-nodes $i_1, \ldots, i_{\nu-1}, i$ to be uncontracted among each other in $G_1$. All the new edges $u_{ki_r}, u_{k'i_r}$ must be contracted to some old edges in $G$. Note, however, that in this case and if

$\nu \geq 4$, we can form a more efficient contraction admitting a pairing in $G$. Indeed, instead of contracting all the $p$-nodes $i_1, \ldots, i_{\nu-1}, i$, contract separately the two groups $\{i_1, i_2\}$ and $\{i_3, \ldots, i_{\nu-1}, i\}$. This gives an additional $p$-degree of freedom in contracted $G$:

$$q = q_1 - \nu + 2; \quad q_1 = q + \nu - 2. \tag{108}$$

At the same time, this contraction admits a valid paring in $G$. Indeed, in the group $\{i_1, i_2\}$, each of the new edges $u_{ki_1}, u_{ki_2}, u_{k'i_1}, u_{k'i_2}$ is paired with some old edge in $G$. We can remove the four new edges and form two pairs among the respective old edges (reflecting the pairs $u_{ki_1}, u_{ki_2}$ and $u_{k'i_1}, u_{k'i_2}$). Similarly, a consistent pairing can be constructed in the group $\{i_3, \ldots, i_{\nu-1}, i\}$.

Thus, the value $q_1$ cannot be optimal, since $q_1 < q + \nu - 1$ for some $q$ corresponding to a valid pairing in $G$ with the same $n$.

Note that the above sub-optimality argument breaks down for $\nu = 2$ since in this case we have just two $p$-nodes $\{i_1, i\}$. In fact, the respective pairings and contractions in $G$ and $G_1$ can generally be optimal in this case.

**Remark:** The above proof reveals a "merger-contraction commutativity" holding for $\nu = 4$: the optimal contractions of the mergers $G \star D_{2\nu}$ are precisely those obtained by merging optimal contractions of $G$ and $D_{2\nu}$. Specifically, the diagram $D_{2\nu}$ has two kinds of optimal contractions, associated with contracting $H$-nodes or $p$-nodes:

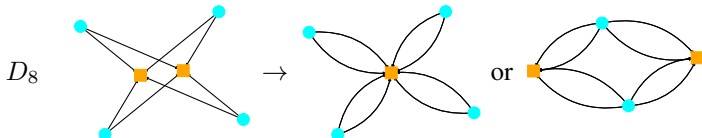

These two kinds correspond to the above "retaining $n$" and "increasing $n$" scenarios, respectively.

$\square$

**Expected loss.**

$$\mathbb{E}[L(t)] \sim \frac{1}{2} \sum_{s=0}^{\infty} \mathbb{E}[D_{2\nu}^{\star(s+1)}] \frac{(-t)^s}{(2T)^s s!} = \frac{1}{2} \sum_{s=0}^{\infty} \sum_{n=1}^{s+2} C_{ns} p^{Q(n,s)} H^n \sigma^{2(\nu-1)s+2\nu} \frac{(-t)^s}{(2T)^s s!} = \tag{109}$$

$$= \frac{p^{3\nu/2} \sigma^{2\nu}}{2} \sum_{s=0}^{\infty} \frac{C_{ns}}{s!} \left( -p^{(\nu-1)} \sigma^{2(\nu-1)} \frac{t}{2T} \right)^s \sum_{n=1}^{s+2} \left( \frac{H}{p^{\nu/2}} \right)^n = \tag{110}$$

$$= \frac{p^{3\nu/2} \sigma^{2\nu}}{2} f\left( -p^{(\nu-1)} \sigma^{2(\nu-1)} \frac{t}{2T}, \frac{H}{p^{\nu/2}} \right), \tag{111}$$

where generating function $f(x, y)$ is defined by

$$f(x, y) = \sum_{s=0}^{\infty} \frac{1}{s!} C_{ns} x^s \sum_{n=1}^{s+2} y^n. \tag{112}$$

Absorbing $s!$ into coefficients and using the proposition we get recurrence

$$s\, C_{ns} = 4\nu\big((\nu-1)s+1\big)\Big((\nu-1)!!\, C_{n-1,s-1} + C_{n,s-1}\Big).$$

Then we can obtain differential equation for $f(x, y)$:

$$\Big[1 - 4\nu(\nu-1)\big(1 + (\nu-1)!!\, y\big) x\Big] \partial_x f(x, y) = 4\nu^2 \big(1 + (\nu-1)!!\, y\big) f(x, y), \tag{113}$$

with the initial condition $f(0, y) = \sum_{n=1}^{2} C_{n,0}\, y^n = y + (\nu-1)!!\, y^2$. This equation can easily be solved because dependency on $y$ appears to be parametric.

Solving (113) yields the closed form

$$f(x, y) = f(0, y) \left( 1 - 4\nu(\nu - 1)\left[ 1 + (\nu - 1)!!\, y \right] x \right)^{-\nu/(\nu-1)}. \tag{114}$$

Substituting $x = -p^{\nu-1}\sigma^{2(\nu-1)}\frac{t}{2T}$ and $y = \frac{H}{p^{\nu/2}}$ then provides the corresponding loss evolution

$$\mathbb{E}[L(t)] \sim \frac{p^\nu H \sigma^{2\nu} \left[ 1 + (\nu-1)!!\frac{H}{p^{\nu/2}} \right]}{2} \left( 1 + \frac{2\nu(\nu-1)\left[ 1 + (\nu-1)!!\frac{H}{p^{\nu/2}} \right] p^{\nu-1}\sigma^{2(\nu-1)}}{T} t \right)^{-\nu/(\nu-1)}. \tag{115}$$

## H. NTK limit

**Definition H.1.** Consider two sequences of random functions from $C^\infty(\mathbb{R}_+)$ defined on the same probability space, $(f_{p,H})_{p,H=1}^\infty$ and $(g_{p,H})_{p,H=1}^\infty$, together with their formal argument expansions:

$$f_{p,H}(t) = \sum_{s=0}^\infty f_{p,H}^{(s)} t^s, \qquad g_{p,H}(t) = \sum_{s=0}^\infty g_{p,H}^{(s)} t^s. \tag{116}$$

We write "$f \cong g$ (as $p, H \to \infty$)" whenever $\mathbb{E}\left[ f_{p,H}^{(s)} \right] \sim \mathbb{E}\left[ g_{p,H}^{(s)} \right]$ as $p, H \to \infty\ \forall s \geq 0$.

### H.1. Model summary statistics and their equivalence

It is not clear how to define a limit model, since the input dimension depends on $p$ which grows to infinity. Nevertheless, we can reason about the following scalar summary statistics of the model:

$$R_\nu(\mathbf{u}) = \sum_{i=1}^p f_{i,\dots,i}(\mathbf{u}), \qquad R_\nu^{(2)}(\mathbf{u}) = \sum_{i=1}^p f_{i,\dots,i}^2(\mathbf{u}), \qquad D_{2\nu}(\mathbf{u}) = \sum_{i_1,\dots,i_\nu=1}^p f_{i_1,\dots,i_\nu}^2(\mathbf{u}). \tag{117}$$

Let us define a linearized model as follows:

$$f_{i_1,\dots,i_\nu}^{lin}(\delta\mathbf{u}) = f_{i_1,\dots,i_\nu}(\mathbf{u}_0) + \nabla^\top f_{i_1,\dots,i_\nu}(\mathbf{u}_0)\delta\mathbf{u}. \tag{118}$$

It also admits similar summary statistics, which we denote $R_\nu^{lin}(\delta\mathbf{u})$, $R_\nu^{(2),lin}(\delta\mathbf{u})$, and $D_{2\nu}^{lin}(\delta\mathbf{u})$.

When $\mathbf{u}$ is learned with gradient flow, we will abuse notation and write $R_\nu(t)$ instead of $R_\nu(\mathbf{u}(t))$, same for $R_\nu^{(2)}$ and $D_{2\nu}$. We use similar notation for $\delta\mathbf{u}$ of the linear model learned with the same gradient flow.

We have the following limit equivalence between the summary statistics of $f$ and $f^{lin}$:

**Proposition H.2.** *Consider $f$ defined in Equation* (1) *with no weight symmetry imposed. Then given $\mathbf{u}_0 = \mathbf{u}(0)$ and $\delta\mathbf{u}(0) = 0$, under the scalings corresponding to Point C and Point B of Figure* 2, $R_\nu \cong R_\nu^{lin}$, $R_\nu^{(2)} \cong R_\nu^{(2),lin}$, *and $D_{2\nu} \cong D_{2\nu}^{lin}$.*

### H.2. Loss equivalence

Recall $L(\mathbf{u}) = \frac{1}{2}D_{2\nu}(\mathbf{u}) - R_\nu(\mathbf{u}) + \frac{1}{2}\|F\|_F^2$. We have a similar decomposition for the loss of the linearized model:

$$L^{lin}(\mathbf{u}) = \frac{1}{2} \sum_{i_1,\dots,i_\nu=1}^p \left( f_{i_1,\dots,i_\nu}^{lin}(\mathbf{u}) - F_{i_1,\dots,i_\nu} \right)^2 = \frac{1}{2}D_{2\nu}^{lin}(\mathbf{u}) - R_\nu^{lin}(\mathbf{u}) + \frac{1}{2}\|F\|_F^2. \tag{119}$$

Therefore Proposition H.2 directly implies $L \cong L^{lin}$.

## H.3. Model symmetries and element-wise equivalence

Since the target diagonal is all-ones, while the model diagonal distribution at initialization is also symmetric, i.e. for any given $p \in \mathbb{N}$, $\{f_{i,\dots,i}(0)\}_{i=1}^p$ are iid, we should necessarily have $pf_{i,\dots,i} \cong R_\nu$ and $pf_{i,\dots,i}^2 \cong R_\nu^{(2)}$ for any fixed $i \in \mathbb{N}$. The same is true for the linearized model. Together with Proposition H.2, this implies $f_{i,\dots,i} \cong f_{i,\dots,i}^{lin}$ and $f_{i,\dots,i}^2 \cong \left(f_{i,\dots,i}^{lin}\right)^2$ for any fixed $i \in \mathbb{N}$.

In fact, the model distribution at initialization is symmetric not only on the diagonal, i.e. for any given $p \in \mathbb{N}$, $\{f_{i_1,\dots,i_\nu}(0)\}_{i_1,\dots,i_\nu=1}^p$ are identically distributed (but not independent). Since all off-diagonal terms of the target are zeros, we get $(p^\nu - p)f_{i_1,\dots,i_\nu}^2 \cong D_{2\nu} - R_\nu^{(2)}$ whenever not all $i_1,\dots,i_\nu$ are equal. Since the same is true for the linearized model, together with Proposition H.2 and the previous paragraph, this implies $f_{i_1,\dots,i_\nu}^2 \cong \left(f_{i_1,\dots,i_\nu}^{lin}\right)^2$ for any fixed $i_1,\dots,i_\nu \in \mathbb{N}$.

## H.4. Kernel method equivalence

Since $f^{lin}$ is linear in weights, it evolves as a kernel method under gradient flow:

$$f_{i_1,\dots,i_\nu}^{lin}(t) = \sum_{i'_1,\dots,i'_\nu=1}^{p} e^{-\frac{t}{T}\Theta_{i_1,\dots,i_\nu;i'_1,\dots,i'_\nu}(0)} \left(f_{i'_1,\dots,i'_\nu}(0) - F_{i'_1,\dots,i'_\nu}\right) + F_{i_1,\dots,i_\nu}, \tag{120}$$

where the exponential is taken from a linear operator on $\mathbb{R}^{p^\nu}$. Here $\Theta$ is the Neural Tangent Kernel (NTK):

$$\Theta_{i_1,\dots,i_\nu;i'_1,\dots,i'_\nu}(\mathbf{u}) = \sum_u \frac{\partial f_{i_1,\dots,i_\nu}(\mathbf{u})}{\partial u} \frac{\partial f_{i'_1,\dots,i'_\nu}(\mathbf{u})}{\partial u} = \sum_{k=1}^{H} \sum_{\tilde m=1}^{\nu} \delta_{i_{\tilde m}=i'_{\tilde m}} \prod_{m\neq\tilde m} u_{k,i_m}^{(m)} u_{k,i'_m}^{(m)}. \tag{121}$$

We also abused the notation above using $\Theta_{i_1,\dots,i_\nu;i'_1,\dots,i'_\nu}(\mathbf{u}(t)) = \Theta_{i_1,\dots,i_\nu;i'_1,\dots,i'_\nu}(t)$.

The Law of Large Numbers results in the following kernel concentration at initialization:

$$\frac{1}{H\sigma^{2\nu-2}}\Theta_{i_1,\dots,i_\nu;i'_1,\dots,i'_\nu}(0) \to \frac{1}{H\sigma^{2\nu-2}}\mathbb{E}\left[\Theta_{i_1,\dots,i_\nu;i'_1,\dots,i'_\nu}(0)\right] = \nu \prod_{m=1}^{\nu} \delta_{i_m=i'_m} \quad \text{a.s. as } H \to \infty. \tag{122}$$

Note also that since $\mathbb{E}\left[f_{i_1,\dots,i_\nu}(0)\right] = 0$, while $\mathbb{E}\left[f_{i_1,\dots,i_\nu}^2(0)\right] = H\sigma^{2\nu}$, we have $f_{i_1,\dots,i_\nu}(0) \to 0$ a.s. as $H \to \infty$ as long as $H\sigma^{2\nu} \to 0$.

Therefore whenever the following conditions hold,

1. $H\sigma^{2\nu} \to 0$ as $H \to \infty$, which is guaranteed for Point B and Point C in the asymmetric scenario;

2. $T \sim \eta^{-1}H\sigma^{2\nu-2}$, which is the relevant scaling for A-B-C in the asymmetric scenario,

the linearized model behaves as a usual linear regression with zero initialization:

$$f_{i_1,\dots,i_\nu}^{lin}(t) \to \left(1 - e^{-\eta t}\right) F_{i_1,\dots,i_\nu} \quad \text{a.s. as } H \to \infty. \tag{123}$$

In the case of $f_{i_1,\dots,i_\nu} \cong f_{i_1,\dots,i_\nu}^{lin}$ and $F_{i_1,\dots,i_\nu} = \delta_{i_1=\dots=i_\nu}$, which we discussed above, this also gives

$$f_{i,\dots,i}(t) \cong 1 - e^{-\eta t}, \qquad f_{i,\dots,i}^2(t) \cong \left(1 - e^{-\eta t}\right)^2, \qquad f_{i_1,\dots,i_\nu}^2(t) \cong 0, \quad \text{where not all } i_1,\dots,i_\nu \text{ are equal.} \tag{124}$$

This implies that the $t$-expansion terms of $f_{i_1,\dots,i_\nu}$ converge to those of $(1 - e^{-\eta t}) \delta_{i_1=\dots=i_\nu}$ in probability. This proves the following result:

**Proposition H.3.** *Consider $f$ defined in Equation (1) in ASYM case. Then for $T \sim \eta^{-1}H\sigma^{2\nu-2}$,*

1. *The initial NTK of $f$ converges almost surely to the identity tensor as $H \to \infty$: $T^{-1}\Theta_{i_1,\dots,i_\nu;i'_1,\dots,i'_\nu}(0) \to \eta\nu \prod_{m=1}^{\nu} \delta_{i_m=i'_m}$.*

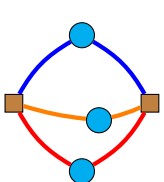 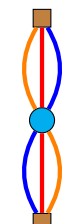 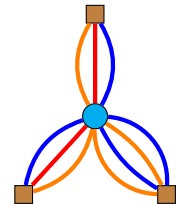 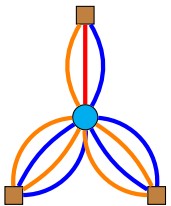 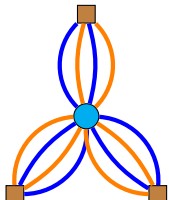

*Figure 8.* Contracted optimal mergers for Point C.

2. *Under the scalings corresponding to Point C and Point B of Figure 2, the $t$-expansion terms of $f_{i_1,\ldots,i_\nu}$ converge to those of $(1 - e^{-\eta t})\,\delta_{i_1=\ldots=i_\nu}$ in probability.*

Unfortunately, this suggests, but does not prove kernel stability:

**Conjecture H.4.** *Under the same setting as for Proposition H.3, $T^{-1}\Theta_{i_1,\ldots,i_\nu;i_1',\ldots,i_\nu'}(t) \to \eta\nu\prod_{m=1}^{\nu}\delta_{i_m=i_m'}$ almost surely as $H \to \infty\ \forall t \geq 0$.*

We now prove Proposition H.2.

### H.5. Proof of Proposition H.2, Point C

Consider first the parameter scaling corresponding to Point C. Let us study mergers corresponding to the evolution of $D_{2\nu}$. The $s$-term in the formal time expansion contains the following mergers: $D_{2\nu}(\star L)^s$. Let us study those mergers that give Pareto-optimal terms after contraction. As follows from Appendix D.2, they are precisely characterized by the following process illustrated in Figure 8.

We contract all the $p$-vertices into one to get a "flower" with two "petals", each consisting of $\nu$ edges, each of a different color, as depicted on the first two diagrams of Figure 8. If there is an odd number of edges for every color in a petal, we call this petal odd. If there are even of them, we call it even. As we shall see soon, all petals we encounter in our construction are either odd or even.

When merging a new diagram, we contract all new $p$ vertices into one to keep the flower structure. Merging a $D$-diagram with a petal alters its parity, while adding a new odd petal, see Figure 8, center. Merging with a $R$-diagram with a petal just alters its parity, see Figure 8, right. In order to get a contractible diagram, we need all petals to be even. Since we start with two odd petals, all mergers have to be performed on odd petals, and there have to be exactly two mergers with $R$-diagrams.

During this process, we never increase the number of odd petals. Therefore, the whole process could be characterized with a pair of color sequences of total length $s$: $((m_1,\ldots,m_k),(m_1',\ldots,m_{k'}'))$, $k + k' = s$, $k, k' \geq 1$. We read these sequences as follows. Let us call one of the initial petals left and the other — right. We merge a $D$-diagram with the left petal with color $m_1$. This petal becomes even, and a new odd petal is created as a substitute; now call it left. We keep merging $D$-diagrams to the left petal with colors specified in the first sequence until we arrive into its last term $m_k$. We then merge the left petal with a $R$-diagram with color $m_k$.

We do the same with the right petal and the second sequence. Since the left and right mergers are independent, we can perform them in any order.

Since we merge all $p$-vertices into one in the limit of Point C, the evolutions of $R_\nu^{(2)}$ and $D_{2\nu}$ coincide. The evolution of $R_\nu$ can be covered as well with a pair of sequences as above. In this case, there is only one starting petal, and we simply take $k' = 0$ to encode the subsequent mergers.

We proceed with demonstrating that we can realize all terms resulted from the above process using $D$- and $R$-diagrams corresponding to the linearized model. Since the set of mergers corresponding to the linearized model is a subset of that corresponding to the full model, the above statement implies equivalence of the full and the linearized models in the limit.

Recall the definition of the linearized model: $f_{i_1,\ldots,i_\nu}^{lin}(\delta\mathbf{u}) = f_{i_1,\ldots,i_\nu}(\mathbf{u}_0) + \nabla^\top f_{i_1,\ldots,i_\nu}(\mathbf{u}_0)\delta\mathbf{u}$, where $\mathbf{u}_0 = \mathbf{u}(0)$, the weight initialization of $f$, and we initialize $\delta\mathbf{u}$ with zeros. For the identity target tensor, the corresponding loss function admits the following decomposition:

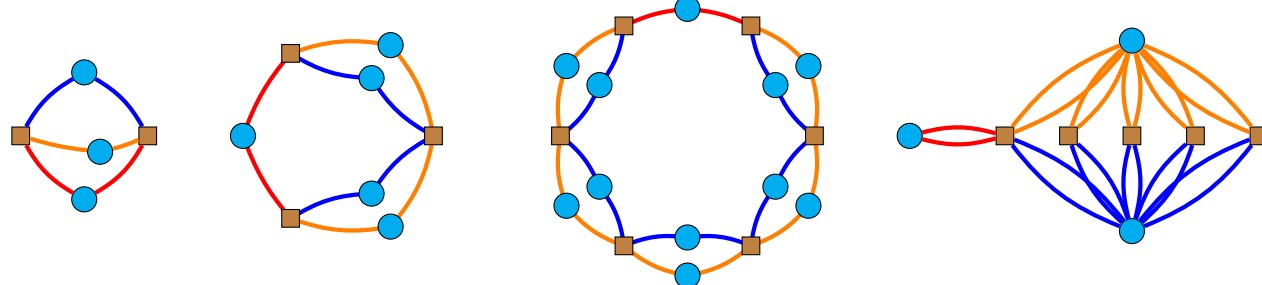

*Figure 9.* Contracting the optimal merger for Point B.

$$L^{lin}(\mathbf{u}) = \frac{1}{2}D_{2\nu}^{lin} - R_{\nu}^{lin} + \frac{p}{2}, \qquad D_{2\nu}^{lin} = \sum_{i_1,\ldots,i_\nu=1}^{p}\left(f_{i_1,\ldots,i_\nu}^{lin}\right)^2, \qquad R_{\nu}^{lin} = \sum_{i=1}^{p} f_{i,\ldots,i}^{lin}. \tag{125}$$

Here, $D_{2\nu}^{lin}$, $R_{\nu}^{lin}$, and $R_{\nu}^{(2),lin}$ are linear combinations of diagrams with edges of $2\nu$ different types, instead of $\nu$ types ("colors") for $f$. Indeed, since $\mathbf{u} = \mathbf{u}(0) + \delta\mathbf{u}$, every edge is decomposed into a non-learnable initialization $\mathbf{u}(0)$, initialized with variance $\sigma^2$, and a learnable increment $\delta\mathbf{u}$, initialized with zero. We will depict learnable increment edges solid and non-learnable initialization edges dashed. We then have $D_{2\nu}^{lin} = \sum_{m,m'=1}^{\nu}\hat{D}_{2\nu}^{m,m'} + 2\sum_{m=1}^{\nu}\hat{D}_{2\nu}^{m} + \hat{D}_{2\nu}$, where $\hat{D}_{2\nu}^{m,m'}$ has the same structure as $D$, but only the edge of color $m$ being solid in the left half and only the edge of color $m'$ being solid in the right half. In $\hat{D}_{2\nu}^{m}$, the same holds for the left part, while the right part is completely dashed. In $\hat{D}_{2\nu}$, all edges are dashed. Similarly, $R_{\nu}^{(2),lin} = \sum_{m,m'=1}^{\nu}\hat{R}_{\nu}^{(2),m,m'} + 2\sum_{m=1}^{\nu}\hat{R}_{\nu}^{(2),m} + \hat{R}_{\nu}^{(2)}$ and $R_{\nu}^{lin} = \sum_{m=1}^{\nu}\hat{R}_{\nu}^{m} + \hat{R}_{\nu}$.

We then implement the sequence of mergers corresponding to $((m_1,\ldots,m_k),(m'_1,\ldots,m'_{k'}))$ with the following sequence of mergers of linearized diagrams. To model the evolution of $D_{2\nu}$, we start with $\hat{D}^{m_1,m'_1}$. The left mergers are modeled with mergers with $\hat{D}^{m_j,m_{j+1}}$ for $j$ running from 1 to $k-1$, culminated with a merger with $\hat{R}^{m_k}$. We do the same with the right part. We follow the same steps to model the evolution of $R_\nu$ and $R_\nu^{(2)}$. This finishes the construction of Pareto-optimal mergers corresponding to Point C.

### H.6. Proof of Proposition H.2, Point B

In this case, all mergers with $R_\nu$ in $L(\star L)^s$ are suboptimal. As proven in Appendix D.2, mergers of $D$-diagrams resulting to Pareto-optimal contractions could be constructed with the following process illustrated in Figure 9. We start by picking a color (red in the figure). We then merge new $D$-diagrams by edges of the chosen color. During this process, we always keep exactly two edges of this color, see the first three pictures of Figure 9. We contract the resulting diagram the following way. The two edges of the chosen color are contracted with each other. The rest are contracted in such a way that exactly $\nu-1$ $p$-vertices and $s+1$ $H$-vertices remain. The remaining $p$ and $H$ vertices form a full bipartite graph: the $m$-th $p$-vertex is connected with each $H$-vertex with a pair of edges of color $m$ for all colors except for the one initially chosen, see Figure 9, right.

All the mergers described above are as well contained in $D_{2\nu}^{lin}\left(\star D_{2\nu}^{lin}\right)^s$. Indeed, let $m$ be the chosen color. Then the above process is simply merging $D_{2\nu}^{m,m}$ together.

## I. Non-stability of the NTK in the SYM matrix model

Consider $f$ defined in Equation (1). Recall the definition of its NTK:

$$\Theta_{i_1,\ldots,i_\nu;i'_1,\ldots,i'_\nu}(\mathbf{u}) = \sum_u \frac{\partial f_{i_1,\ldots,i_\nu}(\mathbf{u})}{\partial u}\frac{\partial f_{i'_1,\ldots,i'_\nu}(\mathbf{u})}{\partial u}. \tag{126}$$

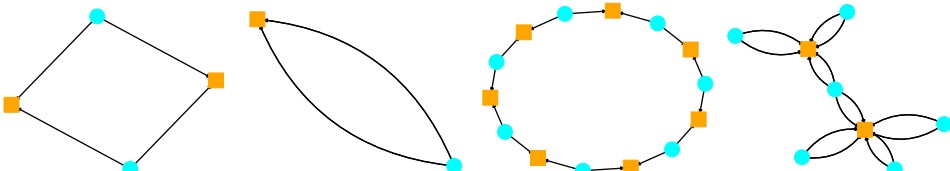

*Figure 10.* Diagrams in the SYM $\nu = 2$ scenario. **Left to right:** $D_4$; $R_2$; a circular diagram from $(\frac{1}{2}D_4 - R_2)^{\star(s+1)}$; a minimal contraction of a circular diagram to a tree.

For the SYM model, the NTK for $\nu = 2$ is given by

$$\Theta_{i,j;i',j'}(\mathbf{u}) = \sum_{k=1}^{H} (\delta_{i=i'} u_{k,j} u_{k,j'} + \delta_{i=j'} u_{k,j} u_{k,i'} + \delta_{j=i'} u_{k,i} u_{k,j'} + \delta_{j=j'} u_{k,i} u_{k,i'}) \tag{127}$$
$$= \delta_{i=i'} f_{j,j'}(\mathbf{u}) + \delta_{i=j'} f_{j,i'}(\mathbf{u}) + \delta_{j=i'} f_{i,j'}(\mathbf{u}) + \delta_{j=j'} f_{i,i'}(\mathbf{u}).$$

That is, for the SYM matrix model, the NTK could be defined in terms of the model itself.

**Proposition I.1.** *For the SYM model with $\nu = 2$, $f_{i,j}(0) = \lim_{t\to\infty} f_{i,j}(t)$ under gradient flow $\forall i, j \in [p]$ is equivalent to $\Theta_{i,j;i',j'}(0) = \lim_{t\to\infty} \Theta_{i,j;i',j'}(t)$ under gradient flow $\forall i, j, i', j' \in [p]$.*

*Proof.* The "$\Rightarrow$" part is trivial. As for the "$\Leftarrow$" part, consider $i, j, j'$ with no two of them being equal. Then $\Theta_{i,j;j,j'}(\mathbf{u}) = f_{i,j'}(\mathbf{u})$. Comparing $t = 0$ and $t \to \infty$ concludes the proof. $\qquad\square$

Therefore, whenever the model eventually learns the target, its NTK cannot be stable.

## J. SYM $\nu = 2$: explicit solution

**Diagram expansion.** In contrast to the asymmetric scenario, in the symmetric case the edges in the diagrams have only one color. Similarly to the symmetric scenario, all uncontracted diagrams occurring in $(\frac{1}{2}D_4 - R_2)^{\star(s+1)}$ are circular, with alternating $p$- and $H$-nodes and $2(s_D + 1)$ edges (see Fig. 10). The absence of different colors simplifies combinatorics. However, in contrast to the free evolution scenarios considered in Section 7 (Appendix G), we have to account for all possible merges $(\frac{1}{2}D_4 - R_2)^{\star(s+1)}$, not just $(\frac{1}{2}D_4)^{\star(s+1)}$.

By Theorem 4.1, a minimal contraction of a circular diagram of length $2(s_D + 1)$ has $n$ $H$-nodes and $s_D + 2 - n$ $p$-nodes, with $n \in \{1, \ldots, s_D + 1\}$ (the flower contractions correspond to the extreme cases $n = 1$ and $n = s_D + 1$). It is easy to see inductively that minimal contractions contract the circle to trees as in Figure 10 (right). (Indeed, since the total number of contracted nodes is $s_D + 2$, at least one node of the circle must be uncontracted. Then its neighboring nodes must be contracted to enable edge pairing. By removing this node and its two edges we then reduce the question to a smaller $s_D$.) Alternatively, one can identify a minimal contraction with a non-crossing partition of the set of $H$-nodes (into contracted groups). The number of such contractions is known to be given by *Narayana numbers* $N_{s_D+1,n}$.

**The general generating function.** This discussion shows that within the approximation by minimal diagrams, we can write

$$\mathbb{E}[(\tfrac{1}{2}D_4 - R_2)^{\star(s+1)}] \sim \sum_{s_D=0}^{s+1} \sum_{n=1}^{s_D+1} M_{s,s_D} N_{s_D+1,n} p^{s_D+2-n} H^n \sigma^{2(s_D+1)}, \tag{128}$$

where $M_{s,s_D}$ is the coefficient in the expansion $(\frac{1}{2}D_4 - R_2)^{\star(s+1)} = \sum_{s_D=0}^{s+1} M_{s,s_D} G_{s_D}$ over circular diagrams $G_{s_D}$ of length $2(s_D + 1)$. Given our standard loss expansion (11), we can then write

$$\mathbb{E}[L(t)] \sim \frac{p}{2} + p^2 \sigma^2 \Psi(-t/T, H/p, p\sigma^2) \tag{129}$$

with the generating function

$$\Psi(x, y, z) = \sum_{s_D=0}^{\infty} z^{s_D} \sum_{s=\max(0,s_D-1)}^{\infty} \frac{M_{s,s_D}}{s!} x^s \sum_{n=1}^{s_D+1} N_{s_D+1,n} y^n. \tag{130}$$

**Reduced generating functions.**   Note that the coefficient $\frac{M_{s,s_D}N_{s_D+1,n}}{s!}$ appearing in the general generating function is partially factorized, in the sense of being a product of the coefficient $\frac{M_{s,s_D}}{s!}$ depending on $s, s_D$, and the Narayana numbers $N_{s_D+1,n}$ depending on $s_D, n$. It is convenient to introduce two respective two-variable generating functions:

$$f(x,z) \sim \sum_{s=0}^{\infty} \sum_{s_D=0}^{s+1} \frac{M_{s,s_D}}{s!} x^s z^{s_D}, \tag{131}$$

$$h(z,y) \sim \sum_{s_D=0}^{\infty} \sum_{n=1}^{s_D+1} N_{s_D+1,n} z^{s_D} y^n. \tag{132}$$

The g.f. $h$ for the Narayana numbers is known:

$$h(z,y) = \frac{1 - z(y+1) - \sqrt{1 - 2z(y+1) + z^2(y-1)^2}}{2z^2}. \tag{133}$$

We will find $f$ using a reduction to first-order PDE.

**The generating function $f$.**   Reducing the case of $s$ to $s-1$ by writing $(\frac{1}{2}D_4 - R_2)^{\star(s+1)} = (\frac{1}{2}D_4 - R_2)^{\star s} \star (\frac{1}{2}D_4 - R_2)$ and considering different possibilities gives the recurrence

$$M_{s,s_D} = -4(s_D + 1)M_{s-1,s_D} + 4s_D M_{s-1,s_D-1}. \tag{134}$$

Taking into account the additional factor $\frac{1}{s!}$ and using Theorem 6.1, we find the PDE

$$x\partial_x f = 4x(-(z\partial_z + 1) + z(z\partial_z + 1))f, \tag{135}$$

i.e.

$$(\partial_x + 4z(1-z)\partial_z)f = 4(z-1)f. \tag{136}$$

We write this as

$$\mathbf{\Phi}^T \nabla f = \phi f, \tag{137}$$

where

$$\mathbf{\Phi} = \begin{pmatrix} 1 \\ 4z(1-z) \end{pmatrix}, \tag{138}$$

$$\phi = 4(z-1). \tag{139}$$

The solution along an integral curve $\mathbf{x}(\tau)$ is

$$f(\mathbf{x}_0) = f(\mathbf{x}_1)e^{-\int_{\tau_0}^{\tau_1} \phi(\mathbf{x}(\tau))d\tau}. \tag{140}$$

The integral curves are given by

$$\frac{dz}{dx} = 4z(1-z), \tag{141}$$

i.e.

$$4dx = d\ln\frac{z}{1-z}. \tag{142}$$

We get a first integral

$$I = \frac{1-z}{z}e^{4x}. \tag{143}$$

We also note that

$$d\ln z = 4(1-z) = -\phi(z), \tag{144}$$

so that

$$e^{-\int_{\tau_0}^{\tau_1} \phi(\mathbf{x}(\tau))d\tau} = \frac{z_1}{z_0}. \tag{145}$$

Accordingly, along an integral curve

$$f(x_0, z_0) = f(x_1, z_1)\frac{z_1}{z_0}. \tag{146}$$

In particular, we can set $x_1 = 0$, then

$$\frac{1 - z_0}{z_0}e^{4x_0} = \frac{1 - z_1}{z_1}, \tag{147}$$

implying

$$z_1 = \frac{1}{\frac{1 - z_0}{z_0}e^{4x_0} + 1} \tag{148}$$

Then, the general solution can be written as

$$f(x_0, z_0) = f(0, z_1)\frac{z_1}{z_0} = f\left(0, \frac{1}{\frac{1 - z_0}{z_0}e^{4x_0} + 1}\right)\frac{1}{(1 - z_0)e^{4x_0} + z_0}. \tag{149}$$

In our setting

$$f(0, z) = M_{0,0} + M_{0,1}z = \frac{z}{2} - 1. \tag{150}$$

It follows that

$$f(x, z) = f(0, z_1)\frac{z_1}{z_0} \tag{151}$$

$$= \frac{2(z - 1)e^{4x} - z}{2(z - (z - 1)e^{4x})^2}. \tag{152}$$

**Merging the two g.f.'s.**   In general, given two sequences $a_n, b_n$ with g.f.'s $A(z), B(z)$ :

$$A(z) \sim \sum_{n=0}^{\infty} a_n z^n, \tag{153}$$

$$B(z) \sim \sum_{n=0}^{\infty} b_n z^n, \tag{154}$$

the g.f. $C(z)$ for the product sequence $a_n b_n$,

$$C(z) = \sum_{n=0}^{\infty} a_n b_n z^n, \tag{155}$$

can be found by

$$C(z) = \frac{1}{2\pi i}\oint_{\gamma} A(\zeta)B\left(\frac{z}{\zeta}\right)\frac{d\zeta}{\zeta} \tag{156}$$

with a suitable contour so that the arguments of $A, B$ lie in the convergence discs.

It follows that we can find the full 3-variate g.f. $\Psi$ by

$$\Psi(x, y, z) = \frac{1}{2\pi i}\oint_{\gamma} h(\zeta, y)f\left(x, \frac{z}{\zeta}\right)\frac{d\zeta}{\zeta}. \tag{157}$$

We have

$$f\left(x, \frac{z}{\zeta}\right) = \frac{\zeta(z(1 - e^{-4x}/2) - \zeta)e^{-4x}}{(\zeta - z(1 - e^{-4x}))^2}, \tag{158}$$

so

$$\Psi(x, y, z) = \frac{1}{2\pi i}\oint_{\gamma} h(\zeta, y)\frac{(z(1 - e^{-4x}/2) - \zeta)e^{-4x}}{(\zeta - z(1 - e^{-4x}))^2}d\zeta. \tag{159}$$

As the contour $\gamma$, we can take a circle $|\zeta| = \epsilon$ with some small $\epsilon$; then the above integral representation for $\Psi$ is valid for all sufficiently small $z$.

The integral is computed by calculating the residue at the pole $\zeta = z(1 - e^{-4x})$ :

$$\Psi(x, y, z) = \partial_\zeta \Big( h(\zeta, y)(z(1 - e^{-4x}/2) - \zeta)e^{-4x} \Big)\Big|_{\zeta = z(1-e^{-4x})} \tag{160}$$

$$= \Big( \frac{ze^{-4x}}{2} \partial_1 h(z(1 - e^{-4x}), y) - h(z(1 - e^{-4x}), y) \Big)e^{-4x}. \tag{161}$$

## K. SYM even-$\nu \geq 4$

**General observations.** Let the values $\mathbf{r} = (r_2, r_4, \ldots)$ denote the multiplicities of the edges in Pareto-optimal contractions of diagrams in $(\frac{1}{2}D_{2\nu} - R_\nu)^{\star(s+1)}$. The number of Pareto-optimal pairings consistent with a particular contracted diagram is defined as

$$N_{\mathbf{r}} = \begin{cases} 0, & \exists \text{ odd } m : r_m > 0, \\ \prod_{m=1}^\infty ((2m-1)!!)^{r_{2m}}, & \text{otherwise.} \end{cases} \tag{162}$$

The key observation is that for even $\nu \geq 4$ all Pareto-optimal contractions of the merged diagrams $(\frac{1}{2}D_{2\nu} - R_\nu)^{\star(s+1)}$ can be obtained by merging respective Pareto-optimal contractions of $D_{2\nu}, R_\nu$ ("commutativity of mergers and contractions"). In the underparameterized limit, the diagrams $D_{2\nu}$ are only contracted to "flowers with $\nu$ petals", while in the general case they can also be contracted to pairs of $H$-nodes connected through $\nu/2$ $p$-nodes by double edges. There are $(\nu - 1)!!$ such diagrams. Define the values $M_{s,n,\mathbf{r}}$ as the total coefficient of contracted diagrams in $(\frac{1}{2}D_{2\nu} - R_\nu)^{\star(s+1)}$ with edge multiplicities $\mathbf{r}$ and $n$ $H$-nodes.

**The extended generating function.** The powers $n$ and $q$ in Pareto-optimal terms $p^q H^n \sigma^{2l}$ in $(\frac{1}{2}D_{2\nu} - R_\nu)^{\star(s+1)}$ are connected by the identity

$$\frac{\nu}{2}(n - 1) + q = \sum_m r_m. \tag{163}$$

Using this identity, we get

$$L(t) \sim \frac{p}{2} + \sum_{s=0}^\infty \frac{\mathbb{E}[(\frac{1}{2}D_{2\nu} - R_\nu)^{\star(s+1)}]}{s!}(-t/T)^s \tag{164}$$

$$= \frac{p}{2} + \sum_{s=0}^\infty \sum_n \sum_{\mathbf{r}} \frac{M_{s,n,\mathbf{r}} N_{\mathbf{r}}}{s!}(-t/T)^s p^q H^n \sigma^{\sum_{m=2}^\infty m r_m} \tag{165}$$

$$= \frac{p}{2} + \sum_{s=0}^\infty \sum_{n=1}^\infty \sum_{\mathbf{r}} \frac{M_{s,n,\mathbf{r}} N_{\mathbf{r}}}{s!}(-t/T)^s p^{\sum_{m=2}^\infty r_m - \frac{\nu}{2}(n-1)} H^n \sigma^{\sum_{m=2}^\infty m r_m} \tag{166}$$

$$= \frac{p}{2} + p^{\frac{\nu}{2}} \sum_{s=0}^\infty \sum_{n=1}^\infty \sum_{\mathbf{r}} \frac{M_{s,n,\mathbf{r}} N_{\mathbf{r}}}{s!}(-t/T)^s (H/p^{\frac{\nu}{2}})^n \prod_{m=2}^\infty (p\sigma^m)^{r_m} \tag{167}$$

$$= \frac{p}{2} + p^{\nu/2} f(-t/T, H/p^{\nu/2}, p\sigma^2, p\sigma^3, \ldots), \tag{168}$$

where

$$f(x, y, \mathbf{z}) = \sum_{s=0}^\infty \sum_{n=1}^\infty \sum_{\mathbf{r}} \frac{M_{s,n,\mathbf{r}} N_{\mathbf{r}}}{s!} x^s y^n \mathbf{z}^{\mathbf{r}}, \tag{169}$$

where $\mathbf{z} = (z_2, z_3, \ldots)$. Instead of $f$, we can consider the simpler generating function

$$g(x, y, \mathbf{z}) = \sum_{s=0}^\infty \sum_{n=1}^\infty \sum_{\mathbf{r}} \frac{M_{s,n,\mathbf{r}}}{s!} x^s y^n \mathbf{z}^{\mathbf{r}}. \tag{170}$$

We have

$$f(x, y, z_2, z_3, \ldots, z_{2n-1}, z_{2n}, \ldots) = g(x, y, z_2, 0, \ldots, 0, (2n-1)!! z_{2n}, \ldots) \tag{171}$$

and

$$L(t) \sim \frac{p}{2} + p^{\nu/2} g(-t/T, H/p^{\nu/2}, \mathbf{z}_{p,\sigma}), \tag{172}$$

$$\mathbf{z}_{p,\sigma} = (p\sigma^2, 0, 3p\sigma^4, 0, \ldots, 0, (2n-1)!! p\sigma^{2n}, \ldots). \tag{173}$$

**The recurrence and PDE.** Denoting $C_{s,n,\mathbf{r}} = M_{s,n,\mathbf{r}}/s!$ and $\mathbf{e}_m = (0, \ldots, 0, \underset{r_m}{1}, 0, \ldots)$, we get the recurrence

$$sC_{s,n,\mathbf{r}} = \nu\Big(2(r_2 - (\nu - 1)) + \sum_{m=3}^{\infty} mr_m\Big)\Big(C_{s-1,n,\mathbf{r}-(\nu-1)\mathbf{e}_2} + (\nu - 1)!!C_{s-1,n-1,\mathbf{r}-(\nu-1)\mathbf{e}_2}\Big) \tag{174}$$

$$- \nu \sum_{m=\nu}^{\infty} (m - \nu + 2)(r_{m-(\nu-2)} + 1)C_{s-1,n,\mathbf{r}-\mathbf{e}_m+\mathbf{e}_{m-(\nu-2)}}. \tag{175}$$

This gives the equation

$$\Big(x\partial_x - \nu x(1 + (\nu - 1)!!y)z_2^{\nu-1} \sum_{m=2}^{\infty} mz_m\partial_{z_m} + \nu \sum_{m=\nu}^{\infty} (m - \nu + 2)x\frac{z_m}{z_{m-\nu+2}}z_{m-\nu+2}\partial_{z_{m-\nu+2}}\Big)g = 0, \tag{176}$$

i.e.

$$\Big(\partial_x + \nu \sum_{m=2}^{\infty} m(z_{m+\nu-2} - \theta z_2^{\nu-1}z_m)\partial_{z_m}\Big)g = 0 \tag{177}$$

with

$$\theta = 1 + (\nu - 1)!!y. \tag{178}$$

We write it as

$$\nabla_F g = 0 \tag{179}$$

with the vector field

$$F(x, y, \mathbf{z}) = \begin{pmatrix} F_x \\ F_y \\ F_{z_2} \\ F_{z_3} \\ \cdots \\ F_{z_m} \\ \cdots \end{pmatrix} = \begin{pmatrix} 1 \\ 0 \\ 2\nu(z_\nu - \theta z_2^\nu) \\ 3\nu(z_{\nu+1} - \theta z_2^{\nu-1}z_3) \\ \cdots \\ m\nu(z_{m+\nu-2} - \theta z_2^{\nu-1}z_m) \\ \cdots \end{pmatrix} \tag{180}$$

Thus, $y$ appears in the dynamics only as a constant parameter.

The initial condition is

$$g(0, y, \mathbf{z}) = \frac{1}{2}(y + (\nu - 1)!!y^2)z_2^\nu - yz_\nu = y\Big(\frac{1}{2}\theta z_2^2 - z_\nu\Big). \tag{181}$$

Consider the integral curves

$$\frac{d}{d\tau}(x, y, \mathbf{z})^T = F(x, y, \mathbf{z}). \tag{182}$$

Given a point $(x_0, \mathbf{z}_0)$, we find the PDE solution at this point as

$$g(x_0, y, \mathbf{z}_0) = g(0, y, \mathbf{z}(\tau_*)) = y\Big(\frac{\theta z_2^\nu(\tau_*)}{2} - z_\nu(\tau_*)\Big), \tag{183}$$

where

$$x(0) = x_0, \quad \mathbf{z}(0) = \mathbf{z}_0, \quad x(\tau_*) = 0. \tag{184}$$

Thanks to the equation $\dot{x} = 1$, we have $\tau_* = -x_0$. Also, since $g(0, y, \mathbf{z}) = y(\frac{\theta z_2^\nu}{2} - z_\nu)$, we only need to find the values $z_2(\tau_*)$ and $z_\nu(\tau_*)$ on the integral curve.

**Solution.** The equation for $z_m$ involves $z_k$ with $k = m + \nu - 2 > m$, so we need to simultaneously solve all the equations for $\dot{z}_{2+i(\nu-2)}$, $i = 0, 1, 2, \ldots$. The equation $\dot{z}_m = m\nu(z_{m+\nu-2} - \theta z_2^{\nu-1}z_m)$ can be written as

$$\frac{d}{d\tau}\Big(e^{m\nu\theta \int z_2^{\nu-1}}z_m\Big) = m\nu e^{m\nu\theta \int z_2^{\nu-1}}z_{m+\nu-2} \tag{185}$$

so that

$$z_m = e^{-m\nu\theta \int z_2^{\nu-1}} \left( c_m + m\nu \int e^{m\nu\theta \int z_2^{\nu-1}} z_{m+\nu-2} \right) \tag{186}$$

$$= e^{-m\nu\theta \int z_2^{\nu-1}} \left( c_m + m\nu \int e^{(2-\nu)\nu\theta \int z_2^{\nu-1}} \left( c_{m+\nu-2} + (m+\nu-2)\nu \int e^{(m+\nu-2)\nu\theta \int z_2^{\nu-1}} z_{m+2(\nu-2)} \right) \right) \tag{187}$$

$$= \ldots \tag{188}$$

$$\sim \phi^{\frac{m}{\nu-2}} \sum_{k=0}^{\infty} c_{m+k(\nu-2)} \prod_{l=0}^{k-1} (m + l(\nu-2)) \nu^k \underbrace{\int \phi \int \phi \ldots \int}_{k} \phi, \tag{189}$$

where

$$\phi = e^{(2-\nu)\nu\theta \int z_2^{\nu-1}}. \tag{190}$$

Let us agree from now on that in all the indefinite integrals the lower integration limit is $\tau = 0$:

$$\int \equiv \int_0^{\tau}. \tag{191}$$

Then $\phi(\tau = 0) = 1$, and also the constants $c_n$ are equal to our particular initial conditions:

$$c_m = z_m(\tau = 0) = \begin{cases} (m-1)!!p\sigma^n, & \text{even } m \\ 0, & \text{odd } m. \end{cases} \tag{192}$$

### K.1. The case $\nu = 4$

**Integral representation of $z_2$.** In this case we conveniently have

$$c_{2+k(\nu-2)} \prod_{l=0}^{k-1} (2 + l(\nu-2)) = (2k+1)!!(2k)!!p\sigma^{2+2k} = (2k+1)!p\sigma^{2+2k}. \tag{193}$$

As a result,

$$z_2 \sim p\sigma^2 \phi \sum_{k=0}^{\infty} (2k+1)!(2\sigma)^{2k} \underbrace{\int \phi \int \phi \ldots \int}_{k} \phi. \tag{194}$$

Using the Euler-Borel substitution $(2k+1)! = \int_0^{\infty} u^{2k+1} e^{-u} du$, we get

$$z_2 \sim p\sigma^2 \phi \int_0^{\infty} \left( \sum_{k=0}^{\infty} (2\sigma u)^{2k} \underbrace{\int \phi \int \phi \ldots \int}_{k} \phi \right) e^{-u} u \, du \tag{195}$$

$$= p\sigma^2 \phi \int_0^{\infty} e^{(2\sigma u)^2 (\int \phi) - u} u \, du. \tag{196}$$

The last integral requires that

$$\psi = \int_0^{\tau} \phi < 0. \tag{197}$$

Since $\phi > 0$ always, this means $\tau < 0$.

**Integral representation of $z_m$.** We can write a similar representation for general $z_m = 4, 6, \ldots$ (for odd $m$ the value $z_m \equiv 0$). We write

$$c_{m+k(\nu-2)} \prod_{l=0}^{k-1} (m + l(\nu-2)) = p\sigma^{m+2k}(m+2k-1)!! \prod_{l=0}^{k-1} (m+2l) = \frac{(m+2k-1)!}{(m-2)!!} p\sigma^{m+2k}. \tag{198}$$

Then

$$z_m \sim p\sigma^m \phi^{\frac{m}{2}} \sum_{k=0}^{\infty} \frac{(m+2k-1)!}{(m-2)!!} (2\sigma)^{2k} \underbrace{\int \phi \int \phi \dots \int}_{k} \phi \tag{199}$$

$$= \frac{p\sigma^m \phi^{\frac{m}{2}}}{(m-2)!!} \int_0^{\infty} \Big( \sum_{k=0}^{\infty} (2\sigma u)^{2k} \underbrace{\int \phi \int \phi \dots \int}_{k} \phi \Big) e^{-u} u^{m-1} du \tag{200}$$

$$= \frac{p\sigma^m \phi^{\frac{m}{2}}}{(m-2)!!} \int_0^{\infty} e^{(2\sigma u)^2(\int \phi)-u} u^{m-1} du. \tag{201}$$

Recall that we introduced $\phi$ by the condition

$$\phi = e^{-8\theta \int z_2^3}. \tag{202}$$

We can check that under this condition the derived formulas indeed provide a solution of characteristic ODEs at $x \geq 0$:

$$\dot{z}_m = \frac{m}{2\phi}(-8\theta z_2^3)\phi z_m + \frac{p\sigma^m \phi^{\frac{m}{2}}}{(m-2)!!} \int_0^{\infty} (2\sigma u)^2 \phi e^{(2\sigma u)^2(\int \phi)-u} u^{m-1} du \tag{203}$$

$$= 4m(z_{m+2} - \theta z_2^3 z_m). \tag{204}$$

The initial condition for $z_m$ is also straightforward.

**Integral representation of the generating function.** The solution of the PDE at $x \geq 0$ and $\mathbf{z} = \mathbf{z}_{p,\sigma}$ is then given by

$$g(x, y, \mathbf{z}_{p,\sigma}) = y \Big( \frac{\theta z_2^4(-x)}{2} - z_4(-x) \Big) \tag{205}$$

$$= \frac{\theta y}{2} \Big( p\sigma^2 \phi(-x) \int_0^{\infty} e^{(2\sigma u)^2(\int_0^{-x} \phi)-u} u \, du \Big)^4 - \frac{yp\sigma^4 \phi^2(-x)}{2} \int_0^{\infty} e^{(2\sigma u)^2(\int_0^{-x} \phi)-u} u^3 \, du. \tag{206}$$

**Finding $\phi$.** Let us denote

$$F(a) = \int_0^{\infty} e^{4au^2-u} u \, du, \tag{207}$$

so that, in terms of $\psi = \int \phi$,

$$z_2 = p\sigma^2 \dot{\psi} F(\sigma^2 \psi). \tag{208}$$

The function $F$ can be expressed in terms of the complementary error function $\text{erfc}(z) = \frac{2}{\sqrt{\pi}} \int_z^{\infty} e^{-t^2} dt$:

$$F(x) = -\frac{1}{8x} + \frac{1}{32x} \sqrt{\frac{\pi}{-x}} e^{-\frac{1}{16x}} \text{erfc} \Big( \frac{1}{4\sqrt{-x}} \Big). \tag{209}$$

Recall that we introduced $\phi$ as:

$$\phi = e^{-8\theta \int z_2^3}. \tag{210}$$

We can then write a second-order ODE on $\psi$:

$$\ddot{\psi} = -8\theta z_2^3 \dot{\psi}, \tag{211}$$

$$z_2 = p\sigma^2 \dot{\psi} F(\sigma^2 \psi), \tag{212}$$

$$\psi(x_0) = 0, \quad \dot{\psi}(x_0) = 1. \tag{213}$$

We write this ODE as

$$\frac{d\dot{\psi}}{d\psi} = -8\theta p^3 \sigma^6 \dot{\psi}^3 F^3(\sigma^2 \psi) \tag{214}$$

and separate the variables:

$$\frac{d\dot{\psi}}{\dot{\psi}^3} = -8\theta p^3 \sigma^6 F^3(\sigma^2 \psi) d\psi. \tag{215}$$

This gives the first integral

$$I = \frac{1}{\dot{\psi}^2} - 16\theta p^3 \sigma^6 \int F^3(\sigma^2\psi)d\psi = \frac{1}{\dot{\psi}^2} - 16\theta p^3 \sigma^4\Big(\int F^3\Big)(\sigma^2\psi). \tag{216}$$

Using the initial condition, we find that $I = 1$. Then,

$$\frac{d\psi}{d\tau} = \pm\Big(1 + 16\theta p^3 \sigma^4\Big(\int F^3\Big)(\sigma^2\psi)\Big)^{-1/2} \tag{217}$$

and hence $\psi$ is implicitly given by

$$\pm\int\sqrt{1 + 16\theta p^3 \sigma^4\Big(\int F^3\Big)(\sigma^2\psi)}d\psi = \tau. \tag{218}$$

**Two convergence regimes.** Note that $F > 0$ while $\tau \leq 0$ and $\psi \leq 0$ in our setting, so also $\int F^3 \leq 0$. Therefore, the expression under the root may become negative at sufficiently large negative $\psi$. We have $F(a) \asymp a^{-1}$ as $a \to -\infty$, so $\int_0^{-\infty} F^3$ is finite. Then, the condition for the expression under the root to stay positive at all $\psi$ is

$$p^3\sigma^4 < \frac{1}{-16\theta\int_0^{-\infty} F^3}. \tag{219}$$

Thus, we have two regimes:

1. **[Low-noise]** If this condition holds, then $\psi(\tau)$ is defined for all $\tau \leq 0$. As $\tau \to -\infty$, $\dot{\psi}(\tau)$ converges to a positive value while $\psi(\tau)$ becomes approximately linear. In the solution (206) for the generating function $g$, the second term dominates as $\tau \to -\infty$:

$$g(-\tau, \mathbf{z}_{p,y,\sigma}) \asymp -\frac{yp\sigma^4\dot{\psi}^2}{\sigma^4\psi^2} \asymp -\frac{yp}{\tau^2}. \tag{220}$$

   This is consistent with $g(0, y, \mathbf{z}_{p,\sigma}) = y\theta\frac{p^4\sigma^8}{2} - 6yp\sigma^4$ being negative at sufficiently small $p^3\sigma^4$.

2. **[High-noise]** If the condition is violated, then $\dot{\psi}$ blows up at a finite $\tau = \tau_{crit} < 0$. This corresponds to a finite $\psi = \psi_{crit}$. We have

$$p^{3/2}\sigma^3|\psi - \psi_{crit}|^{3/2} \asymp |\tau - \tau_{crit}|, \tag{221}$$

$$\dot{\psi} \asymp p^{-3/2}\sigma^{-3}|\psi - \psi_{crit}|^{-1/2} = p^{-1}\sigma^{-2}|\tau - \tau_{crit}|^{-1/3}. \tag{222}$$

   It follows that the first term dominates in $g(-\tau, y, \mathbf{z}_{p,\sigma})$ and

$$g(-\tau, y, \mathbf{z}_{p,\sigma}) \asymp (p\sigma^2\dot{\psi})^4 = |\tau - \tau_{crit}|^{-4/3}. \tag{223}$$

   This agrees with the divergence of the free model at negative times, and has the right power $-\frac{4}{3}$ as in the free model:

$$\mathbb{E}[L(t)] = \mathbb{E}[L(0)](1 + ct)^{-\frac{\nu}{\nu-1}}. \tag{224}$$

## L. Experiments

**Reproducibility.** Our jupyter notebooks used to reproduce the experiments are available at https://github.com/Yarikyaroslav/GFthroughDiagrams_ICML2026.

**Finite difference integration of gradient flow.** In order to compare the theoretical continuous gradient flow (4) with numerical experiments, we discretize the dynamics by means of an explicit Euler scheme. Concretely, we fix the maximal integration time $t_{\max}$ and the number of steps $N_{\text{steps}}$, which determines the integration step

$$\tau = \frac{t_{\max}}{N_{\text{steps}}}.$$

The continuous flow

$$\frac{du}{dt} = -\frac{1}{T}\,\partial_u L(\mathbf{u})$$

is then approximated by the finite-difference update rule

$$u^{(k+1)} = u^{(k)} - \frac{\tau}{T}\,\partial_u L(u^{(k)}), \qquad k = 0, 1, \ldots, N_{\text{steps}} - 1.$$

Thus the effective learning rate of the numerical scheme is $\eta = \tau/T$, depending jointly on the physical scale $T$ and the discretization step $\tau$.

We emphasize that the explicit Euler discretization above is precisely the standard gradient descent iteration with step size $\eta = \tau/T$. Classical results in numerical analysis show that, under mild smoothness assumptions, the Euler scheme converges to the solution of the underlying gradient–flow ODE with a global error of order $O(\eta)$. Consequently, for sufficiently small $\eta$, the discrete dynamics remain uniformly close to the continuous gradient flow; conversely, gradient flow provides an accurate infinitesimal description of the behavior of discrete gradient descent.

**Memory efficient loss computation.** The evaluation of the quadratic loss (3) requires summation over all $p^\nu$ index tuples $(i_1, \ldots, i_\nu)$, which becomes prohibitive to store in memory when both $p$ and $\nu$ are large. To overcome this difficulty, we employ a batching procedure at the level of loss computation. Specifically, we partition the full index set

$$\{1, \ldots, p\}^\nu = \bigcup_{b=1}^{B} \mathcal{B}_b,$$

where each $\mathcal{B}_b$ is a batch of multi-indices. For a given batch $\mathcal{B}_b$ we compute the partial loss

$$L_{\mathcal{B}_b}(\mathbf{u}) = \frac{1}{2} \sum_{(i_1, \ldots, i_\nu) \in \mathcal{B}_b} \left( f_{i_1, \ldots, i_\nu} - F_{i_1, \ldots, i_\nu} \right)^2, \tag{225}$$

and accumulate its gradient contribution. Iterating over all batches and summing their contributions yields exactly the full loss (3) and its gradient:

$$L(\mathbf{u}) = \sum_{b=1}^{B} L_{\mathcal{B}_b}(\mathbf{u}).$$

In this way the computation can be performed in a memory-efficient manner, since only one batch is loaded and processed at a time. Importantly, unlike stochastic gradient descent, this batching procedure does not approximate the loss but reproduces it exactly after all batches have been processed, so the optimization dynamics still correspond to the full gradient descent with loss (3).

## L.1. Free SYM $\nu = 4$

Here we discuss the experimental part of our work that we performed for the free evolution in SYM $\nu = 4$ problem. For both under- and overparameterized scenarios the expected loss has the following limiting shape (see Appendix G):

$$\mathbb{E}\left[L(t)\right] = (1 + ct)^{-4/3}, \tag{226}$$

where

$$c = \begin{cases} \frac{24p^3\sigma^6}{T}, & \text{underparameterized}, \\ \frac{72pH\sigma^6}{T}, & \text{overparameterized}, \end{cases} \tag{227}$$

and the initial loss expectation $\mathbb{E}\left[L(0)\right]$ was normalized to 1 by the appropriate scaling of $\sigma$. The parameter $c$ was set to a constant (in particular, we set $c = 1/4$) by the appropriate scaling of $T$. This way, starting from some sufficiently large $p$ we expected all experimental loss curves (regardless of the *actual* values of parameters $p, H, \sigma, T$) to converge to the theoretical formula and have a similar shape.

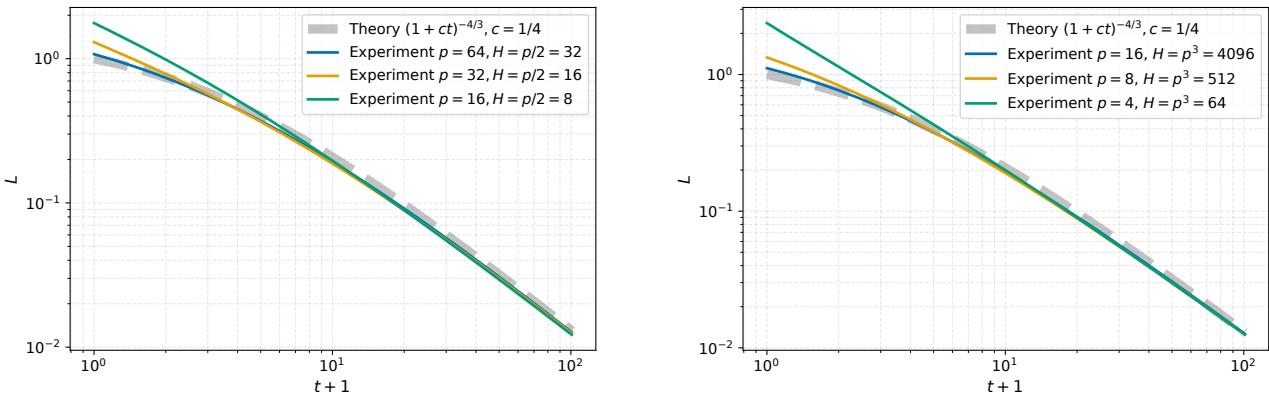

*Figure 11.* Free SYM $\nu = 4$. **Left:** underparameterized scenario. **Right:** overparameterized scenario.

**Results.** The experimental results confirmed these theoretical predictions: for sufficiently large $p$ experimental curves are very close to the theoretically predicted curve (see Figure 11). We also emphasize an interesting and counterintuitive observation. For a relatively small $p$ we can see a difference between experiment and theory at small $t$, but for large $t$ we see a convergence of experimental curves to the theoretical curve, although our theory uses an expansion of the loss at $t = 0$ (see (6)).

## L.2. SYM $\nu = 2$

Let us recall the expression of the loss expectation for this case (see Appendix J) and divide it by $p/2$:

$$\frac{\mathbb{E}[L(t)]}{p/2} \sim 1 + 2p\sigma^2\Psi(-t/T, H/p, p\sigma^2). \tag{228}$$

If we then set our parameters, in such a way that $T, p\sigma^2, H/p$ are some constants, then the RHS is independent of the *actual* values of $p, H, \sigma$, so we can similarly study how close the experimental lines become to the theory as we increase $p, H$. We considered four scenarios, each of them with $T = 1$, but different $p\sigma^2$ and $H/p$.

1. **Noisy underparameterized.** We set $H/p = 1/4$ and $p\sigma^2 = 16$. In this case the initial loss is significantly larger than target loss $p/2$ and the model has relatively small number of parameters.

2. **Noisy overparameterized.** We set $H/p = 8$ and $p\sigma^2 = 10$. Then the initial loss is significantly larger than target loss $p/2$ and the model has large number of parameters.

3. **Low-noise underparameterized.** We set $H/p = 1/8$ and $p\sigma^2 = 10^{-2}$. Then the initial loss equals the target loss $p/2$ and the model has small number of parameters, so that the final loss remains close to the initial without approaching zero.

4. **General scenario.** We set $H/p = 1$ and $p\sigma^2 = 1$. The model has just enough parameters to fit the target, the loss approaches zero.

**Results.** As shown in Figure 12, experimental results confirmed the theoretical predictions in all considered scenarios. Surprisingly, in the noisy overparameterized case our explicit theoretical solution coincides with the experiment even for the smallest possible $p = 1$, although the theory assumes large $p$.

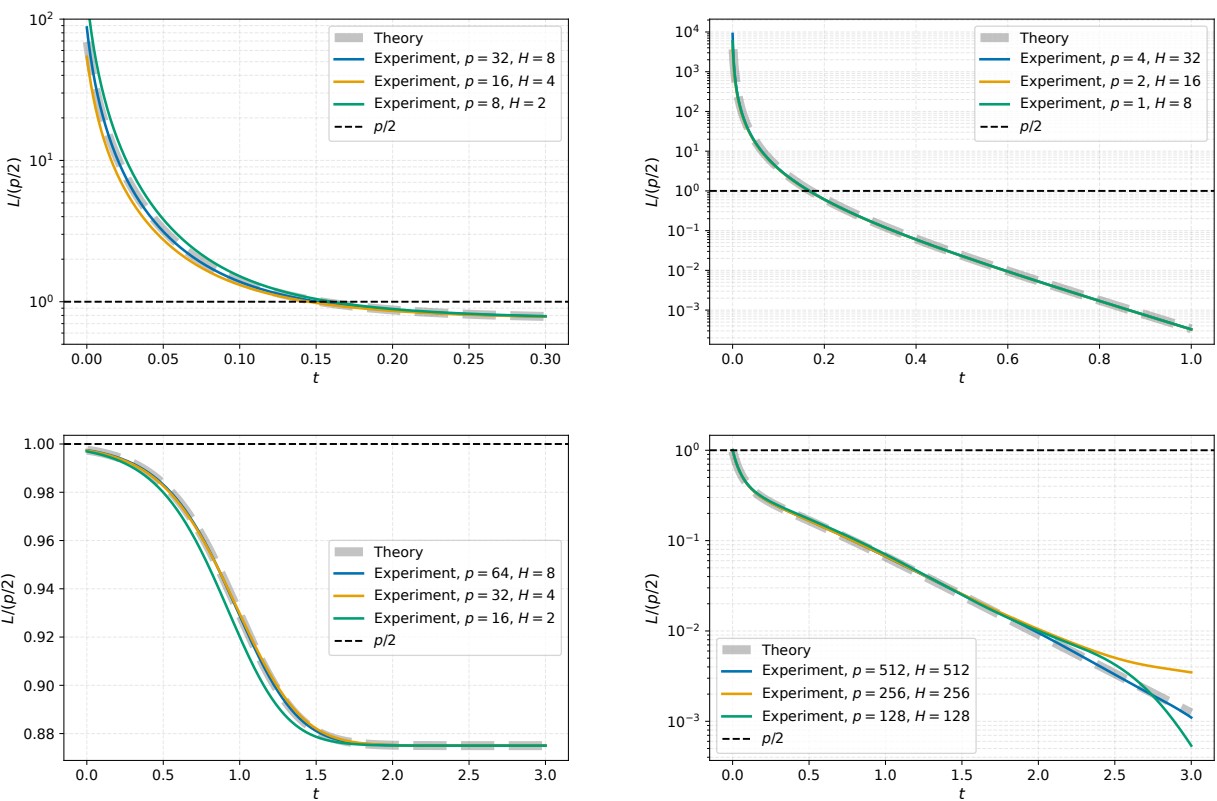

*Figure 12.* Experimental confirmation of our theory in SYM $\nu = 2$. **Upper left:** noisy underparameterized limit. **Upper right:** noisy overparameterized limit. **Lower left:** Low-noise underparameterized limit. **Lower right:** General scenario. See Appendix L.2 for details about the experiments.

