# OpenReview forum: "Gradient Flow Through Diagram Expansions: Learning Regimes and Explicit Solutions"
_ICML.cc/2026/Conference — ICML 2026 spotlight_

### Official Review · Reviewer_2sCc · 2026-03-11

**Soundness:** 3
**Presentation:** 3
**Significance:** 2
**Originality:** 3
**Overall Recommendation:** 5
**Confidence:** 3

**Summary:**

This paper studies gradient flow dynamics on the CP decomposition objective (both symmetric and asymmetric) when the target tensor is the diagonal tensor $F = \sum_i e_i^{\otimes k}$. It analyzes the dynamics by Taylor expanding in $t$ around $t=0$, representing each polynomial in this series by a corresponding diagram, keeping only the dominant terms, and then using the diagrams to take the Gaussian expectations using Wick's formula. The results are heuristic formulas for the asymptotic loss curves, which are then compared to experiments.

**Compliance With Llm Reviewing Policy:**

Affirmed.

**Final Justification:**

I am satisfied by the authors' responses and maintain my positive assessment of the paper.

**Key Questions For Authors:**

- How difficult is it to extend the results in Section 7 to the balanced regime?
- Is there any intuition for why the formulas in section 10 are only valid for $t \le 0$? Could the authors comment on how this approach might extend to $t > 0$?
- Is there hope to extend this approach to non-polynomial settings (e.g. population dynamics of a two layer neural network)?
- In figure 5, the $\rho^\star$ line doesn't appear to line up with the phase transition in the heat map, and there is noticeable error in the high noise plot on the right. Is this just a finite-$p$ error?

**Limitations:**

yes

**Strengths And Weaknesses:**

Strengths:
- To the best of my knowledge, this approach is new and seems relatively powerful
- The paper studies both the symmetric and asymmetric cases and discusses how they differ
- The heuristic derivations are justified by synthetic experiments and the agreement across most of the plots is quite good

Weaknesses:
- Many of the formulas presented in this paper can be derived with significantly simpler heuristics (see below), which challenges the necessity of this formalism. The only result that cannot be re-derived with elementary machinery is Section 10 ($\nu=4$ + gradient ascent).
- The techniques in this paper seem specialized to the tensor setting so that the derivatives at $t = 0$ are all polynomials in $w$. It is not clear how these techniques could generalize to a more general setting (e.g. non-polynomial activations).
- As the authors acknowledge throughout the paper, the final formulas are not rigorously justified and are only heuristics. This is not necessarily a significant weakness if they are predictive.
- While the paper has many results (definitely a strength), this leaves less room in the main text to introduce the tensor diagrams. Page 3 is especially dense and difficult to follow. It may be worth including a section at the start of the appendix which walks the reader through a simple toy computation.

Simpler heuristics:
- **Section 7:** The balanced regime appears nontrivial, but this was deferred to future work (lines 350-351). In contrast, the extreme vertices (A,B) are relatively simple. In the underparameterized regime, each particle will act independently and the squared norm $r$ of each particle satisfies $r' \propto -r^\nu$. Tracking constants recovers the equations for the symmetric and asymmetric cases. In the overparameterized regime, you can make the mean field ansatz that the particles remain isotropic with a time-dependent variance $\sigma^2(t)$, which gives an autonomous equation for $\sigma^2(t)$ that matches lines 341-342. Both of these can be rigorously justified in their corresponding limits.
- **Section 9:** For $\nu=2$ (matrix factorization), you can SVD the initialization and the singular values decouple giving the well known formula: $L = \frac{p}{2} \mathbb{E}_\lambda\left[\left(\frac{1-\lambda}{1 + \lambda (e^{4t} - 1)}\right)^2\right]$ where the expectation is taken with respect to the spectrum of $UU^T$ at initialization. In the limit, this follows a Marcenko-Pastur law and if you substitute its Stieljes transform you get eq. (17). This formula is therefore relatively elementary.

---

> ### Author Rebuttal · Authors · 2026-03-30
>
> We sincerely thank Reviewer 2sCc for the positive evaluation of our work and numerous insightful comments. If our paper is accepted and Reviewer 2sCc happens to attend the conference, we would be delighted to discuss this and related topics in person.
>
> ## Weaknesses:
>
> >Many of the formulas presented in this paper can be derived with significantly simpler heuristics (see below), which challenges the necessity of this formalism. The only result that cannot be re-derived with elementary machinery is Section 10 ($\\nu=4$ + gradient ascent).
>
> That's an insightful observation. Indeed, in many cases there are alternative solutions. However, these solutions are disconnected and applicable on a case-by-case basis. In contrast, our goal has been a framework that would allow to 1) systematically reveal and classify all the scaling regimes, in particular identifying their boundaries; 2) derive the solutions in a standardized unified way.
>
> >While the paper has many results (definitely a strength), this leaves less room in the main text to introduce the tensor diagrams. Page 3 is especially dense and difficult to follow. It may be worth including a section at the start of the appendix which walks the reader through a simple toy computation.
>
> Thank you for this feedback, that's indeed a very reasonable suggestions.
>
> ## Questions:
>
> >How difficult is it to extend the results in Section 7 to the balanced regime?
>
> We know how to solve the balanced regime only in the symmetric even-$\\nu$ scenario.  The extension is obtained differently at $\\nu=2$ and $\\nu\\ge 4$. At $\\nu=2$ the balanced free evolution can be obtained as a degenerate case of the full Narayana/Marchenko-Pastur solution of Section 9. The relevant diagrams are single loops optimally contracted to trees. In contrast, at $\\nu\\ge 4$ the diagrams are multi-looped. However, the relevant counts of optimal pairings/contractions admit an even simpler recurrence than for $\\nu=2$, leading to
> $$\\frac{\\mathbb E[L(t)]}{\\mathbb E[L(0)]} \\sim  \\bigg( 1 + \\frac{2\\nu(\\nu-1) \\left[1 + (\\nu-1)!! \\frac{H}{p^{\\nu/2}}\\right] p^{\\nu-1} \\sigma^{2(\\nu-1)}}{T} t\\bigg)^{-\\nu/(\\nu-1)}.$$
>
> >Is there any intuition for why the formulas in section 10 are only valid for $t\\le0$? Could the authors comment on how this approach might extend to $t>0$?
>
> That's a good question. The issue seems to be the complex distribution pattern of the model degrees of freedom, with emerging singularities, when fitting the identity target tensor for $\\nu>2$. This issue is already present in the extreme underparameterized setting of model rank $H=1$, when the model is just the tensor power $\\mathbf u^{\\otimes \\nu}$ of a weight vector $\\mathbf u\\in\\mathbb R^p$. In this case one of the components of $\\mathbf u$ - the largest in magnitude - evolves to $\\pm 1$, while the others evolve to 0. (At this point one can see the difference between $\\nu=2$ and $\\nu>2$: for $\\nu=2$ the optimal vectors $\\mathbf u$ are all vectors of unit norm, corresponding to a $O(p)$ symmetry, while for $\\nu>2$ this symmetry gets broken to $S_p\\times \\mathbb {Z}^p_2$.) For any fixed $p$ the full gradient ascent/descent trajectory can be described analytically, but it does not seem to have a nice macroscopic (large-$p$) analytic approximation for $t>0$. When using the natural function/parameter $\\psi$ as in our solution, $\\psi$ gets a singularity at $t>0$, and this singularity approaches 0 as $p\\to\\infty$. We don't know yet how to resolve this difficulty. Perhaps a more careful treatment of the sub-optimal monomials in the $H,p\\to\\infty$ limit is needed.
>
> >Is there hope to extend this approach to non-polynomial settings (e.g. population dynamics of a two layer neural network)?
>
> The natural approach would be to formally Taylor-expand the activation function (let us assume it is analytic).
> This would yield again a formal $t$-expansion, but with some infinite sums as its coefficients. It is conceivable that in certain regimes these sums might simplify and yield closed recurrences, leading to solvable PDEs. However, this is of course purely hypothetical; we haven't actually looked into this direction.
>
> >In figure 5, the $\\rho^*$ line doesn't appear to line up with the phase transition in the heat map, and there is noticeable error in the high noise plot on the right. Is this just a finite-$p$ error?
>
> Yes, we believe this is mainly due to finite-$p$ effects. Experiments at larger $p$ are computationally very expensive because of the dimensionality of the problem, so in Fig. 5 we could only use $p=16$ for the left panel (heat map) and $p=64$ for the right panel. We also observed worse agreement at smaller $p$, so we expect the correspondence to improve for larger $p$.

---

> > ### Author Rebuttal · Reviewer_2sCc · 2026-04-04
> >
> > Thank you for answering all of my questions. I am satisfied with the answers and would like to maintain my positive score.

---

### Official Review · Reviewer_Bq13 · 2026-03-12

**Soundness:** 3
**Presentation:** 3
**Significance:** 2
**Originality:** 3
**Overall Recommendation:** 4
**Confidence:** 2

**Summary:**

This paper studies gradient flow through a diagram-expansion framework. It writes the loss trajectory as a formal power series in time, represents the coefficients by diagrammatic objects, and uses the large-size limit of the leading terms to identify different learning regimes. The analysis is developed mainly for CP decomposition, mostly with the identity target, and the paper derives explicit descriptions of free evolution, NTK-like behavior, and mean-field behavior. In some settings, the formal series is further reduced to a PDE, which leads to explicit predictions for the loss trajectory.

**Compliance With Llm Reviewing Policy:**

Affirmed.

**Key Questions For Authors:**

See weaknesses

**Limitations:**

yes

**Strengths And Weaknesses:**

## Strengths

1. The paper gives a diagram-based way to organize the scaling regimes of gradient flow. The Pareto-polygon construction makes the different regimes and their scaling conditions explicit.
2. The paper works out several concrete cases in detail. In particular, it gives explicit results for free evolution, for the SYM $\nu=2$ case, and for the SYM $\nu=4$ gradient-ascent setting.

## Weaknesses:

1. While the proposed diagrammatic expansion is mathematically elegant and provides a rigorous way to track gradient flow, its contribution currently reads more like an exercise in theoretical physics than a tool for machine learning. The paper does not convince the reader why this heavy machinery is necessary if it only recovers known phase transitions (e.g., lazy vs. rich regimes) in simplified tensor models, without offering fundamentally new, actionable insights into the learning dynamics of deep neural networks.
2. Most of the detailed analysis is still concentrated on CP decomposition with the identity target. The paper says the framework may apply more broadly, but that broader applicability is not established in the current version

---

> ### Author Rebuttal · Authors · 2026-03-30
>
> We thank the reviewer for the careful reading and positive evaluation of our work.
>
> >While the proposed diagrammatic expansion is mathematically elegant and provides a rigorous way to track gradient flow, its contribution currently reads more like an exercise in theoretical physics than a tool for machine learning. The paper does not convince the reader why this heavy machinery is necessary if it only recovers known phase transitions (e.g., lazy vs. rich regimes) in simplified tensor models, without offering fundamentally new, actionable insights into the learning dynamics of deep neural networks.
>
> We respectfully acknowledge the reviewers' opinion. Nevertheless, we mention that, apart from the general framework and new analytic solutions, our results include the presence/absence of the NTK regime in ASYM/SYM tensor learning, and the two distinct regimes of gradient ascent, which we believe to be new insights.
>
> >Most of the detailed analysis is still concentrated on CP decomposition with the identity target. The paper says the framework may apply more broadly, but that broader applicability is not established in the current version
>
> We mention a few specific extensions that are in our plans:
>
> 1. Deep linear networks, including studying their generalization on Gaussian datasets.
> 2. Adding weight decay and momentum.
> 3. Studying generalization a CP-decomposition when only a random subset of the target indentity tensor is present.
> 4. Discrete-time gradient descent dynamics; possibly, with stochastic batch sampling (see our response to Reviewer jDft).
> 5. Normalized gradient methods (like RMSProp).
> 6. Neural nets with polynomial activation functions (see our response to Reviewer JMkL).

---

### Official Review · Reviewer_jDft · 2026-03-13

**Soundness:** 3
**Presentation:** 2
**Significance:** 3
**Originality:** 4
**Overall Recommendation:** 4
**Confidence:** 2

**Summary:**

This paper proposes a unified theoretical framework for analyzing gradient-flow dynamics in large-scale learning problems, they consider a time-series expansion of the loss and a Feynman-like diagram calculus for encoding and computing the expansion terms. In the setting of learning the identity tensor via CP tensor decomposition, the authors use Pareto-optimal scaling, Pareto polygons, and PDE-based summation to systematically characterize different learning regimes such as free evolution, NTK, and mean-field, and they derive explicit loss dynamics in several cases.

**Compliance With Llm Reviewing Policy:**

Affirmed.

**Final Justification:**

I appreciate the novelty of this paper, but I hope the authors can present it more clearly so that the main ideas are easier for readers to follow.

**Key Questions For Authors:**

See Weakness

**Limitations:**

Yes

**Strengths And Weaknesses:**

# Strengths
1. This paper introduces an original and fairly unified analytical framework for gradient-flow dynamics in ML models. The combination of diagram expansions, Pareto-optimal scaling analysis, and PDE summation is technically distinctive.
2. This paper obtains several explicit solutions and concrete theoretical predictions, some of which are non-obvious and experimentally validated.

# Weaknesses
1.  This paper is written using concepts and terminology that seem to assume a statistical-physics background, which makes soundness hard to assess for a general ML audience.
2. The summation of the formal large-size series and the PDE reduction remains formal rather than fully rigorous, with no general convergence or error analysis.
3. This paper studies an asymptotic gradient-flow regime, but does not address finite-width behavior or how the theory extends to discrete-time or stochastic optimization.

---

> ### Author Rebuttal · Authors · 2026-03-30
>
> We thank the reviewer for reading the paper and providing feedback. However, we respectfully do not agree with some of the weaknesses, and we do not see why they justify the "weak reject" score.
>
> >This paper is written using concepts and terminology that seem to assume a statistical-physics background, which makes soundness hard to assess for a general ML audience.
>
> We stress that our exposition **does not assume prior knowledge of physics**. The reviewer might have the wrong impression that it does because of our references to Feynman diagrams and physics in the introduction and conclusion. We do discuss these connections, but only to provide a broader context of our work. The exposition of our method is, however, purely mathematical, and formulated using standard mathematical concepts (polynomials, power series, graphs, generating functions, differential equations, etc.) We invite the reviewer to point out a specific place in our derivations that cannot be understood without a physics background.
>
> >The summation of the formal large-size series and the PDE reduction remains formal rather than fully rigorous, with no general convergence or error analysis.
>
> Indeed, we repeatedly acknowledge in the paper that our approach is not entirely rigorous. However, we stress that our approach is systematic, produces nontrivial predictions, and these predictions are confirmed by experiment. Moreover, some essential parts of our approach are rigorous (e.g., Theorems 3.1, 4.1, 6.1). We don't think that a method with such qualities has to be rejected just because it is not entirely rigorous. Moreover, we view the rigorous justification of the empirically successful description of the large-model dynamics by the formal limiting power series as an interesting open problem raised by our paper and having value in its own right.
>
> >This paper studies an asymptotic gradient-flow regime, but does not address finite-width behavior
>
> We stress that, contrary to the statement, our paper **directly addresses finite-width behavior**. In our setting, the role of network width is played by the CP rank parameter $H$. Throughout the paper, we assume $H$ to be large but finite, and $H$ appears in most of our solution formulas. Figure 4 even compares several solutions corresponding to different $H$, showing curves that are clearly different but nevertheless very well matching  the respective empirical curves.
>
> We suspect that the reviewer may have in mind the special case of the NTK regime, in which there is a well-defined $H=\infty$ limiting dynamics corresponding to a simple linear kernel model. In this case the interesting question is indeed the deviation from this limit resulting from a finite width. However, our setting is more general, and addresses various regimes that do not necessarily have a well-defined $H=\infty$ limit. In these cases the leading term at $H\to \infty$ is nontrivial and is the main object of study. Regarding the NTK regime, while we did not focus in our paper on the finite-width corrections, we believe this could also be done by a suitable extension of our method, by examining next-to-leading terms in the expansion.
>
> >how the theory extends to discrete-time or stochastic optimization.
>
> Our diagrammatic method can in principle be applied to gradient descent with a finite step size. Indeed, one gradient step can be expressed in terms of polynomials in the weights, and a sequence of steps can also be expressed in this way by unrolling. The relevant polynomials can again
> be described by diagrams, and the expected loss can be evaluated using Wick's formula.
>
> The advantage of considering continuous gradient flow is that the corresponding loss function is always a generating function of a specific sequence.
> It is not obvious whether there is such a nice parallel in the discrete case.
>
> On the other hand, discrete optimization steps allow to naturally consider the batch-wise stochastic gradient descent.
> In our terms, the training batch of size $B$ at time $t$ could be represented as a matrix $X_t \\in \\mathbb{R}^{p \\times B}$.
> One then unrolls the loss function under the gradient descent dynamics as described above, assuming independent Gaussian $X_t$ at each time step.

---

> > ### Author Rebuttal · Reviewer_jDft · 2026-04-02
> >
> > Thanks for your detailed response,  after I read the paper again, I appreciate the novelty of this paper, but I hope the authors can present it more clearly so that the main ideas are easier for readers to follow. I will increase my score to 4.

---

### Official Review · Reviewer_JMkL · 2026-03-13

**Soundness:** 4
**Presentation:** 4
**Significance:** 4
**Originality:** 4
**Overall Recommendation:** 5
**Confidence:** 3

**Summary:**

This paper analyzes the dynamics of gradient flow, in the setting of CP decomposition, where the target is the identity tensor of any order, and of p dimensions (and is thus of rank p). The model is a rank-H tensor. The goal is to analyze how $\mathbb{E}[L(t)]$ evolves over time, where L(t) is the loss. The first step is to perform a Taylor expansion of L(t), and its expectation, in terms of t. The derivatives can be calculated using Wick’s formula, and are polynomial functions of the parameters. A key innovation of this paper is introducing diagrams to represent polynomials in the weights. They then expand the loss, which can be represented as a diagram. Then, the higher-order derivatives of L(t) can be computed using diagrams. This can be done using the binary merging operation, which is intuitive. Finally, for the $s^{th}$ order derivative of L(t), evaluated at 0, the expectation of this quantity can be computed using Wick’s formula.

The goal is next to analyze how the polynomials $Y_s$ grow as dimension p and rank H grows. This requires identifying the Pareto-optimal terms in the polynomials (i.e. terms where the powers are not strictly dominated by any other term). Theorem 4.1 thus identifies the Pareto-optimal terms, in the case of the identity target. Theorem 4.1 shows that the powers of the Pareto-optimal terms form a 2D polygon in R^3. The polygon has a normal vector, which corresponds to a special scaling conditions where all the Pareto terms are leading. This corresponds to a rich regime. In general, each lower-dimensional face of the polygon corresponds to a certain scaling behavior. For example, the terms on the right edge of the polygon are dominant when the initialization is large. In this regime, the model attempts to converge to 0 (since the target is relatively small).

The ultimate goal is to study the limit $p, H \rightarrow \infty$. Thus, it requires taking the limit of each coefficient, and then summing the series. They find that in the free evolution case (where the target is small relative to the initialization), in the under-parameterized case, the loss decreases according to a power law in t, while in the over-parameterized case, the loss decreases exponentially for asymmetric CP decomposition, and according to a power law for symmetric CP decomposition. These predictions match the experimental results, as shown in Figure 3. In the case of symmetric tensors of order 2, they give an exact formula for $\mathbb{E}[L(t)]$. The theory matches the experiments in this case as well.

Note: In the case of symmetric tensors of order 4, the method produces a formula for $\mathbb{E}[L(t)]$ which only applies when t < 0, i.e. in the case of gradient descent, since the summation diverges otherwise. If the noise is low relative to p, then the loss will converge to p/2, while if the noise is high, the loss will diverge.

**Compliance With Llm Reviewing Policy:**

Affirmed.

**Final Justification:**

I am maintaining my score of 5, as this is a rigorous paper with interesting proof techniques.

**Key Questions For Authors:**

- Is it possible to briefly give more insight into the proof of Theorem 4.1?
- The roadmap in the intro is hard to understand without knowledge of the paper itself.
    - E.g. it would be useful to define “Pareto-optimal terms” in this roadmap, in a few words.
- Could you outline how the generalization to neural networks with polynomial activations might work?
- Is there a connection between the diagrams proposed by this paper, and tensor networks, which are used to represent tensor contractions?
    - See https://arxiv.org/abs/1609.00893 for more details.
- There is perhaps one weakness of the symmetric order 4 setting, which is that it is not clear whether gradient descent can be analyzed or not.
    - Do you have a conjecture on whether it is possible to analyze the symmetric order 4 setting with these methods?

**Limitations:**

yes

**Strengths And Weaknesses:**

Strengths
- The paper is technically sound, with empirical results matching the theoretical predictions. Additionally, it is very well-written overall.
- The result seems novel, as the authors state that their result is not covered by existing work on tensor programs.
- The techniques introduced by the paper are very interesting, particularly the use of diagrams to denote terms in the Taylor expansion, and the classification of the Pareto-optimal terms.

Weaknesses
- Perhaps there could be more detail given on the exact limitations of the technique so far. E.g. is it likely that a similar technique could help to analyze gradient descent for symmetric tensors of order 4, or is it fundamentally limited?

---

> ### Author Rebuttal · Authors · 2026-03-30
>
> We sincerely thank the reviewer for the very careful reading of our work and its very positive evaluation. If our paper is accepted and Reviewer JMkL happens to attend the conference, we would be delighted to discuss with them this and related topics in person.
>
> >Is it possible to briefly give more insight into the proof of Theorem 4.1?
>
> Basically, the proof consists in finding out which diagrams appearing in the merger $(\\tfrac{1}{2}D_{2\\nu}+R_\\nu)^{\\star(s+1)}$ can support an edge pairing that leaves the given numbers $q,n$ of contracted $p$- and $H$-nodes. The easier part of the proof is to show that for particular $(q,n)$ there are suitable diagrams: this can be done by explicit construction. The trickier, complementary part is to show that for particular $(q,n)$ there are no diagrams. This is done by induction on the number $s_D$ of free diagrams $D_{2\\nu}$ appearing in the merger: we show that if there is a diagram with $s_D$ factors $D_{2\\nu}$ and some numbers of contracted nodes, then there must be a diagram with $s_D-1$ factors $D_{2\\nu}$ and suitably  decremented numbers of nodes. Also, parity considerations allow to rule out exceptional cases: diagrams merged using an odd number of interaction diagrams $R_\\nu$, and free diagrams with fully uncontracted $H$-nodes.
>
> In fact, the whole proof simplifies greatly for $\\nu = 2$.
> In this case, all diagrams that appear in the merger $(\\tfrac{1}{2}D_{2\\nu}+R_\\nu)^{\\star(s+1)}$ are loops of length $2 s_D + 2$ with alternating $p$- and $H$-vertices.
> In the symmetric scenario, there is only one color, hence all Pareto-optimal contractions are trees.
> As is easy to see, one can get any number $n$ of $H$-vertices from 1 to a $s_D + 1$ after contraction.
> As $s_D$ runs from zero to $s+1$, we get a triangle.
> In the asymmetric scenario, there are two colors.
> If $s_R$ is odd then the loop does not allow for a valid contraction.
> If $s_R$ is even and at least two then one can merge all $D$-diagrams keeping only two edges of a given color, then recolor these two edges to get a loop of one color and proceed as for the symmetric case.
> Finally, if $s_R$ is zero then we merge $D$-diagrams in such a way that all edges of a given color form a single connected component; then we can get any number $q$ of $p$-vertices from $2$ to $s_D+1$ after contraction.
> We cannot, however, contract all $p$-vertices into one, since any vertex with adjacent edges of different colors has two be of type $H$ when no $R$-diagrams are present in the merger.
> Because of this, there is no contraction keeping all $H$-vertices.
>
> >it would be useful to define “Pareto-optimal terms” in this roadmap, in a few words.
>
> Thank you for the suggestion, we will expand all the necessary notions in the intro in a revised version of our submission.
>
> >Could you outline how the generalization to neural networks with polynomial activations might work?
>
> Let us consider a network with two layers and a square activation function: $f_k(x) = \\sum_{j=1}^H V_{k,j} (\\sum_{i=1}^p U_{ji} x_i)^2$ for $k \\in [p]$.
> For the sake of brevity of the exposition, let us consider its free evolution on a dataset $X \\in \\mathbb{R}^{p \\times N}$ of $N$ iid samples from $\\mathcal{N}(0, I_p)$.
> Then the loss function is
> $$
> L(U,V) = \\sum_{l=1}^N \\sum_{k=1}^p (\\sum_{j=1}^H V_{k,j} (\\sum_{i=1}^p U_{ji} X_{il})^2)^2.
> $$
> It could be expressed as the following analog of the $D$-diagram, with three types of vertices, $p, H$, and $N$, and three types of edges, $U, V$, and $X$.
> There is a single $N$-vertex with four adjacent $X$-edges, with distinct $p$-vertices at their ends.
> Each of these $p$-vertices has a single adjacent $U$-edge, connecting the four $p$-vertices to two $H$-vertices, two to each vertex.
> Finally, those two $H$-vertices are connected to a common $p$-vertex with a $V$-edge each.
>
> >Is there a connection between the diagrams proposed by this paper, and tensor networks?
>
> While both our diagrams and tensor networks (TN) provide graphical representations of tensor contractions, there is a difference in the convention: in TN, tensors are vertices and indices are edges, whereas in our diagrams the variables are edges and the contraction indices are vertices. This choice is natural in our setting because the model parameters are matrices, so representing them as edges is especially convenient. We also emphasize that the role of the diagrams is different: TN are mainly used as general tools for the problems like compressed tensor representation, while in our work the diagrams are introduced to study gradient flow learning dynamics.
>
> >Do you have a conjecture on whether it is possible to analyze the symmetric order 4 setting with these methods?
>
> That's a very good question. We think that it is possible, but certainly requires some extension of our framework; perhaps a more careful treatment of sub-leading terms in the expansion. Please also see our answer to the similar question by Reviewer 2sCc.

---

> > ### Author Rebuttal · Reviewer_JMkL · 2026-04-04
> >
> > I appreciate the detailed response by the authors. I will keep my current score.

---

### Decision · Program_Chairs · 2026-04-30

**Decision:**

Accept (spotlight)

**Comment:**

This submission develops a mathematical framework for analyzing the gradient flow dynamics of Canonical Polyadic (CP) decompositions. The authors perform a Taylor expansion in time and compute the polynomial coefficients via a graph representation akin to Feynman diagrams. While the theory is heuristic and not rigorously justified, all reviewers appreciate its technical novelty and predictive power. The meta-reviewer therefore recommends acceptance.